# Global supply chains amplify economic costs of future extreme heat risk

Yida Sun[1,14], Shupeng Zhu[2,3,14], Daoping Wang[4,5,14], Jianping Duan[6], Hui Lu[1,7], Hao Yin[8], Chang Tan[1], Lingrui Zhang[9], Mengzhen Zhao[10], Wenjia Cai[1], Yong Wang[1], Yixin Hu[11], Shu Tao[12] & Dabo Guan[1,13 ✉]

Evidence shows a continuing increase in the frequency and severity of global heatwaves[1,2], raising concerns about the future impacts of climate change and the associated socioeconomic costs[3,4]. Here we develop a disaster footprint analytical framework by integrating climate, epidemiological and hybrid input–output and computable general equilibrium global trade models to estimate the midcentury socioeconomic impacts of heat stress. We consider health costs related to heat exposure, the value of heat-induced labour productivity loss and indirect losses due to economic disruptions cascading through supply chains. Here we show that the global annual incremental gross domestic product loss increases exponentially from 0.03 ± 0.01 (SSP 245)–0.05 ± 0.03 (SSP 585) percentage points during 2030–2040 to 0.05 ± 0.01–0.15 ± 0.04 percentage points during 2050–2060. By 2060, the expected global economic losses reach a total of 0.6–4.6% with losses attributed to health loss (37–45%), labour productivity loss (18–37%) and indirect loss (12–43%) under different shared socioeconomic pathways. Small- and medium-sized developing countries suffer disproportionately from higher health loss in South-Central Africa (2.1 to 4.0 times above global average) and labour productivity loss in West Africa and Southeast Asia (2.0–3.3 times above global average). The supply-chain disruption effects are much more widespread with strong hit to those manufacturing-heavy countries such as China and the USA, leading to soaring economic losses of 2.7 ± 0.7% and 1.8 ± 0.5%, respectively.

Research has been showing a trend in rising temperature and increasing occurrence of extreme heatwaves since the 1950s[1,2]. This continuous pattern raises concerns about the potential impacts of climate change and its associated socioeconomic costs. Notable effects of heat stress are on human health and labour productivity. On the one hand, global heat stress makes it difficult for the body to maintain its core temperature, thereby increasing morbidity and mortality from heat stroke[5–7]. Countries across all latitudes, including Russia[8], the USA[9], China[5], Australia[10] and North Africa[11] have suffered from increased heat stress since the deadly European heatwave in 2003[12], which caused considerable mortality and morbidity. On the other hand, biometeorological studies suggest that heat stress can seriously decrease labour productivity[13–16], measured in terms of lost worktime from recommended work/rest ratios during heat stress, reduced work efficiency as estimated from exposure–response functions and self-reported reduced work efficiency[13,17,18].

In the context of increasingly integrated global supply chains, the impacts of heat stress are not just confined to specific populations and industrial sectors in low latitudes but extend to wider regions and sectors[19–22]. For example, a Western European country such as the UK is rarely directly and severely affected by heat stress. However, consumption of beer or coffee in the UK can drop as a result of the severe impact of heat stress on wheat and coffee bean suppliers in Africa and South America[23]. This kind of spillover effect can have important consequences in terms of global food security[24–26], energy supply[27] and the supply of various mineral products[28].

The direct mortality and productivity loss resulting from heat stress have been extensively studied. However, the indirect losses due to supply-chain disruptions have not been fully analysed[29,30], as previous literature has either devoted insufficient discussion to the indirect effects by only reporting the total/aggregated effects[31–33] or ignored the amplifying effect of the global trade system on direct losses. As climate change will make the impacts of heat stress worse over time, developing

[1]Department of Earth System Science, Ministry of Education Key Laboratory for Earth System Modeling, Institute for Global Change Studies, Tsinghua University, Beijing, China. [2]Department of Atmospheric Sciences, School of Earth Sciences, Zhejiang University, Hangzhou, China. [3]Advanced Power and Energy Program, University of California Irvine, Irvine, CA, USA. [4]Department of Geography, King's College London, London, UK. [5]Centre for Climate Engagement, Department of Computer Science and Technology, University of Cambridge, Cambridge, UK. [6]State Key Laboratory of Earth Surface and Ecological Resources, Faculty of Geographical Science, Beijing Normal University, Beijing, China. [7]Tsinghua University (Department of Earth System Science)—Xi'an Institute of Surveying and Mapping Joint Research Center for Next-Generation Smart Mapping, Beijing, China. [8]Department of Economics, University of Southern California, Los Angeles, CA, USA. [9]Department of Economics, University of Waterloo, Waterloo, Ontario, Canada. [10]School of Management and Economics, Beijing Institute of Technology, Beijing, China. [11]School of Economics and Management, Southeast University, Nanjing, China. [12]College of Urban Environment, Peking University, Beijing, China. [13]The Bartlett School of Sustainable Construction, University College London, London, UK. [14]These authors contributed equally: Yida Sun, Shupeng Zhu, Daoping Wang. ✉e-mail: guandabo@tsinghua.edu.cn

methodologies that allow comprehensive quantifications of both the direct and indirect impacts of heat stress on human systems can help policy-makers to develop more effective climate change mitigation and adaptation policies. In this study, a disaster footprint analytical framework, by integrating climate, epidemiological and hybrid input–output and computable general equilibrium global trade modules, was constructed to provide a comprehensive assessment of the impact of heat stress on socioeconomic systems to 2060, including health loss (excess mortality due to extreme heatwaves), labour productivity loss (decreased daily labour productivity due to higher temperature and humidity) and indirect loss (production stagnation due to lack of supply or demand) across 141 regions and 65 sectors worldwide. Details of our analytical approach are provided in the Methods. In summary, we use the sixth phase of the coupled model intercomparison project phase 6 (CMIP6)[34,35], where 14 widely applied global climate models (GCMs) are averaged to assess future daily temperature and humidity parameters. Grid-scale daily excess mortality (health loss) and labour loss rates (labour productivity loss) are calculated on the basis of empirical functions and statistics from previous studies[36]. On the basis of the above labour constraints in different regions and industries, a hybrid input–output and computable general equilibrium global trade module was developed in the disaster footprint analytical framework to assess the pattern of heat-related economic losses transmitted through the global supply chain. By quantifying indirect effects that were hardly analysed before, this model provides insight into the far-reaching impacts of heat stress across global supply chains and how such impacts evolve spatially and over long time scales. The estimated results are based on static production and trade relationships which may not accurately address the dynamic nexus among industries and countries in the long-term.

This study examines three scenarios combining various representative concentration pathways (RCPs) and shared socioeconomic pathways (SSPs). RCPs represent greenhouse gas concentration trajectories as adopted by the Intergovernmental Panel on Climate Change (IPCC). Each RCP scenario implies different magnitudes of future heat stress. SSPs represent socioeconomic development pathways. Different SSP scenarios imply different amounts of risks of heat stress exposure and societal adaptive capacity. Three SSP–RCP scenarios were considered: SSP 585, SSP 245 and SSP 119. Scenario SSP 585 represents a world of rapid and unconstrained growth in economic output and energy use. Scenario SSP 245 represents the middle of the range of plausible future pathways[37], reflecting the continuation of historical mitigation efforts[38]. In scenario SSP 119, the world shifts pervasively toward a more sustainable path, emphasizing more inclusive development that respects perceived environmental boundaries. These three scenarios, from high carbon to sustainable trajectories, allow the quantification of the potential economic benefits of ambitious emissions reduction policies that have previously received little attention.

## Nonlinear growth trend of global heat-related losses

Figure 1a–d depicts the total global economic loss and the specific components. Under the SSP 119 scenario, the total global gross domestic product (GDP) loss is 0.9% (0.6–1.1%) in 2040 and each component is estimated as follows: health loss (0.5%), labour productivity loss (0.3%) and indirect loss (0.1%). In 2060, global GDP loss slightly decreases to 0.8% (0.4% health loss, 0.3% labour productivity loss and 0.1% indirect loss), amounting to about US $3.75 trillion (values are constant 2020 price). The number of global average heatwave days (definition and calculation detailed in the Methods) would increase by 24% compared to 2022 and the average annual number of heatwave deaths would be around 0.59 million (0.44–0.74 million). In the case of a high-emissions, high-growth development path, SSP 585, economic losses in 2060 increase by 500% compared to the SSP 119 scenario, up to 3.9% (2.9–4.5%) (1.6% health loss, 0.8% labour loss and 1.5% indirect

loss), with a value of about US $24.70 (18.36–28.80) trillion. The global annual heatwave days would be 104% higher compared to 2022 and the global average annual number of heat-induced deaths would increase to around 1.12 million (0.85–1.39 million). The labour and health loss on regional and global scales are close to the results of previous studies[39–41].

Global economic losses show a nonlinear growth trend with respect to time and degree of heat stress, driven by increased indirect losses. Over time, total losses grow from 1.5% of GDP in 2040 to 2.5% of GDP in 2050 and to 3.9% of GDP in 2060 (Fig. 1d) under the SSP 585 scenario. However, the proportion of global GDP loss due to supply-chain disruptions is 0.1%, 0.3%, 0.7% and 1.5% per decade from 2030 onwards (Fig. 1c), showing an exponential-like growth pattern (Extended Data Fig. 5 and Supplementary Figs. 7 and 8). Growing indirect losses gradually become the dominant contributor to total losses. Looking at the scenario scale, total GDP losses in 2060 are 0.8% under SSP 119, 2.0% under SSP 245 and 3.9% under SSP 585, of which the losses due to indirect effects are 0.1% of global GDP (13% of total) under SSP 119, 0.5% (25% of total) under SSP 245 and 1.5% (38% of total) under SSP 585. As the degree of heat stress increases progressively, the indirect effects gain more weight in the total losses.

Figure 1e–p explains the mechanism behind the growing weight of the indirect effect in the total losses as the degree of heat stress increases: in terms of spatial patterns, when direct losses are of low to medium magnitude, their impact on the supply network is limited to the regional area; however, when direct losses are severe, they have wider ramifications impacting the supply chain globally and giving rise to further, indirect, losses. Under the SSP 119 scenario, health losses are most significant in South-Central Africa and Eastern Europe (Fig. 1e); labour productivity losses are concentrated in lower latitudes, including West Africa and South Asia (Fig. 1f); indirect losses are concentrated in Central America and East Asia (Fig. 1g); in general, Central and Southern Africa, Southeast Asia and Latin America have the most severe total losses (Fig. 1h). The spatial patterns of direct economic loss of labour and health under the three scenarios are similar. However, it is noteworthy that persistent and severe heat stress expected under the rapid growth SSP 585 scenario leads to substantial disruptions beyond the regional scale through to global value chains (GVCs). Countries such as Brazil, China and Norway all suffer substantial economic ripple losses. China's indirect economic losses due to supply-chain disruptions soar from 0.4% under SSP 119 to 2.7% of GDP under SSP 585, Brazil from 0.2% to 2.5% and Norway from less than 0.1% to 2.1%. Although developed countries at high latitudes can mitigate most potential losses through adaptation strategies such as air conditioning under SSP 119 scenario, they remain exposed to risk of declining supply or demand in the GVCs under SSP 585 scenario (Fig. 1o and Supplementary Fig. 10). European Union (EU) countries will face considerable indirect losses due to their trading partners' reduced production capacity of minerals and food products, especially developing countries. Although severely affected countries in South Asia or Africa are not core trading partners of the EU and trade volumes between such countries and the EU are relatively small, indirect economic losses in the EU will be amplified when many of those developing countries are affected by heat stress.

## Different sensitivities to heat stress across countries

Different economies face different risk of losses from heat stress, depending on their geographical locations and the position they occupy in the global supply chain. First, countries whose densely inhabited districts are expected to suffer from severe future warming and temperature anomalies, are the most vulnerable to health losses in terms of excess mortality. Under the SSP 119 scenario, South-Central Africa's GDP loss due to heatwave deaths is 1.8% (1.2–2.5%) in 2060, the highest in the world. It is followed by Trinidad and Tobago (1.7%), Sri Lanka (1.5%) and Indonesia (1.5%; Fig. 2a). Vulnerability to health impacts depends on the frequency of extreme weather events and

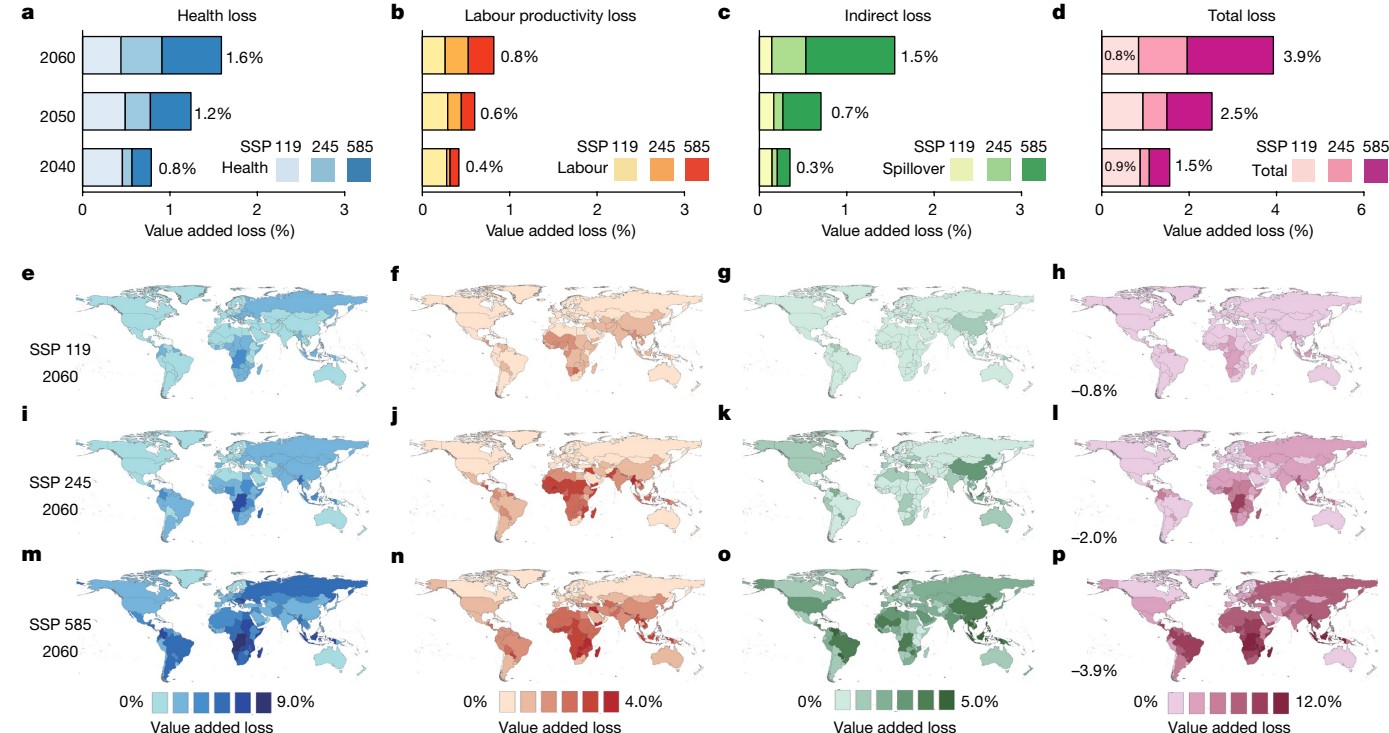

**Fig. 1 | Global heat-related losses in the midcentury and their distributions across the world. a–d**, Evolutionary trends of the four types of losses from 2040 to 2060 under different scenarios (health loss (**a**); labour productivity loss (**b**); indirect loss (supply-chain disruptions) (**c**); and the total losses (**d**)). The colours from light to dark, represent the economic losses from the three scenarios SSP 119, SSP 245, SSP 585, respectively. **e–p**, The spatial distribution of global losses as a percentage of each country's GDP at midcentury under the SSP 119 (**e–h**), SSP 245 (**i–j, l**) and SSP 585 (**m–p**) scenarios. The values shown are 10-year averages (for example, loss reported in 2060 represents the average loss calculated over the period between 2055 and 2065).

the amount of adaptive capacity. For example, Hungary and Croatia suffer considerable health losses, even though in these countries the climate is cooler than in the Middle East and North Africa. Unlike labour losses, which occur in regions with very high average temperature and humidity, health losses depend largely on the variance and abrupt changes in summer temperatures. As climate change will lead to more frequent and intense heatwaves, populations in cooler climatic zones will experience considerable loss of life if the adaptive capacity does not keep pace with the abrupt and sudden changes.

Second, low-income emerging economies in the warmest climatic zones are more likely to suffer labour productivity losses. Under the SSP 119 scenario for 2060, countries such as Botswana, Nepal and Nigeria suffer substantial labour productivity losses, up to 1.3%, 1.2% and 1.2% of GDP, respectively (Fig. 2b). These emerging economies are predominantly located in southern and western Africa (except Nepal), where scorching climates combined with substantial warming over time result in labour-intensive activities during summer months being conducted under increasingly high temperatures. To add insult to injury, most of these countries depend on primary industries such as agriculture, forestry, mining and construction, where workers are mostly outdoors and will be severely affected by extreme heat. For example, agriculture accounts for 21.3% of Nepal's GDP and 23.4% of Nigeria's, whereas mining contributes to nearly 28% of Botswana's GDP[42]. The widespread suspension and reduction of production in the agroforestry and extractive industries due to heat stress will have serious repercussions on national economies and international trade balances. Consequently, these countries are among the most affected by the loss of labour productivity.

Third, small to medium-sized economies with strong and diverse connections to the most affected regions in the GVC, are highly vulnerable to indirect effects. In the context of the SSP 119 scenario, value chains in Latin America and Southeast Asia are the most severely affected (Fig. 2c). Puerto Rico suffered the highest losses, estimated at 0.8% (0.5–1.1%) of GDP, whereas Venezuela, Malaysia and other Latin American countries, including El Salvador, Panama and Dominican Republic, lost approximately 0.4–0.8% of GDP. Under the SSP 585 scenario, Southeast Asian economies such as Brunei, Malaysia, Singapore and Indonesia suffer the most. These losses stem from strong trade connections with highly vulnerable countries. For example, Brunei and Singapore are exposed to indirect effects as they import nearly 60% of their annual mineral and metal products from China, Malaysia and Indonesia. Caribbean countries like Puerto Rico and Panama generally have less economic diversity and depend heavily on the service sector and international trade. The complex mechanism of transmitting losses along the value chain necessitates thorough consideration by countries for managing future risk of instability across critical industries.

A comparison between the SSP 119 (Fig. 2a–c) and SSP 585 (Fig. 2d–f) scenarios shows that losses do not increase uniformly across developing and developed countries when faced with severe climate change impacts, indicating uneven exposures to climate risk. Under the high-emission SSP 585 pathway, a substantial portion of the rapidly escalating economic losses is shouldered by developing countries. Despite Africa contributing less than 5% of global greenhouse gas emissions, 12 countries in the continent, including Rwanda, Botswana, Uganda and Malawi, are projected to suffer some of the most substantial economic losses globally by the midcentury. Several East African countries such as Malawi, Madagascar and Tanzania are highly expected to suffer labour productivity losses of approximately 2.5–4.0% of GDP. Regarding health losses, South-Central Africa and Rwanda experience GDP losses of 8.6% and 7.2%, respectively, almost five times more compared to the SSP 119 scenario. In the SSP 585 scenario, indirect losses become more widespread, affecting both developed and developing economies. Brunei incurs the highest indirect losses at 4.7% (4.0–5.3%)

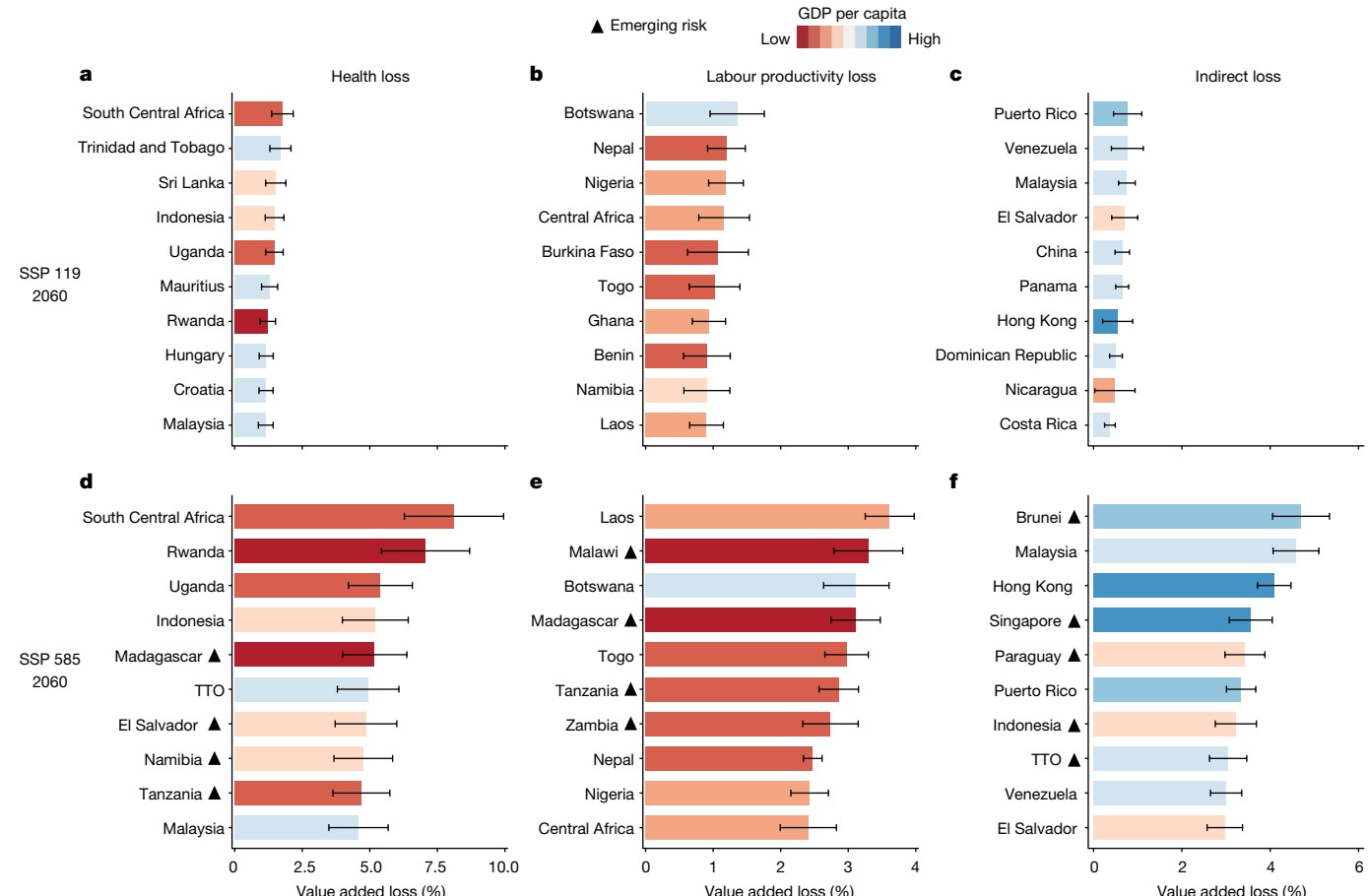

**Fig. 2 | Top ten regions with the severest losses by type under SSP 119 and SSP 585 scenarios. a–c**, Top ten climate change-sensitive regions with the most severe health losses (**a**), labour productivity losses (**b**) and indirect losses (**c**) in 2060, under SSP 119 scenario. **d–f**, Top ten climate change-sensitive countries with the most severe health losses (**d**), labour productivity losses (**e**) and indirect losses (**f**) in 2060, under SSP 585 scenario. The countries marked with black triangles are newly ranked among the most vulnerable countries in 2060 under the SSP 585 scenario compared with SSP 119. The values shown are 10-year averages. Error bars represent 1 s.d. from the mean of decadal data. Upper and lower limits indicate mean + s.d. and mean − s.d., respectively. TTO, Trinidad and Tobago.

of its GDP, whereas other emerging economies like Paraguay and Indonesia lose around 3.3% of their GDP. These findings demonstrated that the rapid growth of income and air-conditioning penetration in emerging economies under the SSP 585 scenario falls short of counteracting the immense impact of climate change on their economies.

## Asymmetric effects of heat stress on global supply chains

Figure 3 highlights the three types of losses for sectors experiencing the highest losses across representative countries. The crop farming, construction and mining sectors are the most affected in most countries, especially in several African and Asian countries that rely on primary industries. For instance, an average summer wet bulb globe temperature (WBGT) above 30 in Tanzania challenges the ability of most outdoor workers to adapt in the midcentury. Sectors requiring workers to be directly exposed to sunlight, such as construction and farming, will suffer a loss of value-added (VA) of 1.9% in 2040 under the SSP 119 scenario. In 2060, rising incomes and a stable climate will result in a slightly reduced VA loss of 0.3% explained by lower labour productivity and health losses. However, under the high warming SSP 585 scenario, the same VA loss increases to 3.9% in 2040 and soars to 8.1% in 2060. In addition, most indoor manufacturing industries suffer a VA loss of 6.0–7.4% in 2060 under the SSP 585 scenario. As demonstrated in Extended Data Fig. 9, countries like Tanzania, Zimbabwe and other

African countries exhibit similar patterns of loss. Countries with comparable loss patterns tend to be situated at low latitudes, particularly in the Middle East, South Asia and Africa—regions most threatened by climate change. Most indoor manufacturing and service industries in developing countries have limited access to air conditioning and, as a result, labour capacity and economic development will be severely undermined by climate change.

Non-metallic products and ferrous metals are vulnerable to climate change because of simultaneous supply-chain shocks from both upstream (supply) and downstream (demand). For example, a country such as India is affected directly by high temperatures and indirectly by the close links with countries severely impacted by heat stress. In 2040, losses in non-metallic manufacturing are second only to construction and agriculture sectors at 2.2% of sectoral VA, whereas ferrous metals industry loses 1.4% of VA. These losses can be attributed to both insufficient demand in the domestic construction sector and shortage of minerals and coal supplies from countries in Southeast Asia and Africa (for example, Indonesia and South Africa). In 2060, under SSP 585 scenario, with the increasingly frequent shutdowns in mining and construction industries under extreme summer heat stress, the ferrous metals industry in India suffers the most substantial VA loss at 5.0%, of which more than 70% is due to indirect losses, followed by the loss from non-metallic manufacturing industry at 3.9%. The sectoral patterns of loss in India are characterized by a combination of health, labour and indirect losses. The decline in labour

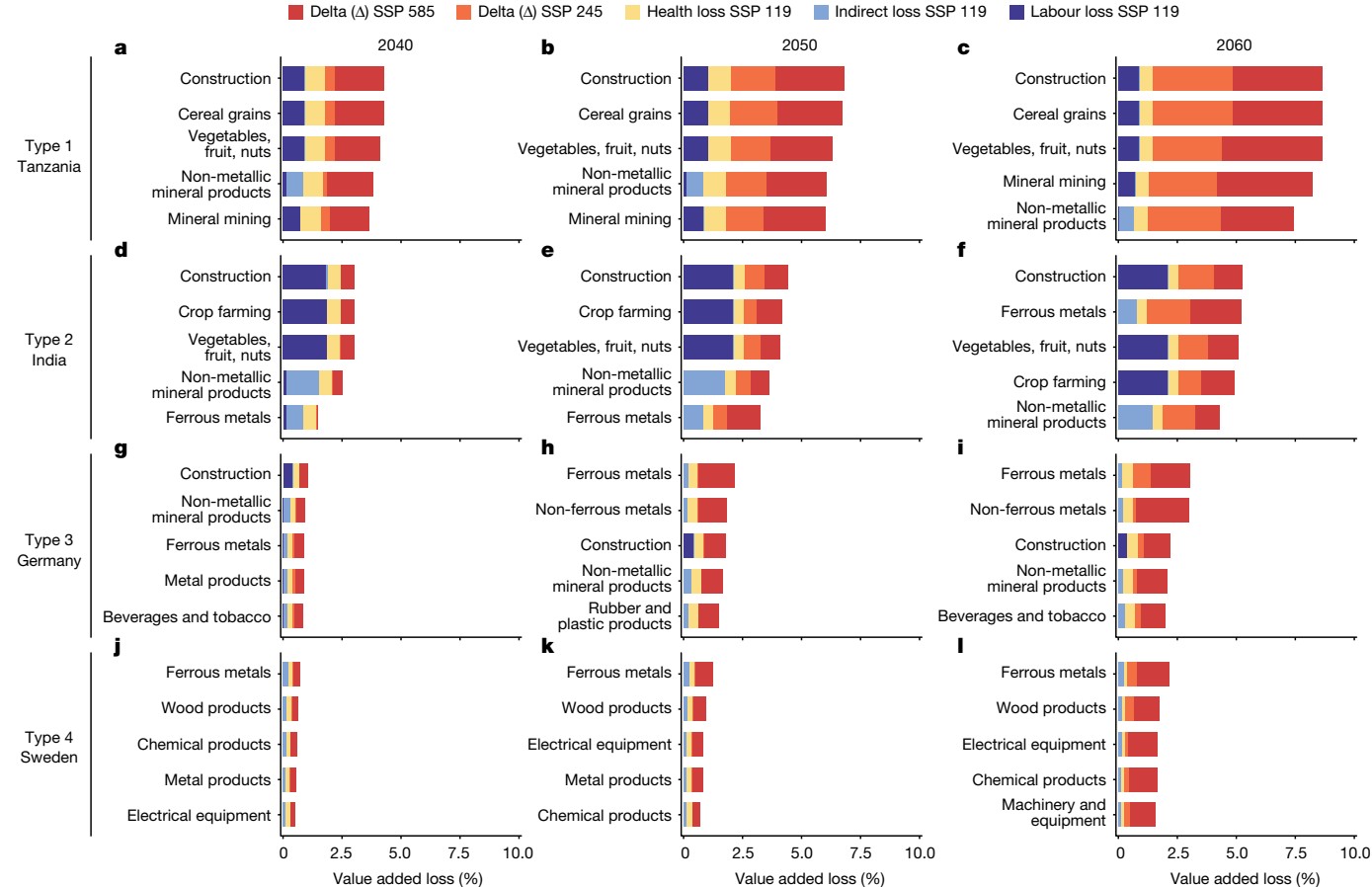

**Fig. 3 | Top five industrial sectors with the most severe heat-related losses in four representative countries. a–l**, Showing the top five most vulnerable sectors in Tanzania (**a–c**), India (**d–f**), Germany (**g–i**) and Sweden (**j–l**) in 2040 (**a,d,g,j**), 2050 (**b,e,h,k**) and 2060 (**c,f,i,l**). Sectors with absolute VA losses exceeding the median were ranked by percentage of VA losses, from highest to lowest. The length of the bar represents the 10-year average percentage VA loss of a sector. Sectors with the same percentage loss (for example, wheat, rice and cereals) were combined. Colours indicate the three categories of losses under SSP 119: health losses (yellow bars), labour productivity losses (purple bars) and indirect losses (azure bars). The orange and red bars represent the increment of the total loss under SSP 245 and SSP 585 (without differentiating by type of loss), respectively. The four types of countries were derived by machine-learning clustering based on sectoral patterns of economic loss (Supplementary Figs. 11 and 12 and Supplementary Table 3).

productivity in the domestic construction and plantation industries leads directly to high economic losses in the country's related value chains. As shown in Extended Data Fig. 10, countries located at low and middle latitudes, such as China and Vietnam, exhibit similar patterns of loss.

Light manufacturing, including metal products, rubber and plastic products, food processing and beverages and tobacco, are vulnerable to indirect effects because of a lack of raw materials supply, such as minerals, metals, crops, oil seeds and vegetables. For example, under the SSP 119 scenario, metal products and tobacco and beverage manufacturing in Germany lose around 0.3% of VA in 2040. Under the SSP 585 scenario, the economic loss of beverages and tobacco would increase by more than six times in 2060, reaching 2.0% of VA as imports of plantation products (palm oil, soybeans, coffee, spices and so on) from South America, Southeast Asia and Africa decline by around 5% to 8% (Extended Data Fig. 8). Losses of metal products rise even faster, reaching 2.4% of VA in 2060. This is because the main producers of raw materials, such as coal and metals, which are essential for the metal products industry, are primarily located in regions that are vulnerable to climate change. This leads to higher losses in the metal product-related chain in most countries with developed manufacturing industries, including Germany, France and Australia (Extended Data Fig. 11). These countries have a relatively low share of agricultural GDP (less than 3%), with slight losses. Labour productivity losses are high

only in the construction or mining sector, whereas indirect losses are higher in the metal-related manufacturing sector because of insufficient supply from foreign trading partners.

Similarly, high-end machinery, equipment and chemical products industries suffer indirect losses as a result of multilevel cascading effects, even in very cool climates. Losses in these industries, especially in developed countries such as European countries, emerge slowly and are not substantial under the SSP 119 scenario but increase sharply under SSP 245 and SSP 585 scenarios. For example, Sweden's industry-wide production suffers mainly from indirect losses through supply-chain disruptions and excess mortality due to heatwaves. From 2040 to 2060 under the SSP 119 scenario, impacts on production activities are moderate given the cool climate and dependence on the stable EU supply chain. Sectors like electrical equipment and chemical products experience less than 1% of VA loss, mostly health loss due to sudden extreme heatwaves. However, sector VA losses soar under the SSP 585 scenario. Losses in the mechanical equipment sector increase rapidly, growing by approximately five times compared to the SSP 119 scenario. Ferrous metals (2.2%), electrical equipment (1.9%) and machinery and equipment (1.6%) experience the highest VA losses. Indirect losses become a main constraint in many sectors because national adaptive strategies or close regional trade flows (as in the EU) can no longer support production when heat stress becomes increasingly more severe globally. As shown in Extended Data Fig. 12, developed economies

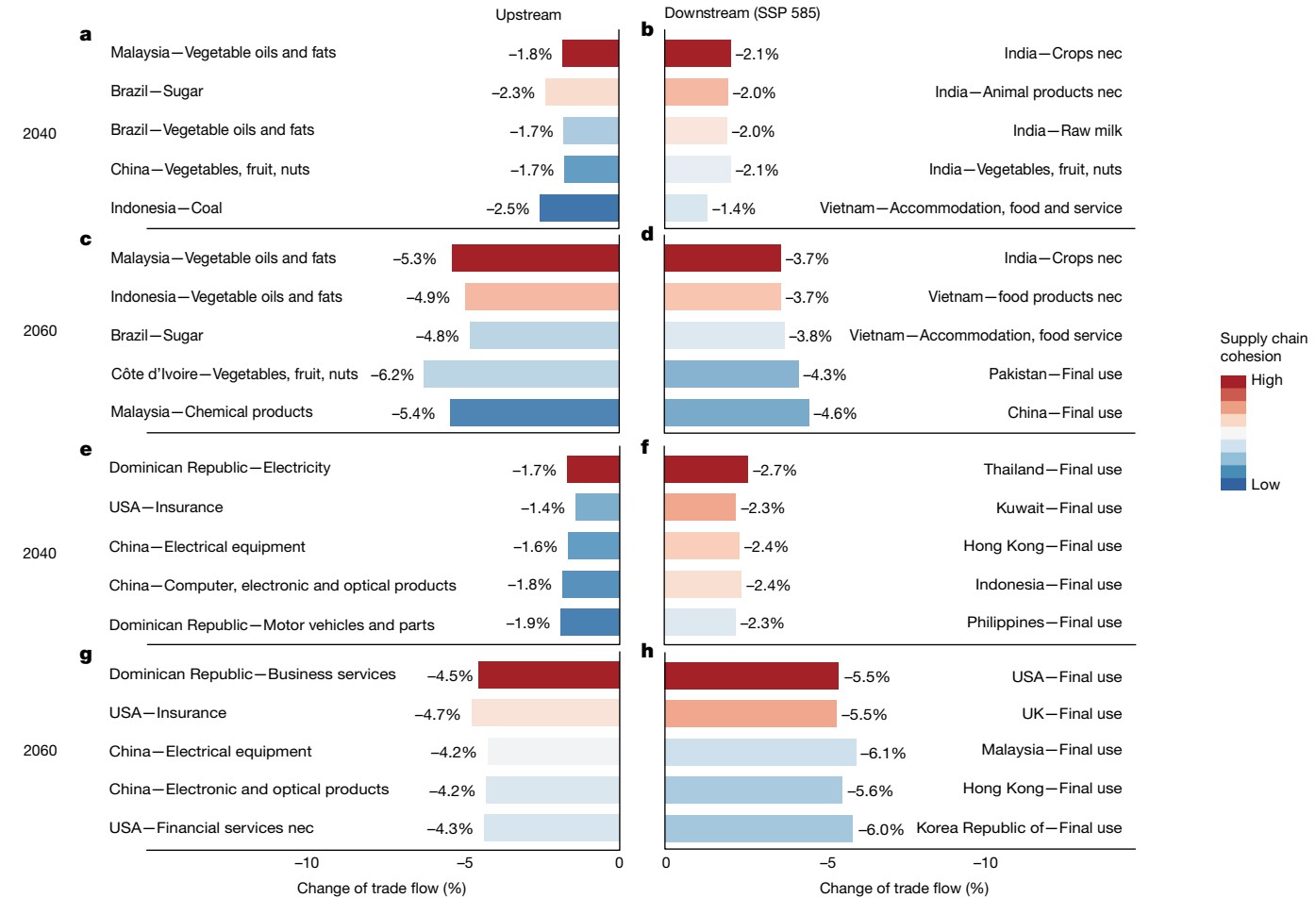

**Fig. 4 | Impacts of heat stress on supply chains of India food production and Dominican Republic tourism. a–d**, Trade flows between India food production sector and upstream (**a,c**) and downstream (**b,d**) sectors in 2040 (**a,b**) and 2060 (**c,d**). **e–h**, Trade flows between Dominican Republic tourism sector and upstream (**e,g**) and downstream (**f,h**) sectors in 2040 (**e,f**) and 2060 (**g,h**). Each bar represents a key trading partner (sector with trade volume above the 50% quartile of trade volumes of the selected sector with all partner sectors) and the length represents the percentage decrease in product flow compared to the base period of 2014. The colours of the bars represent the cohesion level of the particular sector to the Indian food production sector from blue (weak) to red (strong), which is measured by the trade volume between the particular sector and the Indian food production sector. nec, not elsewhere classified.

located at high latitudes, such as Norway and the UK, are characterized by similar loss patterns.

We also analyse the mechanism through which indirect losses from disruptions in international trade flows propagate through national supply chains of specific sectors. Figure 4 illustrates how climate risk propagates through two supply chains, the Indian food production and the Dominican Republic tourism sectors, respectively (see Extended Data Fig. 8 for other typical supply chains). Each of these sectors is important to the respective economies of India (13% of GDP) and the Dominican Republic (18% of GDP) and each is largely dependent on international supply chains. In the case of India's food sector, we see a pattern of 'upstream constraint' through which insufficient upstream supply of intermediates (such as palm oil from Indonesia) impacts the downstream sector and the entire value chain, whereas in the case of tourism in the Dominican Republic, we see a pattern of 'downstream constraint'—the impact of insufficient downstream demand affects the upstream sector and the entire value chain.

The supply chain of the Indian food production industry relies heavily on its upstream suppliers, the oil and fat sectors of Indonesia and Malaysia, and as a result it is vulnerable to higher temperatures. The unmitigated warming under the SSP 585 scenario exacerbates the shortage of raw materials. By 2060, palm oil supplies from Malaysia and Indonesia fall by 5.3% and 4.9%. Additionally, Brazilian sugar,

Southeast Asian and African vegetables, fruits and nuts are also less available, with a supply decreased by around 4–6%. Consequently, downstream countries, including India, Vietnam, Pakistan and other important trading partners, experience a contraction of imports between 3.7% and 5.1% (Extended Data Fig. 6). These impacts can negatively affect food prices and security in both developing and developed countries.

In contrast to the Indian food production industry, the Dominican Republic's tourism industry is more constrained by downstream demands. Under the SSP 585 scenario, the wealth generated by tourism in the Dominican Republic could drop substantially, as the largest source of foreign visitors, the USA, is likely to reduce annual demand for tourism in the Dominican Republic on average by around 5.5%. Demand from Malaysia and Indonesia is likely to fall by 6.1% and 8.1%. A drop in tourism output, the backbone of the Dominican Republic's economy, is likely to reduce demand for upstream business services and manufacturing industries by approximately 4.5–4.7%, causing an extra impact on the Dominican national economy. The decline in the tourism sector is also likely to lead to a 4.7% and 4.3% drop in the Dominican Republic's demand for insurance and financial services from the USA, as well as 4.2% and 4.5% drop in the demand for electronic equipment and chemical products from China (Extended Data Fig. 7). Furthermore, a smaller tourism sector leads to the slowdown in the construction of

tourism infrastructure and the supply of tourism supporting products, posing considerable risks for tourism investment.

## Implication for targeted risk governance and regional cooperation

By coupling climate, epidemiological and economic models, this study investigates the direct impact of heat stress on human activities and the indirect losses across the broader global supply chain. Focusing on the indirect effects of heat stress addresses a substantial gap in the literature. Comprehending the indirect effects of heat stress is crucial for devising effective and targeted adaptation strategies in the context of increasingly complex global supply-chain networks.

Our findings show that supply chains amplify the risk of future heat stress by causing nonlinear economic losses worldwide. In other words, the considerable adverse indirect effects of heat stress across interconnected markets cannot be overlooked. The indirect losses of heat stress highlight the need for countries to strengthen collaboration across global relevant supply-chain stakeholders to achieve successful heat stress adaptation. For instance, our results demonstrate that the impact of a heatwave on the agriculture and food manufacturing industry in India can further lead to a 0.9–2.3% loss of VA in the US food manufacturing industry. If the USA were to support India's adaptation efforts through technology transfer, they would indirectly be reducing their own losses. These considerations could guide policy-makers working towards global cooperation for future climate change mitigation and heat stress adaptation efforts.

We also illustrate the sensitivity of different countries and sectors to the three types of losses caused by heat stress. For example, Caribbean and Central African countries are more likely to suffer health losses, whereas for low-income countries in Africa and Southeast Asia labour losses are more likely. By contrast, small to medium-sized economies dependent on international trade, such as Brunei, are more exposed to indirect losses. The way heat stress-related costs emerge demonstrates how extensive and diverse impacts from heat stress are propagated through global supply chains, resulting in economic losses to a country or sector that may not be immediately apparent. Our quantitative results provide valuable information for designing more targeted and effective heat stress adaptation strategies.

Our developed model and estimations are subject to uncertainties and limitation (detailed description in Supplementary Information sections 1.1–1.3). For example, although the disaster footprint module is widely used and performs well for single-country/single-region analyses, the substitutability of products in a multicountry scenario requires further discussion to ensure robustness. To quantify some of the uncertainties, we conducted a comprehensive sensitivity analysis, with details available in the Methods and Supplementary Information section 1. Specifically, we used different years and versions of the input–output database for comparison to analyse the uncertainty in production and trade structures (Extended Data Fig. 4).

Globally, the estimate of the total amount of indirect losses is robust to changes in the data used (GTAP 2011 and GTAP 2014) for the base period. The results of the loss assessment at global scale differ by less than 5% in 2060. Most countries are distributed around the $y = x$ line, which suggests a consistent assessment across different trade structures. Regionally, for a few countries, indirect loss assessments can show larger differences. By comparison, we find that when using GTAP 2014 data for the base period, indirect economic losses in East and Southeast Asian countries, such as Singapore, Korea and Japan, are amplified (Extended Data Fig. 4 and Supplementary Fig. 6). This can be explained by the fact that, in GTAP 2014, those countries have closer economic ties with climate-sensitive markets, including Malaysia, China, India and Vietnam. For instance, trade between Singapore and emerging economies such as China and Vietnam had increased substantially from 2010 to 2014. According to the Singapore Department

of Statistics (https://www.singstat.gov.sg/) and the United Nations Commodity Trade Statistics Database (https://comtrade.un.org), China became the largest trading partner of Singapore in 2014, up from fourth place in 2011, whereas Vietnam rose to the 13th largest partner in 2014, from the 20th place in 2011. Conversely, Singapore's total trade share with the EU and the USA decreased slightly over the same period. Similarly, Japan, Korea and Myanmar developed closer trade relationships with emerging markets such as China, India and Vietnam.

The assessment of indirect losses under different trade relationships offers important insights into the likely supply-chain risks posed by climate change. As Africa, South America and Southeast Asia become increasingly involved in GVCs, the resilience of GVCs to the impacts of climate change must be properly assessed, rather than merely considering scale effects and comparative advantage in terms of economic efficiency.

For parameters such as the maximum stock ratio and excess production capacity, we conducted the experiment several times in the range of possible values from previous studies. For trade substitutability, upper and lower bounds of perfect substitution and non-substitution (traditional static input–output model) were used. We elaborate in more detail about the uncertainty intervals of the parameter for the three main modules and perform a Monte Carlo analysis, including simulation of economic loss dynamics for 10,000 periods (Supplementary Table 2). We have also conducted an historical validation using several authentic data sources (robustness tests and validation in the Supplementary Information), encompassing government statistics, empirical studies and institution reports[3,43–47] (Supplementary Tables 1, 6 and 7, and Supplementary Fig. 1), in addition to a comparative analysis of previous studies[33,39,48] concerning future periods based on CMIP5 data and similar RCP scenarios (Extended Data Fig. 5).

Despite the uncertainties, our conclusion that projected climate change will continue to increase heat-related risks globally in the coming decades and that global supply chains will amplify economic losses by spreading indirect losses to wider regions, remain robust. Therefore, in the future, the organization of global supply chains should gradually shift from an exclusive focus on efficiency to one that places equal emphasis on efficiency and resilience. A concerted global strategy to reduce emissions will not only directly protect many people in developing economies from direct economic losses of heat stress but will also maintain resilient and efficient global supply chains and contribute to the long-term, sound development of the global economy.

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

## Methods

Our methodology, in essence, combines three modules of climate, health and economy with full validation (Extended Data Fig. 1). The integrated model links climate module (estimating future climate parameters including surface air temperature and relative humidity and so on), demographic and health module (simulating future world population dynamics and exposure–response functions to warming) and economic module (dynamic footprint of heat-induced labour loss on global economy and supply chain).

### Climate module

Fourteen GCMs involved in the framework of CMIP6 (Extended Data Table 1) with ten bias-corrected models from ISIMIP3b[49,50] are used to estimate the modelled heat stress projection for the end of the twenty-first century. Five models were randomly averaged several times from the climate model ensemble as a Monte Carlo uncertainty analysis. ERA5 re-analysis data[51] from 1985 to 2022 are used for bias-correction and validation. Climatic parameters such as maximum and average temperature and relative humidity on a daily scale are integrated, which are closely related to future working environment (Supplementary Fig. 9).

Many institutes, including International Standards Organization (ISO) and US National Institute for Occupational Safety and Health (NIOSH), use WBGT to quantify different amounts of heat stress and define the percentage of a typical working hour that a person can work while maintaining core body temperature. To facilitate the long-term calculation, we use[18] simplified WBGT, which approximates WBGT well using temperature ($T_a$) and relative humidity (RH)[52,53] as parameters such as solar radiation and wind speed have higher uncertainty and weaker effects at the global scale. To take into account indoor heat exposures for industrial and service sector workers, we used the approximation that indoor $WBGT_{indoor} = WBGT_{outdoor} − 4$, based on a deduction of the radiation exposure factor from the formula below[18]:

$$WBGT_{outdoor} = 0.567 \times T_a + 3.94 + 0.393 \times E \quad (1)$$

$$E = \frac{RH}{100} \times 6.105 \times \exp\left(17.27 \times \frac{T_a}{(237.7 + T_a)}\right) \quad (2)$$

We also calculated the spatial and temporal evolutionary trends in the occurrence of future heatwaves to calculate excess mortality. There is no consistent definition for heatwave worldwide because people may have acclimatized to their local climatic zones and different studies have applied various temperature metrics[54,55]. Heatwaves are usually defined by absolute or relative temperature threshold in consecutive days[56]. There are various ways to define a heatwave. For example, the IPCC defines heatwave as "a period of abnormally hot weather, often defined with reference to a relative temperature threshold, lasting from two days to months", whereas the Chinese Meteorological Administration defined heatwave as "at least three consecutive days with maximum temperature exceeding 35 °C". Others[31] identified heatwave using the TX90p criterion, that is, when the 90th percentile of the distribution of regional maximum temperatures spanned by data from the period 1981–2010 was exceeded for at least three consecutive days. In our study, two or more consecutive days above the 95% threshold of the 1985–2015 ERA5 daily mean temperature[51,57] were defined as a heatwave, which is considered to be a moderate estimation and is widely used in epidemiological studies[36,58,59]. Several definitions, such as four or more consecutive days above the 97.5% threshold, are used as sensitivity analysis. Considering certain amounts of climate adaptation of the local resident along the warming climate, dynamic heatwave thresholds[60] are defined as part of the uncertainty analysis in this study; that is two or more consecutive days above the 95% threshold of the daily mean temperature between 1985 and the year before the target year were defined as a heatwave (ERA5 data are used for 1985–2014; climate

projection data are used after 2015). The use of a dynamic threshold based on both historical and climate projections data helps to incorporate the human adaptation of heat stress in a long-term warming scenario, as reported in recent studies[61–64].

### Health costs related to heat exposure

Some studies have shown that the health impact of heatwaves could vary substantially with location[65,66]. Few studies have investigated the heatwave-induced mortality risk at a global scale[41,67]. A primitive health risk function associating heatwave mortality risks with four different climate zones was established by ref. 36 on the basis of a comprehensive study using data from 400 communities in 18 countries/regions across several years (1972–2012). Here, we used the relative risk coefficients (Extended Data Table 2) from figure 4 of ref. 36 for four different climate zones (Extended Data Fig. 3) to estimate potential heatwave-related death due to climate change on a global scale. The simplified four-climate-zone-based estimation may neglect subregional characters and should be interpreted with caution, as further factors affecting heat-induced death (such as air condition accessibility[68], age[69–72] and humidity[73]) are not included in this study.

The number of excess deaths $D_{hw}$ during a heatwave period was calculated at each grid cell level (0.5°) with the following equation:

$$D_{hw} = POP \times MR \times (RR − 1) \times HWN \quad (3)$$

POP is the population at the given location consistent with the SSPs[74]. MR is the average daily mortality rate (2009–2019) at the country level obtained from the World Bank[75]. For 37 countries with large territory and more refined data (for example, European Union (including UK), Russia, Ukraine, China, the USA, Canada, Brazil, South Africa, India and Australia), we used state/provincial statistics based on data from national statistical offices (Source, World Bank; state/province level data for European Union, Eurostat[76]; Russia, The Russian Fertility and Mortality database[77]; China, China Statistical Yearbook 2019[78]; the USA, National Institutes of Health[79]; Brazil, *Fundação Amazônia de Amparo a Estudos e Pesquisas*[80]; Canada, Statistics Canada[81]; Australia, Australian Bureau of Statistics[82]; India, Ministry of Finance Economic Survey[83]). RR is the relative risk of mortality caused by heatwaves. HWN is the number of heatwave days for the given year and location (Extended Data Fig. 2).

The calculated excess deaths are translated to a social-economic loss on the basis of the value of statistical life (VSL). The concept of VSL is widely used throughout the world to monetize fatality risks in benefit–cost analyses. The VSL represents the individual's local money–mortality risk tradeoff value, which is the value of small changes in risk, not the value attached to identified lives. The country-based VSL estimation used in this research is adopted from the global health risks pricing study by ref. 84. The estimation is based on the estimated VSL in the USA (US\$_{2019}11 million) and coupled with an income elasticity of 1.0 to adjust the VSL to other countries using the fixed-effects specification. A similar health valuation method has been adopted in past studies[85,86] and was recommended in the report of the World Bank[87]. Moreover, a sensitivity test is conducted under the assumption that all life would be valued equally across the world (Supplementary Figs. 2 and 3). For such a test, an averaged VSL is calculated by summing up each country's income-based VSL times its population then dividing by the total population of the world.

### Expose function of labour productivity

The increase in daily temperatures affects the efficiency of workers and reduces safe working time. A compromise in endurance capacity due to thermoregulatory stress was already evident at 21 °C. Different studies used similar methods to evaluate the labour loss function. The form of logistic function with 'S' shape has become the consensus of the academic community but the specific functional equation and

parameters are various in different studies. The loss functions used in mainstream research include exponential function[88] as equation (4), cumulative normal distribution function[5,41] as equation (5) and so on. In this research, we adopt the cumulative normal distribution function (equation (5)) as our benchmark function because it was extensively applied and case proven in 3-year reports of the Lancet Countdown on health and climate change[5,41,89,90]. Because the Hothaps function (equation (4)) is subject to parameter uncertainty as a result of being based on a few empirical studies, we use it to test for the sensitivity of our estimates (Supplementary Figs. 4 and 5). Our methodology identifies three ISO standard work intensity amounts: 200 W (assumed to be office workers in the service industry, engaged in light work indoors), 300 W (assumed to be industrial workers, engaged in moderate work indoors) and 400 W (assumed to be construction or agricultural workers, engaged in heavy work outside). For example, to calculate workability loss fraction in India's food production sector (300 W, indoor), we bring the corresponding parameters (Extended Data Table 3) and WBGT$_{indoor}$ into equation (5). Previous studies have tended to ignore indoor workforce loss, assuming that the indoor workforce was very low under current climate condition or protected by air conditioning[91]. However, a growing number of studies have proved that future indoor labour losses cannot be underestimated[31]. For example, only 7% of households in India possess an air conditioner, despite having extremely high cooling needs. Considering the severe adaptation cooling deficit in emerging economies[92], indoor labour losses must be fully considered in global-scale studies. This study uses the climate–income–air conditioner usage function published by ref. 93 to assess the rate of air conditioning protection in conjunction with the per capita income of each country under each SSP scenario. Higher per capita income in each country leads to higher air-conditioning penetration, whereas the climate base determines the rate and trend of increase in air-conditioning penetration (elasticity of penetration to income). In our study, we improved the function by replacing cooling degree days (CDDs) with indoor WBGT, as CDDs only consider temperature neglecting humidity. Only the indoor workforce under air conditioning, will be protected from heat-induced loss.

$$\text{Workability}_{\text{Hothaps}} = 0.1 + \frac{0.9}{\left(1 + \left(\frac{\text{WBGT}}{\alpha_1}\right)^{\alpha_2}\right)} \tag{4}$$

$$\text{Loss fraction} = \frac{1}{2}\left(1 + \text{ERF}\left(\frac{\text{WBGT} - \text{Prod}_{\text{mean}}}{\text{Prod}_{\text{SD}} \times \sqrt{2}}\right)\right) \tag{5}$$

Of which the parameters for a given activity level (Prod$_{\text{mean}}$ and Prod$_{\text{SD}}$, defined as the amount of internal heat generated in performing the activity) are given in Extended Data Table 3, and ERF is the error function defined as:

$$\text{ERF}(z) = \frac{2}{\sqrt{\pi}} \int_0^z e^{-t^2} dt \tag{6}$$

To calculate average daily impacts, we use an approximation for hourly data based on the 4 + 4 + 4 method implemented by ref. 14. We assume that 4 h per day is close to WBGT$_{\text{max}}$ and 4 h per day is close to WBGT$_{\text{mean}}$ (early morning and early evening). The remaining 4 h of a 12 h daylight day is assumed to be halfway between WBGT$_{\text{mean}}$ and WBGT$_{\text{max}}$ (labelled WBGT$_{\text{half}}$). The analysis above gives the summer daily potential workability lost in each grid cell at each amount of work intensity and environment (200–400 W, indoor or outdoor). By combining this with the dynamic population grid under each SSP scenario (see Supplementary Fig. 13 for comparison with static population setting), we aggregate to obtain country-scale labour productivity losses. In the disaster footprint model, we adopt the approach presented by ref. 5

which defines the timeframe for computing labour productivity losses as the warm season (June to 30 September in the Northern Hemisphere and December to 30 March in the Southern Hemisphere) to adjust the overestimation of the risk of moderate hot temperature, as the model is more applicable to sudden and strong shocks rather than moderate changes throughout the year.

## Global disaster footprint analysis module

The global economic loss will be calculated using the following hybrid input–output and computable general equilibrium (CGE) global trade module. Our global trade module is an extension of the adaptive regional input–output (ARIO) model[20,94,95], which was widely used in the literature to simulate the propagation of negative shocks throughout the economy[96–99]. Our model improves the ARIO model in two ways. The first improvement is related to the substitutability of products from the same sector sourced from different regions. Second, in our model, clients will choose their suppliers across regions on the basis of their capacity. These two improvements contribute to a more realistic representation of bottlenecks along global supply chains[100].

Our global trade module mainly includes four modules: production module, allocation module, demand module and simulation module. The production module is mainly designed for characterizing the firm's production activities. The allocation module is mainly used to describe how firms allocate output to their clients, including downstream firms (intermediate demand) and households (final demand). The demand module is mainly used to describe how clients place orders to their suppliers. And the simulation module is mainly designed for executing the whole simulation procedure.

### Production module

The production module is used to characterize production processes. Firms rent capital and use labour to process natural resources and intermediate inputs produced by other firms into a specific product. The production process for firm $i$ can be expressed as follows,

$$x_i = f(\text{for all } p, z_i^{\text{p}}; \text{va}_i)$$

where $x_i$ denotes the output of the firm $i$, in monetary value; $p$ denotes type of intermediate products; $z_i^{\text{p}}$ denotes intermediate products used in production processes; va$_i$ denotes the primary inputs to production, such as labour ($L$), capital ($K$) and natural resources (NR). The production function for firms is $f(\cdot)$. There is a wide range of functional forms, such as Leontief[101], Cobb–Douglas and constant elasticity of substitution production function[102]. Different functional forms reflect the possibility for firms to substitute an input for another. Considering that heat stress tends to be concentrated in a specific short period of time, during which economic agents cannot easily replace inputs as suitable substitutes, might temporarily be unavailable, we use Leontief production function which does not allow substitution between inputs.

$$x_i = \min\left(\text{for all } p, \frac{z_i^{\text{p}}}{a_i^{\text{p}}}; \frac{\text{va}_i}{b_i}\right)$$

where $a_i^{\text{p}}$ and $b_i$ are the input coefficients calculated as

$$a_i^{\text{p}} = \frac{\overline{z}_i^{\text{p}}}{\overline{x}_i}$$

and

$$b_i = \frac{\overline{\text{va}}_i}{\overline{x}_i}$$

where the horizontal bar indicates the value of that variable in the equilibrium state. In an equilibrium state, producers use intermediate

products and primary inputs to produce goods and services to satisfy demand from their clients. After a disaster, output will decline. From a production perspective, there are mainly the following constraints.

**Labour supply constraints.** Labour constraints during heat stress or after a disaster may impose severe knock-on effects on the rest of the economy[21,103]. This makes labour constraints a key factor to consider in disaster impact analysis. For example, in the case of heat stress, these constraints can arise from employees' inability to work as a result of illness or extreme environmental temperatures beyond health threshold. In this model, the proportion of surviving productive capacity from the constrained labour productive capacity ($x_i^L$) after a shock is defined as:

$$x_i^L(t) = (1 - \gamma_i^L(t)) \times \overline{x}_i$$

Where $\gamma_i^L(t)$ is the proportion of labour that is unavailable at each time step $t$ during heat stress; $(1 - \gamma_i^L(t))$ contains the available proportion of employment at time $t$.

$$\gamma_i^L(t) = (\overline{L}_i - L_i(t))/\overline{L}_i$$

The proportion of the available productive capacity of labour is thus a function of the losses from the sectoral labour forces and its predisaster employment level. Following the assumption of the fixed proportion of production functions, the productive capacity of labour in each region after a disaster ($x_i^L$) will represent a linear proportion of the available labour capacity at each time step. Take heatwaves as an example; during extreme heatwaves that last for days on end, governments and businesses often shut down work to reduce the risk of serious illnesses such as pyrexia. This imposes an exogenous negative shock on the economic network.

**Constraints on productive capital.** Similar to labour constraints, the productive capacity of industrial capital in each region during the aftermath of a disaster ($x_i^K$) will be constrained by the surviving capacity of the industrial capital[30,96,104–106]. The share of damage to each sector is directly considered as the proportion of the monetized damage to capital assets in relation to the total value of industrial capital for each sector, which is disclosed in the event account vector for each region ($\gamma_i^K$), following ref. 107. This assumption is embodied in the essence of the input–output model, which is hard-coded through the Leontief-type production function and its restricted substitution. As capital and labour are considered perfectly complementary as well as the main production factors and the full employment of those factors in the economy is also assumed, we assume that damage in capital assets is directly related with production level and, therefore, VA level. Then, the remaining productive capacity of the industrial capital at each time step is defined as:

$$x_i^K(t) = (1 - \gamma_i^K(t)) \times \overline{x}_i$$

Where, $\overline{K}_i$ is the capital stock of firm $i$ in the predisaster situation and $K_i(t)$ is the surviving capital stock of firm $i$ at time $t$ during the recovery process

$$\gamma_i^K(t) = (\overline{K}_i - K_i(t))/\overline{K}_i$$

**Supply constraints.** Firms will purchase intermediate products from their supplier in each period. Insufficient inventory of a firm's intermediate products will create a bottleneck for production activities. The potential production level that the inventory of the $p$th intermediate product can support is

$$x_i^p(t) = \frac{S_i^p(t-1)}{a_i^p}$$

where $S_i^p(t-1)$ refers to the amount of $p$th intermediate products held by firm $i$ at the end of time step $t-1$.

Considering all the limitation mentioned above, the maximum supply capacity of firm $i$ can be expressed as

$$x_i^{max}(t) = \min(x_i^L(t); x_i^K(t); \text{ for all } p, x_i^p(t))$$

The actual production of firm $i$, $x_i^a(t)$, depends on both its maximum supply capacity and the total orders the firm received from its clients, $TD_i(t-1)$ (see section on the 'Demand module'),

$$x_i^a(t) = \min(x_i^{max}(t), TD_i(t-1))$$

The inventory held by firm $i$ will be consumed during the production process,

$$S_i^{p,used}(t) = a_i^p \times x_i^a(t)$$

### Allocation module

The allocation module mainly describes how suppliers allocate products to their clients. When some firms in the economic system suffer a negative shock, their production will be constrained by a shortage to primary inputs such as a shortage of labour supply during extreme heat stress. In this case, a firm's output will not be able to fill all orders of its clients. A rationing scheme that reflects a mechanism on the basis of which a firm allocates an insufficient amount of products to its clients is needed[108]. For this case study, we applied a proportional rationing scheme according to which a firm allocates its output in proportion to its orders. Under the proportional rationing scheme, the amounts of products of firm $i$ allocated to firm $j$, $FRC_j^i$ and household $h$, $HRC_h^i$ are as follows,

$$FRC_j^i(t) = \frac{FOD_i^j(t-1)}{\left(\sum_j FOD_i^j(t-1) + \sum_h HOD_i^h(t-1)\right)} \times x_i^a(t)$$

$$HRC_h^i(t) = \frac{HOD_i^h(t-1)}{\left(\sum_j FOD_i^j(t-1) + \sum_h HOD_i^h(t-1)\right)} \times x_i^a(t)$$

where $FOD_i^j(t-1)$ refers to the order issued by firm $j$ to its supplier $i$ in time step $t-1$, and $HOD_i^h(t-1)$ refers to the order issued by household $h$ to its supplier $j$. Firm $j$ received intermediates to restore its inventories,

$$S_j^{p,restored}(t) = \sum_{i \to p} FRC_j^i(t)$$

Therefore, the amount of intermediate $p$ held by firm $i$ at the end of period $t$ is

$$S_j^p(t) = S_j^p(t-1) - S_j^{p,used}(t) + S_j^{p,restored}$$

### Demand module

The demand module represents a characterization of how firms and households issues orders to their suppliers at the end of each period. A firm orders its supplier because of the need to restore its intermediate product inventory. We assume that each firm has a specific target inventory level based on its maximum supply capacity in each time step,

$$S_i^{p,*}(t) = n_i^p \times a_i^p \times x_i^{max}(t)$$

Then the order issued by firm $i$ to its supplier $j$ is

$$FOD_j^i(t) = \begin{cases} (S_i^{p,*}(t) - S_i^p(t)) \times \dfrac{\overline{FOD}_j^i \times x_j^a(t)}{\sum_{j \to p}(\overline{FOD}_j^i \times x_j^a(t))}, & \text{if } S_i^{p,*}(t) > S_i^p(t); \\ 0, & \text{if } S_i^{p,*}(t) \le S_i^p(t). \end{cases}$$

Households issue orders to their suppliers on the basis of their demand and the supply capacity of their suppliers. In this study, the demand of household $h$ to final products $q$, $HD_h^q(t)$, is given exogenously at each time step. Then, the order issued by household (HOD) $h$ to its supplier $j$ is

$$HOD_j^h(t) = HD_h^q(t) \times \frac{\overline{HOD}_j^h \times x_j^a(t)}{\sum_{j \to q}(\overline{HOD}_j^h \times x_j^a(t))}$$

The total order received (TOD) by firm $j$ is

$$TOD_j(t) = \sum_i FOD_j^i(t) + \sum_h HOD_j^h(t)$$

## Simulation module
At each time step, the actions of firms and households are as follows in Monte Carlo simulations.

Firms plan and execute their production on the basis of three factors: (1) inventories of intermediate products they have, (2) supply of primary inputs and (3) orders from their clients. Firms will maximize their output under these constraints.

**Product allocation.** Firms allocate outputs to clients on the basis of their orders. In equilibrium, the output of firms just meets all orders. When production is constrained by exogenous negative shocks, outputs may not cover all orders. In this case, we use a proportional rationing scheme proposed in the literature[20,108] (see section on 'Allocation module') to allocate products of firms.

Firms and households issue orders to their suppliers for the next time step. Firms place orders with their suppliers on the basis of the gaps in their inventories (target inventory level minus existing inventory level). Households place orders with their suppliers on the basis of their demand. When a product comes from several suppliers, the allocation of orders is adjusted according to the production capacity of each supplier.

This discrete-time dynamic procedure can reproduce the equilibrium of the economic system and can simulate the propagation of exogenous shocks, both from firm and household side or transportation disruptions, in the economic network. From the firm side, if the supply of a firm's primary inputs is constrained, it will have two effects. On the one hand, the decline in output in this firm means that its clients' orders cannot be fulfilled. This will result in a decrease in inventory of these clients, which will constrain their production. This is the so-called forward or downstream effect. On the other hand, less output in this firm also means less use of intermediate products from its suppliers. This will reduce the number of orders it places on its suppliers, which will further reduce the production level of its suppliers. This is the so-called backward or upstream effect. From the household side, the fluctuation of household demand caused by exogenous shocks will also trigger the aforementioned backward effect. Take tourism as an example, when the temperature is well beyond the comfort range of the visitor, the demand for tourism from households all over the world will decline significantly. This influence will further propagate to the accommodation and catering industry through supplier–client links.

## Economic footprint
We define the VA decrease of all firms in a network caused by an exogenous negative shock as the disaster footprint of the shock. For the firm directly affected by exogenous negative shocks, its loss includes two parts: (1) the VA decrease caused by exogenous constraints and (2) the VA decrease caused by propagation. The former is the direct loss, whereas the latter is the indirect loss. A negative shock's total economic footprint ($TEF_{i,r}$), direct economic footprint ($DEF_{i,r}$) and propagated economic footprint ($PEF_{i,r}$) for firm $i$ in region $r$ are,

$$TEF_{i,r} = \overline{va}_{i,r} \times T - \sum_{t=1}^{T} va_{i,r}^a(t)$$

and,

$$DEF_{i,r} = \overline{va}_{i,r} \times T - \sum_{t=1}^{T} va_{i,r}^{max}(t)$$

and,

$$PEF_{i,r} = TEF_{i,r} - DEF_{i,r}$$

## Global supply-chain network
We build a global supply-chain network based on v.10 of the Global Trade Analysis Project (GTAP) database[109] and use GTAP 9 (ref. 110), EMERGING database[111] for robustness analysis. GTAP 10 provides a multiregional input–output (MRIO) table for the year 2014. Also, the database for the year 2011 was used for robustness testing. This MRIO table divides the world into 141 economies, each of which contains 65 production sectors (Supplementary Tables 4 and 5). If we treat each sector as a firm (producer) and assume that each region has a representative household, we can obtain the following information in the MRIO table: (1) suppliers and clients of each firm; (2) suppliers for each household and (3) the flow of each supplier–client connection under the equilibrium condition. This provides a benchmark for our model. We also used a dynamic CGE model consistent with the SSP scenarios for a parallel assessment and as part of the robustness check of the ARIO results. Specifically, the CGE model we used is a G-RDEM[112] with aggregated ten regions and ten sectors[113–115] (Supplementary Information section 1.3).

When applying such a realistic and aggregated network to the disaster footprint model, we need to consider the substitutability of intermediate products supplied by suppliers from the same sector in different regions[115–117]. The substitution between some intermediate products is straightforward. For example, for a firm that extracts spices from bananas it does not make much of a difference if the bananas are sourced from the Philippines or Thailand. However, for a car manufacturing firm in Japan, which uses screws from Chinese auto parts suppliers and engines from German auto parts suppliers to assemble cars, the products of the suppliers in these two regions are non-substitutable. If we assume that all goods are non-substitutable as in the traditional input–output model, then we will overestimate the loss of producers such as the case of the fragrance extraction firm. If we assume that products from suppliers in the same sector can be completely substitutable, then we will substantially underestimate the losses of producers such as the Japanese car manufacturing firm. To alleviate these shortcomings in the evaluation of losses under the two assumptions, we allow for the possibility of substitution for each sector depending on the region and sector of the supplier (Supplementary Information section 1.3).

Nonetheless, our estimates of economic damages from heat stress are subject to some important uncertainties[118] and our methods may not capture all types of economic damages. We only include economic losses caused by heat stress on human activities without considering the impacts on infrastructure, crop growth and other factors. Considering the challenges of predicting changes to socioeconomic systems globally, we have followed the approach from the literature[23,31,91,119] to simulate supply-chain indirect losses by considering the impact of future climate risks on current socioeconomic settings. We have not

considered the potential substitution of labour with capital resulting from technological advances, such as mechanization. Our analysis ignores the different levels of trade openness and globalization among SSP narratives, as well as the role of dynamic factors such as technology and price. Again, although we have conducted robustness tests for different degrees of trade substitutability, the relevant parameter is set randomly in the Monte Carlo simulation rather than derived through a general equilibrium model. The results should therefore be interpreted with caution as indicating potential future climate change risks to the existing economy rather than as quantitative predictions, given that the static representation of the economic structure in our model inevitably skews the assessment in the long run.

## Data availability

Data for the numerical results of this research are provided at https://zenodo.org/records/10032431. The global trade dataset used to simulate the presented results are licensed by the Global Trade Analysis Project at the Centre for Global Trade Analysis, Department of Agricultural Economics, Purdue University. The GTAP v.10 can be obtained for a fee from its official website: https://www.gtap.agecon.purdue.edu/databases/v10/index.aspx. Owing to the restriction in the licensing agreement with GTAP, the authors have no right to disclose the original dataset publicly. Multimodal meteorological data are derived from World Climate Research Programme (WCRP CMIP6): https://esgf-node.llnl.gov/search/cmip6/. Socioeconomic data for the different SSP scenarios are derived from IIASA: https://secure.iiasa.ac.at/web-apps/ene/SspDb/. Global population projection grids are from Socioeconomic Data and Applications Center (SEDAC) (https://sedac.ciesin.columbia.edu/data/set/popdynamics-1-8th-pop-base-year-projection-ssp-2000-2100-rev01/data-download).

## Code availability

The climate and epidemiological module processes daily surface temperature, dynamic population grid and baseline mortality data to determine heatwave days and the associated excess deaths. The economic module simulates changes of values and flows in global multiregional input–output table under shocks. All of the codes can be accessed at https://zenodo.org/records/10334260. The minimal input for the code is multiregional input–output table. The sample code and test data for the minimal inputs are also provided.

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

**Acknowledgements** This study was supported by the National Natural Science Foundation of China (grant nos. 72242105, 72091514 and 72250710169) and the startup funding from Zhejiang University to S.Z.

**Author contributions** D.G. designed and supervised the study. Y.S., S.Z. and D.W. conducted the study, collected the data, analysed the results and drafted the paper. J.D. and Y.W. collected and processed the meteorological data. H.Y., M.Z. and W.C. provided guidance on the calculation of health and labour productivity losses. C.T., Y.H. and L.Z. participated in the writing of the manuscript. S.T. and H.L. guided the uncertainty analysis and validation.

**Competing interests** The authors declare no competing interests.

**Additional information**
**Correspondence and requests for materials** should be addressed to Dabo Guan.

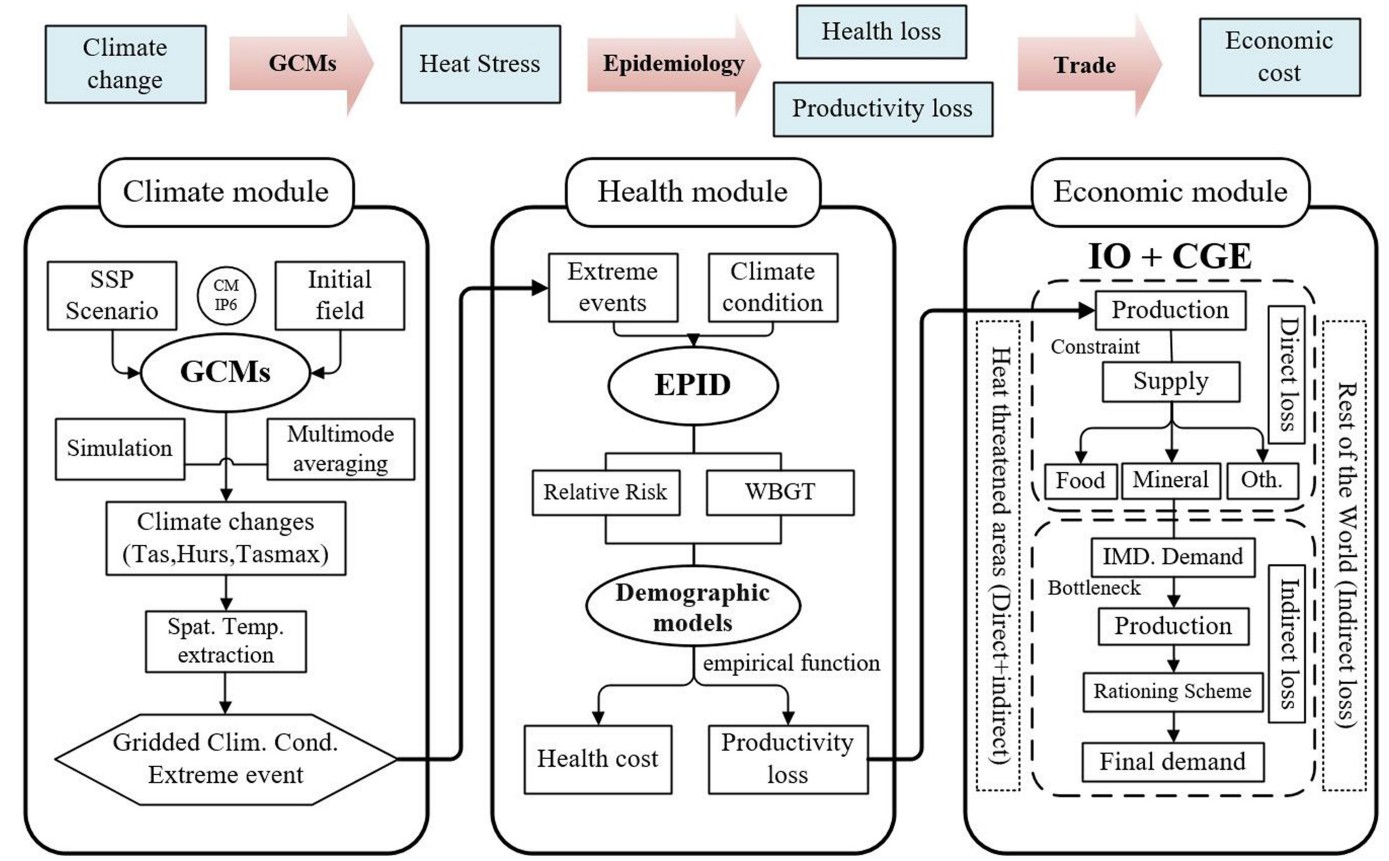

**Extended Data Fig. 1 | Schematic diagram of the methodological framework.** Coupling mechanisms for climate, health and economic modules.

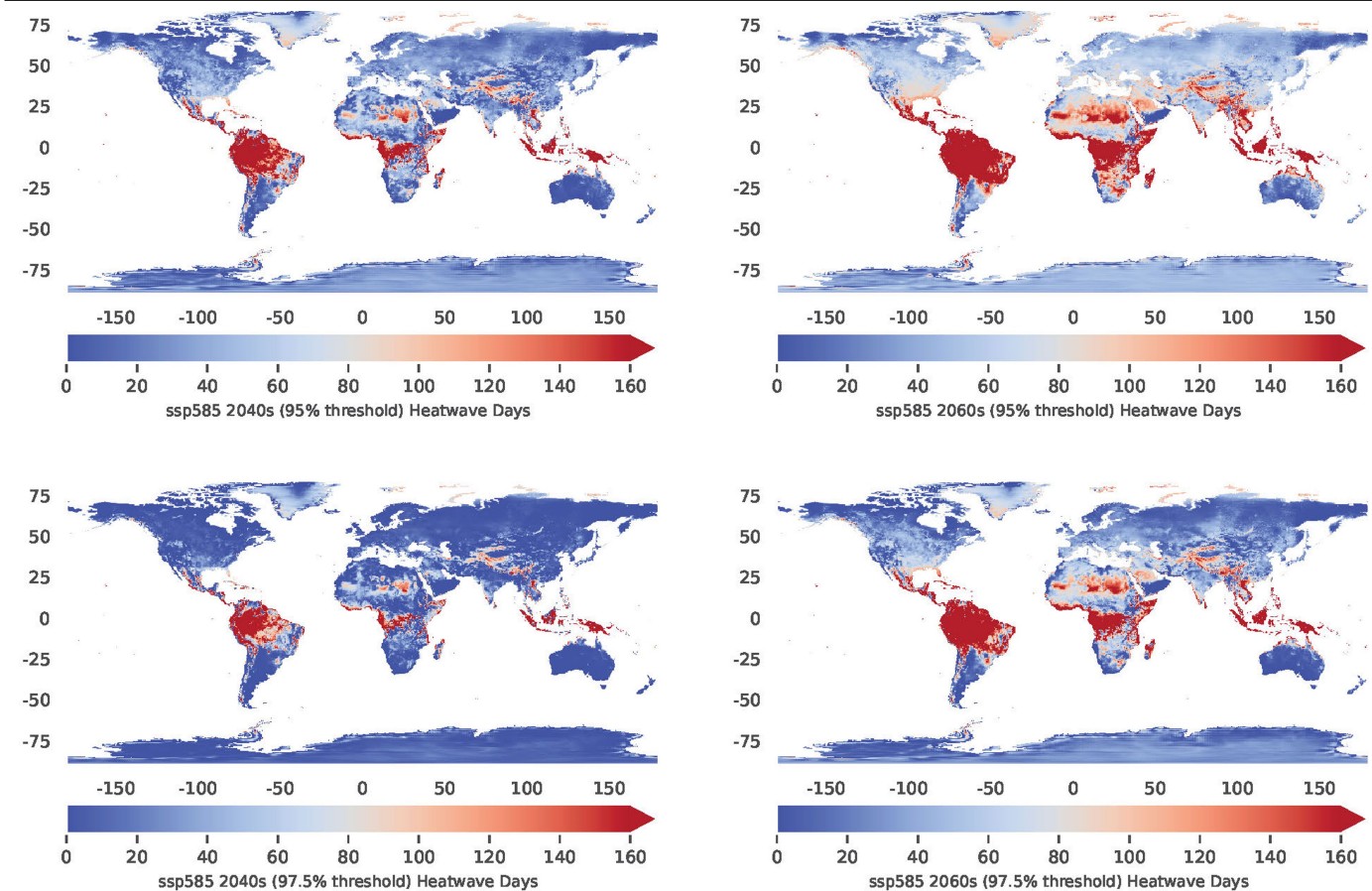

**Extended Data Fig. 2 | Heatwave days in the 2040s and 2060s under SSP585 Scenario.** The number of heatwave days in each cell was calculated from the ten-year average.

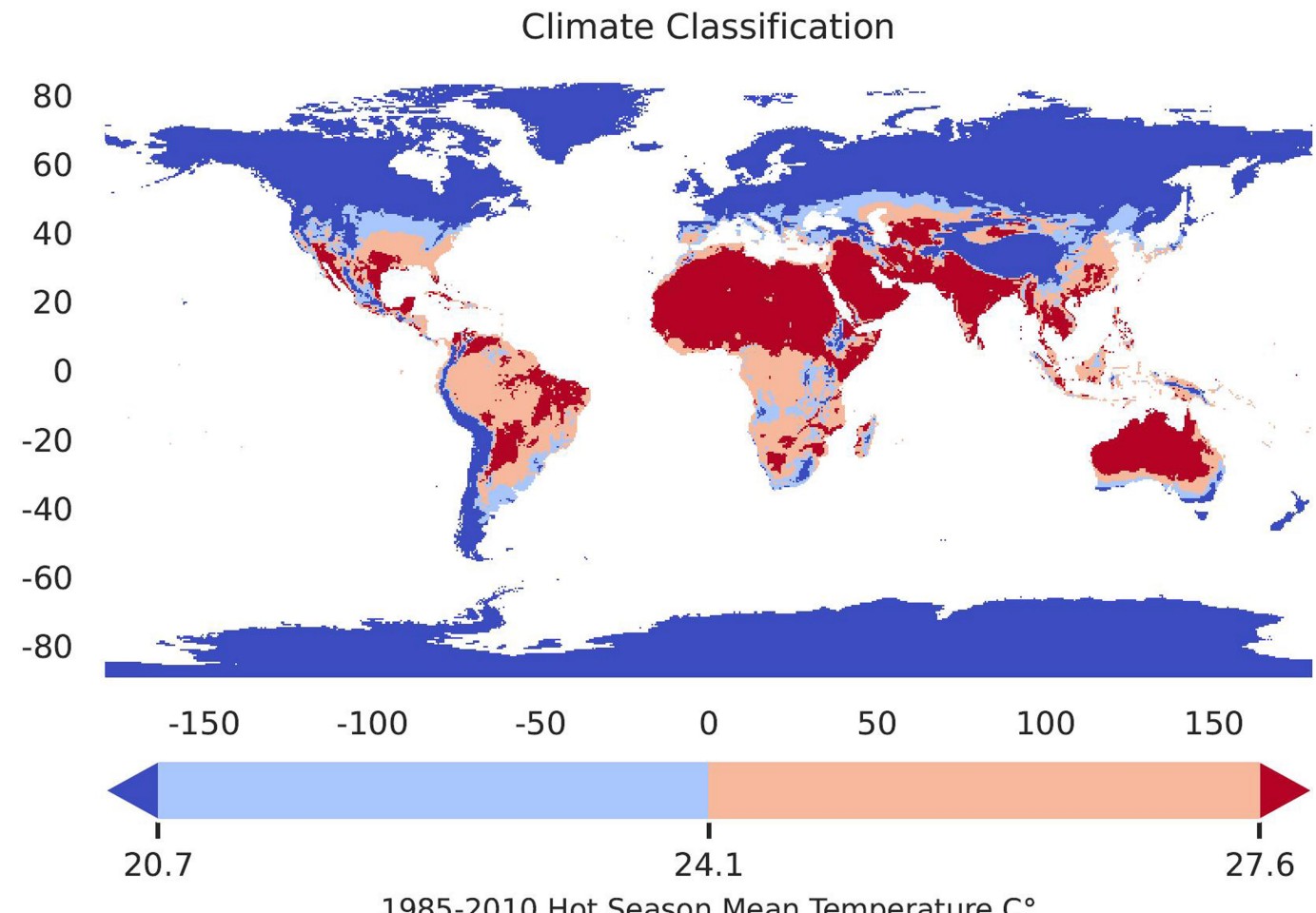

## Climate Classification

### 1985-2010 Hot Season Mean Temperature C°

20.7 24.1 27.6

**Extended Data Fig. 3 | Climate zones classification of relative risk.** Cold area: mean temperature of hot season: ≤<= 20.7 °C; moderate cold areas: mean temperature of hot season: 20.7–24.1 °C; moderate hot areas: mean temperature of hot season: 24.1–27.6 °C; and hot areas: mean temperature of hot season: >27.6 °C, based on ERA5 1985–2010.

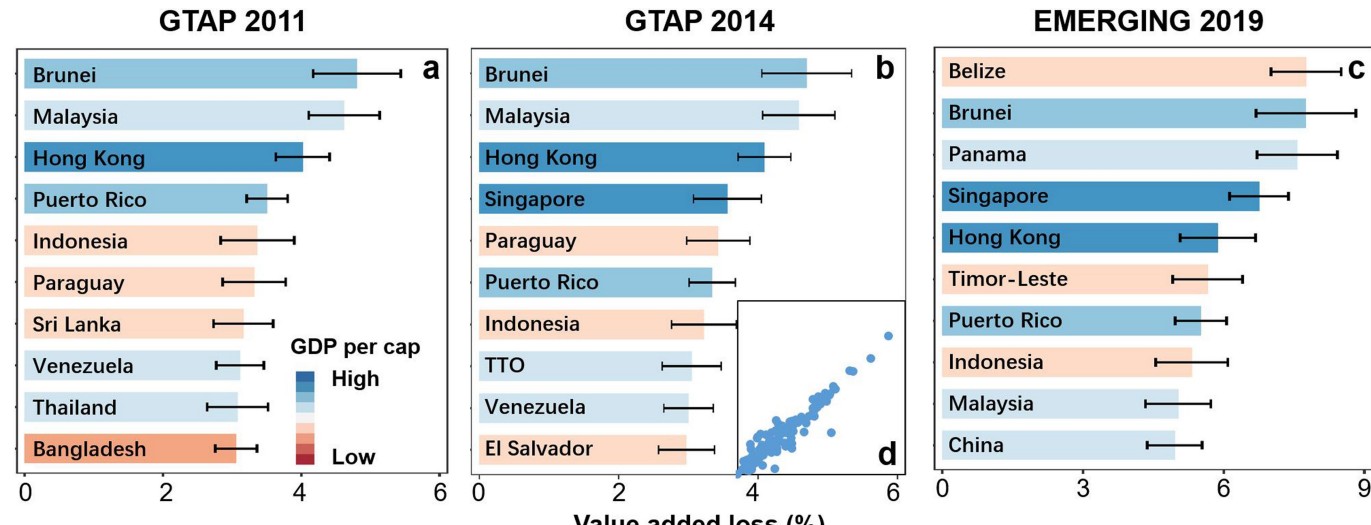

**Extended Data Fig. 4 | Estimates for the ten countries with the highest indirect losses under the SSP585 scenario using different base period trade data.** Estimates are displayed as 10-year averages for the year 2060, using the GTAP2011 (a), GTAP2014 (b) and EMERGING 2019 (c) databases separately. The colours of the bars represent GDP per capita from low to high. (d), indirect losses under different benchmark trade structures in each region. The horizontal axis measures the indirect losses as a percentage of GDP using the GTAP2014 trade structure and the vertical axis measures the indirect losses as a percentage of GDP using the GTAP2011 trade structure. Details of Extended Data Fig. 4d can be checked in Supplementary Fig. 6.

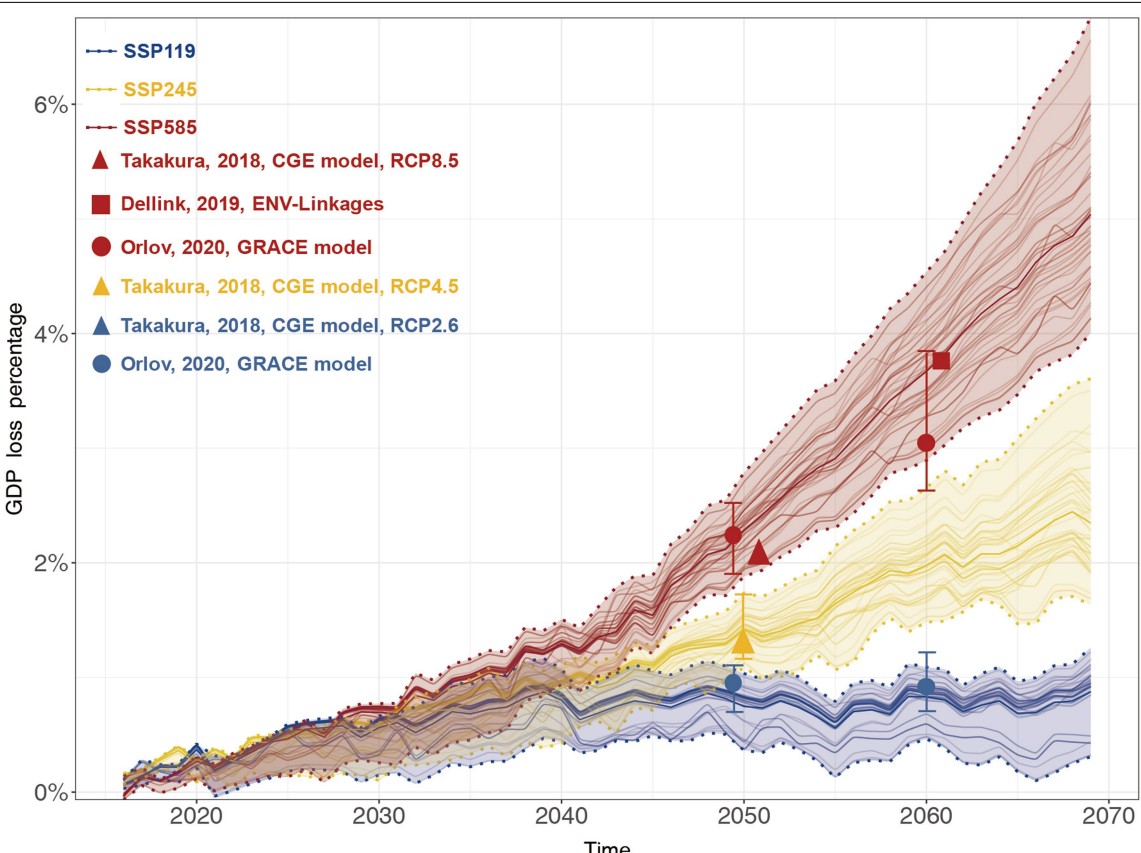

**Extended Data Fig. 5 | Global economic losses for each scenario under Monte Carlo simulations.** The assessment results of existing studies are marked with symbols for comparison. None of the previous studies were based on CMIP6 SSP119 scenario, so we use RCP2.6 to compare with the SSP119 scenario in our study. The studies above did not simulate health loss and the mean values of the health loss simulations in this paper were added for consistency.

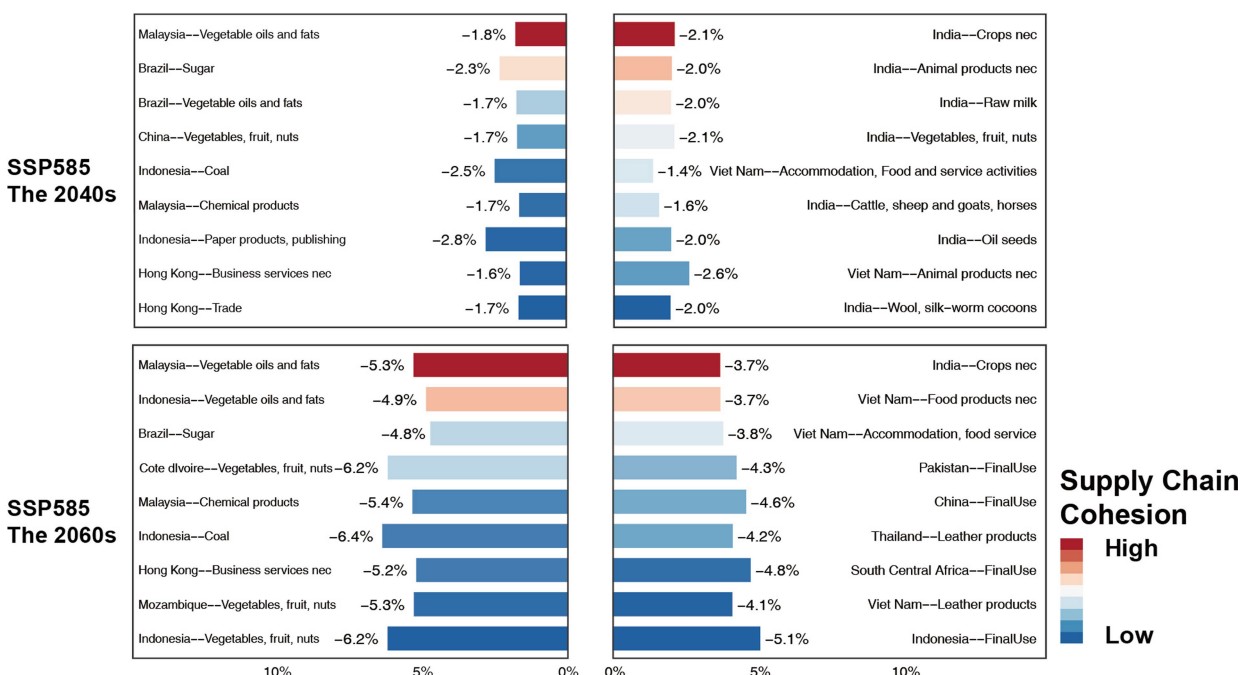

**Extended Data Fig. 6 | Impacts of heat stress on India food manufacturing supply chains.** (**a**), (**c**) panels represent the upstream sectors of the India's food production sector in 2040 and 2060, respectively. (**b**), (**d**) panels represent the downstream sectors. Each bar represents a key trading partner (i.e. sector with trade volume above the 50 percent quartile of trade volumes of the selected sector with all partner sectors) and the length represents the percentage decrease in product flow compared to the base period of 2014. The colours of the bars represent the cohesion level of the particular sector to the Indian food production sector from blue (weak) to red (strong), which is measured by the trade volume between the particular sector and the Indian food production sector.

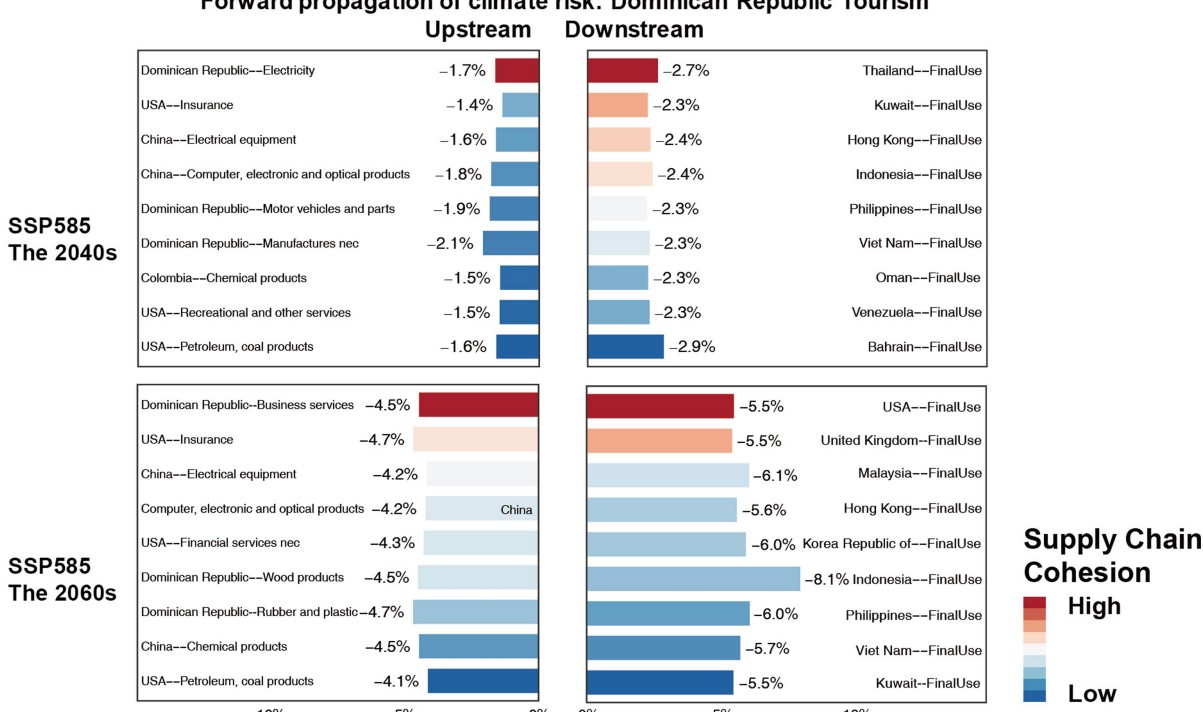

**Extended Data Fig. 7 | Impacts of heat stress on Dominican Republic tourism supply chains.** (**a**), (**c**) panels represent the upstream sectors of the Dominican Republic's tourism sector in 2040 and 2060, respectively. (**b**), (**d**) panels represent the downstream sectors. Each bar represents a key trading partner (i.e. sector with trade volume above the 50 percent quartile of trade volumes of the selected sector with all partner sectors) and the length represents the percentage decrease in product flow compared to the base period of 2014. The colours of the bars represent the cohesion level of the particular sector to the Dominican Republic's tourism sector from blue (weak) to red (strong), which is measured by the trade volume between the particular sector and the Dominican Republic's tourism sector.

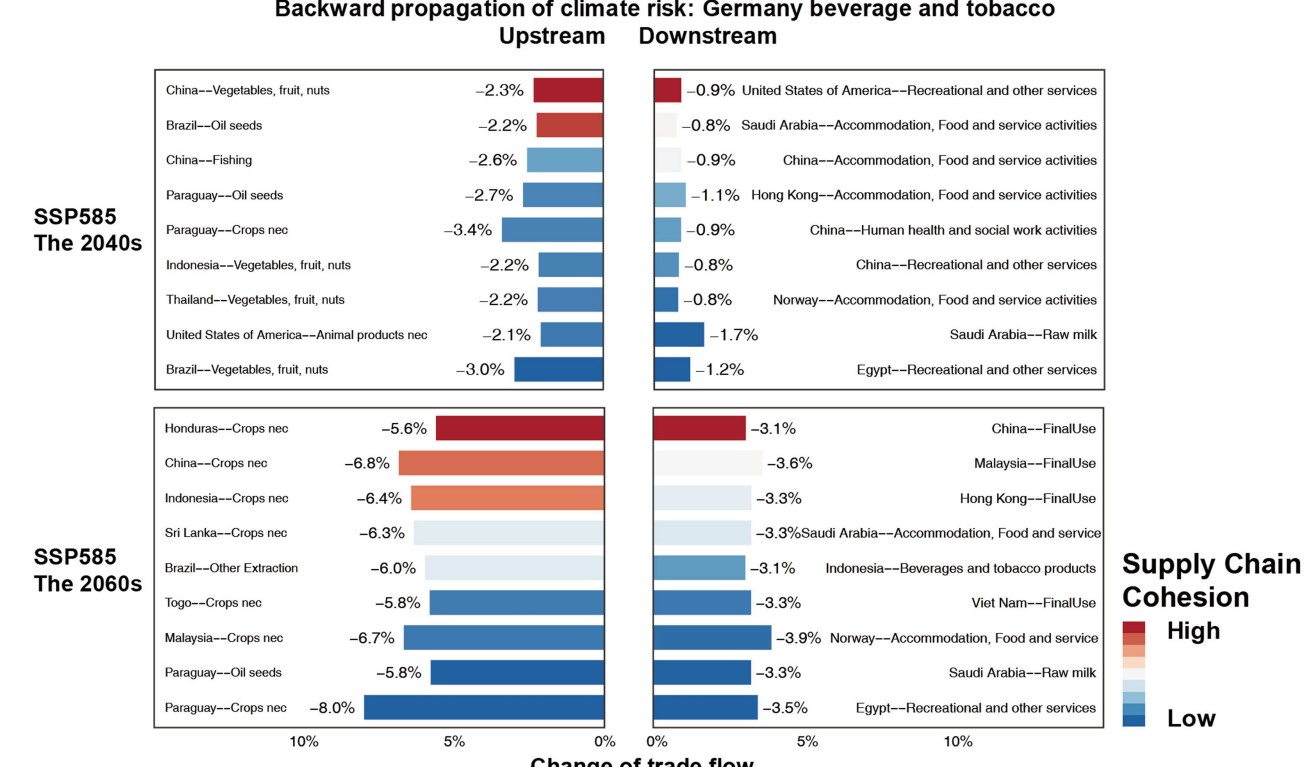

**Extended Data Fig. 8 | Impacts of heat stress on Germany beverages and tobacco products supply chains.** (**a**), (**c**) panels represent the upstream sectors of the Germany's beverages and tobacco products sector in 2040 and 2060, respectively. (**b**), (**d**), panels represent the downstream sectors.

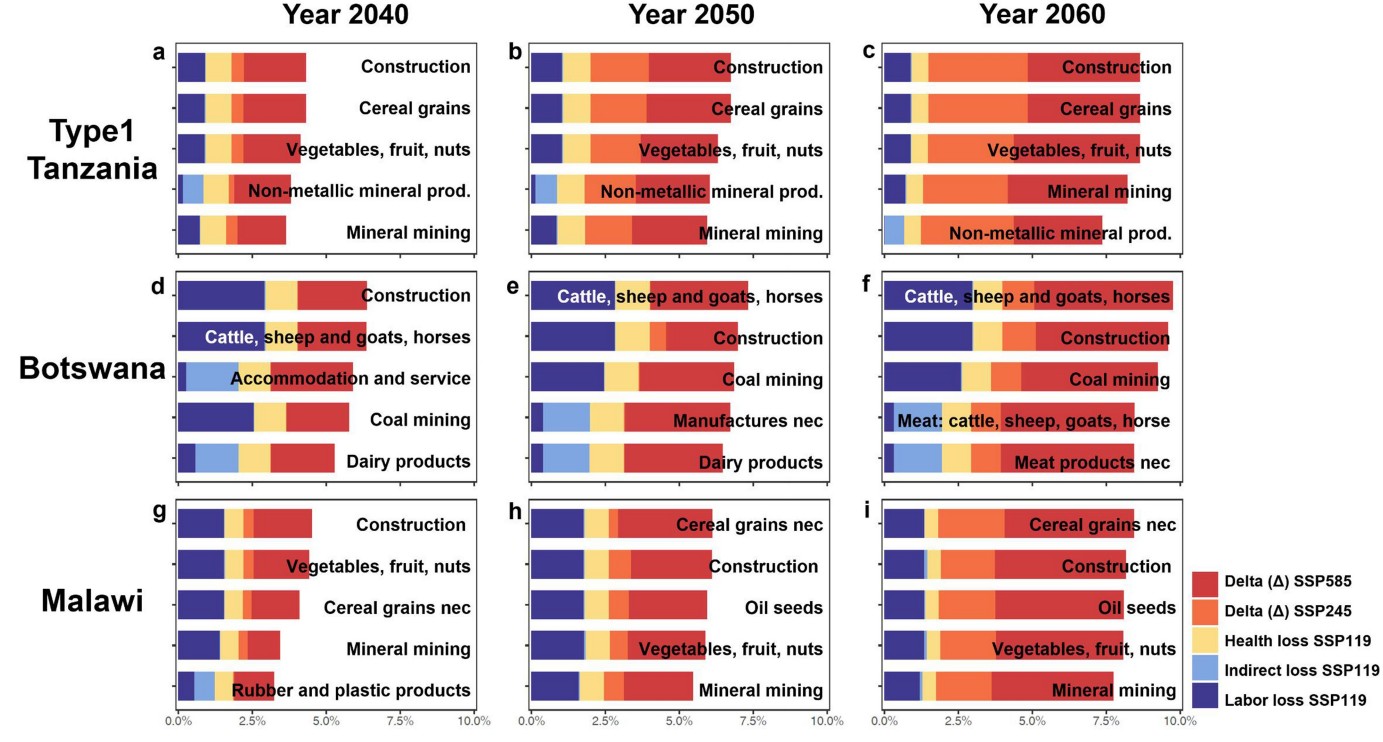

**Extended Data Fig. 9 | Sectoral loss patterns of type 1 countries.** The top 5 most vulnerable sectors in Tanzania (a–c), Botswana(d–f) and Malawi (g–i). The column length represents each sector's percentage loss of annual value-added. Sectors with the same loss percentage (e.g. wheat, rice, cereals, etc.) were combined. Colours indicate the three categories of losses in SSP119: health losses (Yellow bars), labour productivity losses (Blue bars) and supply-chain disruption losses (Green bars). The orange and red bars represent total loss increments for SSP245 and SSP585 (no distinction between types of loss in this part), respectively. The red dashed line indicates the mean value of losses for all sectors in the SSP585 scenario.

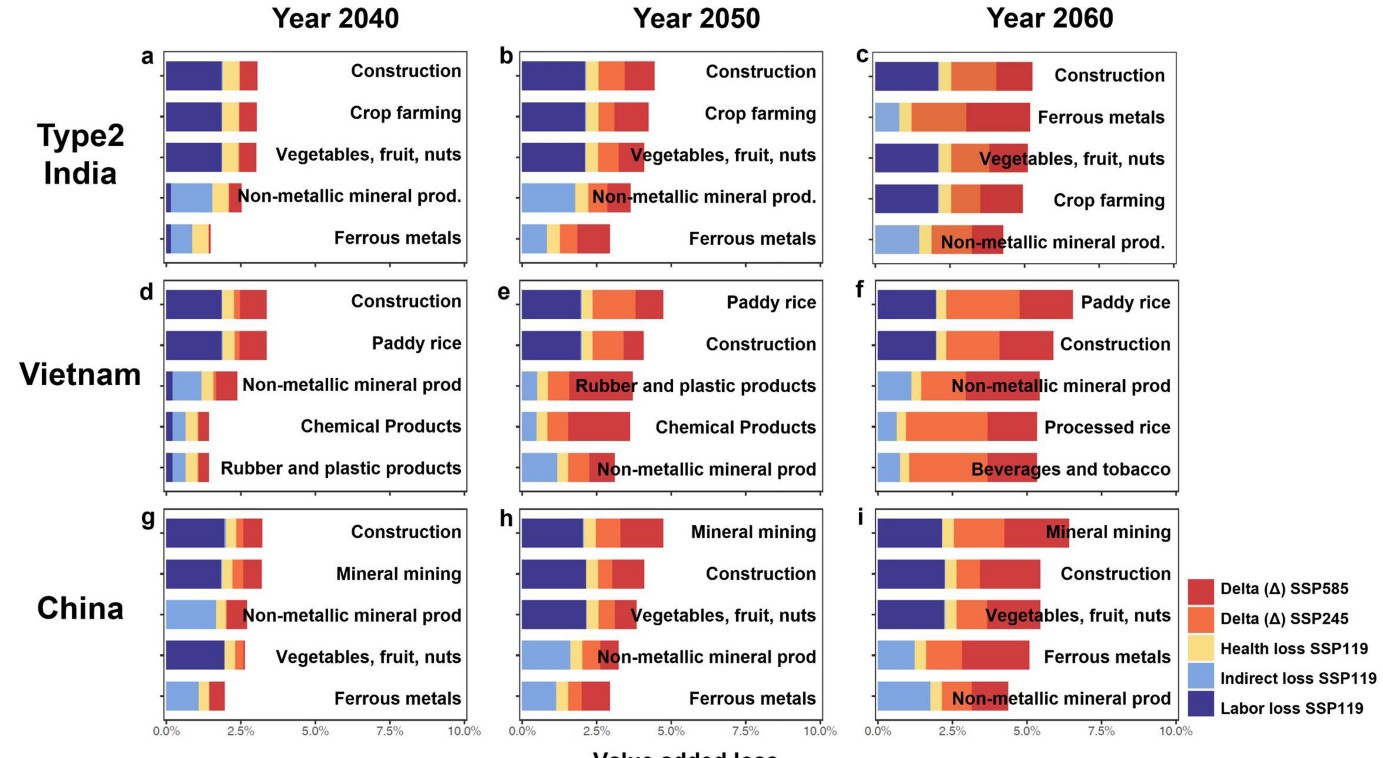

**Extended Data Fig. 10 | Sectoral loss patterns of type 2 countries.** The top 5 most vulnerable sectors in India (a–c), Vietnam (d–f) and China (g–i). The column length represents each sector's percentage loss of annual value-added. Sectors with the same loss percentage (e.g. wheat, rice, cereals, etc.) were combined. Colours indicate the three categories of losses in SSP119: health losses (Yellow bars), labour productivity losses (Blue bars) and supply-chain disruption losses (Green bars). The orange and red bars represent total loss increments for SSP245 and SSP585 (no distinction between types of loss in this part), respectively. The red dashed line indicates the mean value of losses for all sectors in the SSP585 scenario.

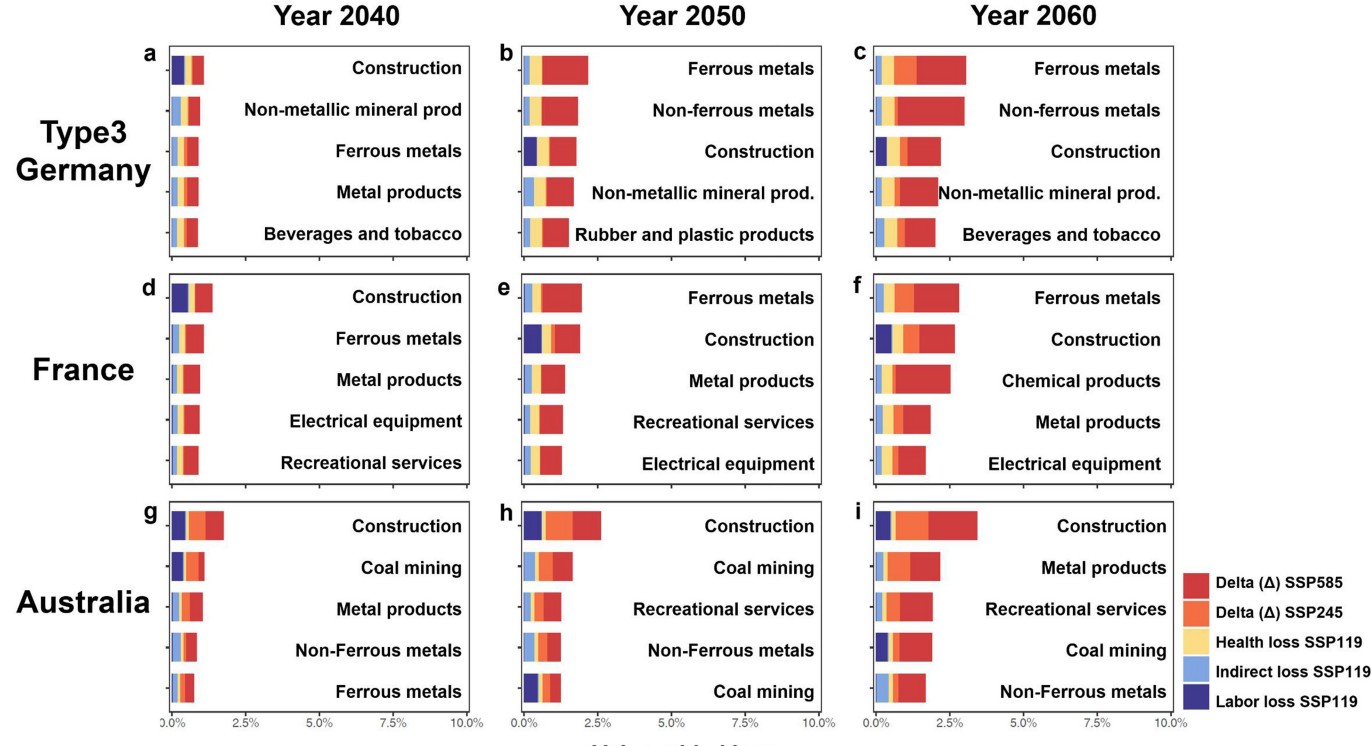

**Year 2040**     **Year 2050**     **Year 2060**

**Extended Data Fig. 11 | Sectoral loss patterns of type 3 countries.** The top 5 most vulnerable sectors in Germany (a–c), France (d–f) and Australia (g–i). The column length represents each sector's percentage loss of annual value-added. Sectors with the same loss percentage (e.g. wheat, rice, cereals, etc.) were combined. Colours indicate the three categories of losses in SSP119: health losses (Yellow bars), labour productivity losses (Blue bars) and supply-chain disruption losses (Green bars). The orange and red bars represent total loss increments for SSP245 and SSP585 (no distinction between types of loss in this part), respectively. The red dashed line indicates the mean value of losses for all sectors in the SSP585 scenario.

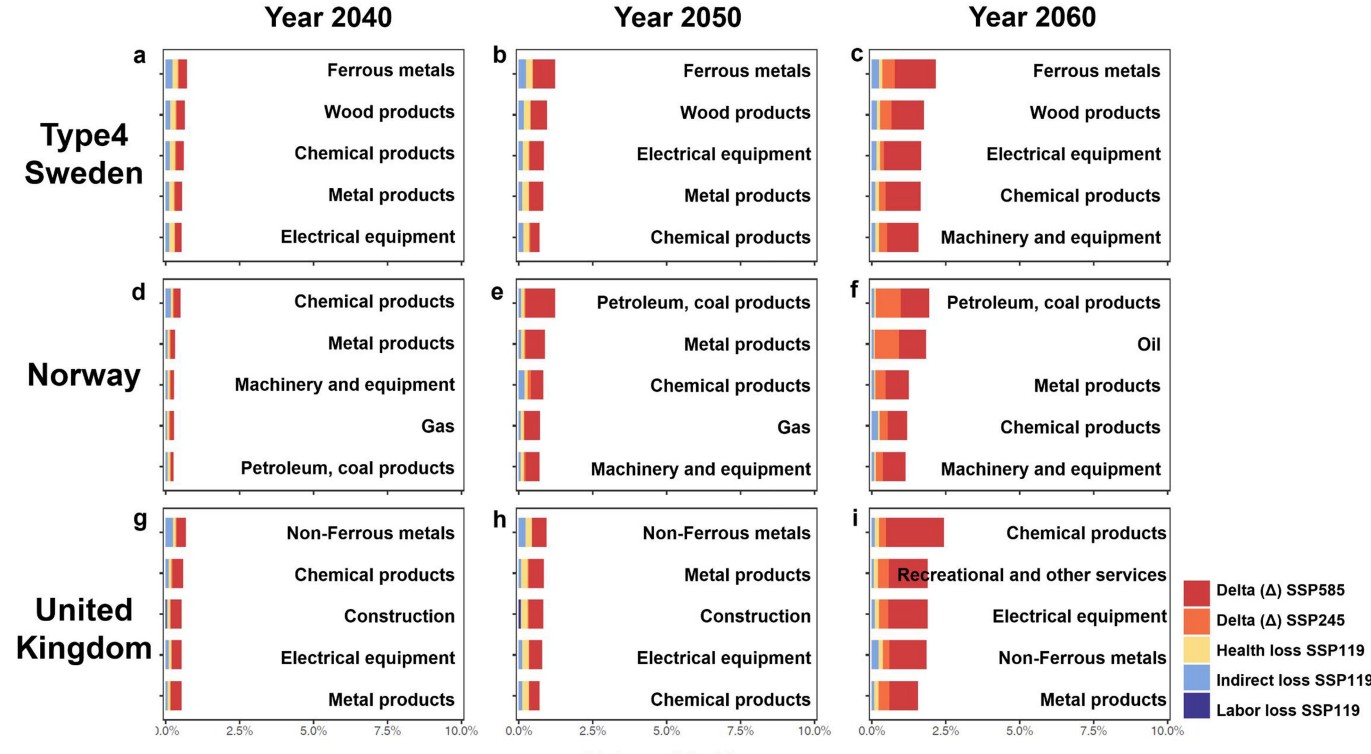

**Extended Data Fig. 12 | Sectoral loss patterns of type 4 countries.** The top 5 most vulnerable sectors in Sweden (a–c), Norway(d–f) and United Kingdom (g–i). The column length represents each sector's percentage loss of annual value-added. Sectors with the same loss percentage (e.g. wheat, rice, cereals, etc.) were combined. Colours indicate the three categories of losses in SSP119: health losses (Yellow bars), labour productivity losses (Blue bars) and supply-chain disruption losses (Green bars). The orange and red bars represent total loss increments for SSP245 and SSP585 (no distinction between types of loss in this part), respectively. The red dashed line indicates the mean value of losses for all sectors in the SSP585 scenario.

**Extended Data Table 1 | Global climate model data from CMIP6 used in our analysis (listed in alphabetical order)**

| Climate models | Research institute |
| --- | --- |
| CAMS-CSM1 | Chinese Academy of Meteorological Sciences (Beijing, China) |
| CanESM5 | Canadian Centre for Climate Modelling and Analysis (Victoria, British Columbia, Canada) |
| CESM2 | National Center for Atmospheric Research (Boulder, Colorado, USA) |
| CNRM-CM6-1 | Centre National de Recherches Météorologiques (Météo-France, Toulouse, France) |
| CNRM-ESM2-1 | Centre National de Recherches Météorologiques (Météo-France, Toulouse, France) |
| EC-Earth3 | European consortium of national meteorological services and research institutes. |
| GFDL-esm4 | Geophysical Fluid Dynamics Laboratory (Princeton, NJ, USA) |
| HadGEM3-GC31-MM | Met Office, Hadley Centre (Exeter, United Kingdom) |
| IPSL-CM6A-LR | Institute Pierre-Simon Laplace (Paris, France) |
| MIROC6 | Japan Agency for Marine-Earth Science and Technology, Atmosphere and Ocean Research Institute (Tokyo, The University of Tokyo), and National Institute for Environmental Studies (Tsukuba, Japan) |
| MPI-ESM1-2-HR | Max Planck Institute for Meteorology (Hamburg, Germany) |
| MRI-ESM2 | Meteorological Research Institute (Tsukuba, Japan) |
| NORESM2-MM | Norwegian Climate Centre (Bergen, Norway) |
| UKESM1 | Met Office Hadley Centre (Exeter, United Kingdom) and NERC (Natural Environment Research Council, United Kingdom) |

**Extended Data Table 2 | Relative risk (RR) of mortality caused by heatwaves (≥2 days of 95% mean temperature) for different climate zones**

| Climate Zone | Low RR estimate | Middle RR estimate | High RR estimate |
| --- | --- | --- | --- |
| Cold | 1.04467 | 1.05866 | 1.07282 |
| Moderate Cold | 1.06919 | 1.08826 | 1.10751 |
| Moderate Hot | 1.05357 | 1.06955 | 1.08571 |
| Hot | 1.0345 | 1.04794 | 1.06174 |

**Extended Data Table 3 | Input values for labour loss fraction**

| Work level (W) | Productivity mean | Productivity sd |
|:---:|:---:|:---:|
| 200 | 35.53 | 3.94 |
| 300 | 33.49 | 3.94 |
| 400 | 32.47 | 4.16 |