## [Peer Review File · Nature]

Manuscript Title: Global supply chains amplify economic costs of future extreme heat risk

Reviewer Comments & Author Rebuttals

Reviewer Reports on the Initial Version:

Referees' comments:

Referee #1 (Remarks to the Author):

First of all, I appreciate all the work the authors have put in this article. The figures look neat, it is well-written and the results are nicely presented. There are, however, a few methodological issues/unclarities that are important to address before this piece is suitable for publication. In particular in a journal like Nature.

While I agree that the concept of a value of a statistical life is widely used through the world, it is also somewhat controversial (in my opinion). Why is a human life of less value in Africa, compared to a life in the US. I find it hard to grasp that a human life in the US is 75 times more worth than a human life in some of the African countries. It should be more extensively discussed why the authors think this is still appropriate to use. And if they decide to pursue this approach with good arguments, at least provide also the results if one would keep the value the same across the world, or provide bandwidths of VOSL estimates. I will come back to that point later, but I think one should not use simply one value. There is a large uncertainty around this estimate and many studies show different estimates. How would that influence the results? I think this meta-analysis is quite interesting to read: <https://doi.org/10.1002/pam.10026>. Also in the methods, the relative risk is not further explained in the main text, which seems to be a very important parameter in this analysis? I would move that from supplementary to the main text (or at least directly refer to the supplementary table).

To be honest, also after reading the supplement (which is almost the same explanation as the main text, which is the case for quite a few supplementary elements), I still don't really get how this labour productivity loss is estimated. I don't understand yet how the indoor labour loss is integrated into this formula and how this is integrated with labour loss due to outside labour activities. The ERFs used should also be provided in this article as well, as it is now quite hard to understand this formula (while very important for the outcome of the estimations). And again, it feels like just one approach is used, while one can understand that the uncertainty range is large here. For publication in Nature, you should really tackle all these uncertainty elements. Or provide more convincing arguments why this is the best approach used. "In this research, we adopt the cumulative normal distribution function for its accuracy." That is a statement that doesn't really say much, and is also not backed up by any literature.

While the ARIIO model is indeed widely used, I believe it is really less suitable for use within a multiregional setting. The model indeed incorporates some form of substitution, but is it really endogenously solved? It is still a mostly linear model. While this may work well for a single-

country/single-region analyses, it is hard to believe that the entire world is negatively affected when a particular region is hit. While there might be indeed large negative ripple effects, there will also be some countries/ sectors that will gain from a disaster somewhere. This is nowhere mentioned in the article, and should be nuanced. Is the ARIO model really the right model to use here? Or are we actually inflating the losses because of the inherent linearity within the model? I would be happy to accept it if the authors can convince me with (quantitative) arguments on how their adjusted-ARIO model behaves realistically in a multiregional setting.

Another element I don't understand yet how the heatwaves are translated into supply chain disruption (e.g. the initial shock). It is the proportion of labour unavailable. Which equation does that relate to in the labour productivity losses? And is this based on the situation on each day? So does the model run daily from now to 2060, while climate change gradually make things worse? This should really be explained more clearly, as I can't find it anywhere. The way the initial shock is constructed is really important for the loss estimation. You can only reduce the labour productivity in different regions if they indeed have a heatwave happening at the same time. Else you would overestimate the losses. And if the model runs daily from now to 2060, how does it work when a new heat wave happens when the system still recovers from the previous heatwave? Do we see additional impacts due to consecutive events? This is not discussed at all.

Finally, I think how uncertainty is considered here, is really marginal. Especially considering the large modelling chain that is created in this study. The robustness test in the supplementary is really marginal. I know that the ARIO model runs fast, and the labour productivity losses also do not seem to be very computationally intensive. As the authors make really strong assumptions (often choosing only one specific equation or one specific parameter value), I would really like to see a more elaborate uncertainty analysis that shows that the signs and the signals are indeed robust (e.g. which countries and regions most affected, which sectors are most affected) etc.

A few small things:

- Figure 1, I would make the x-axis the same for all top figures. This will make it easier to compare. I also find it confusing that the SSPs are stacked, but that might be a matter of taste.
- In the description underneath Figure 1 (unfortunately page and line numbers are missing...) the authors say: "The global heatwave days would be 211% higher". This should be discussed more elaborately. That is a big statement. What is this based on, how is this indeed calculated? And what is the range within the climate models?
- Chapter 6 in the supplementary is called 'Climate sensitivity'. This term is normally used to address the "normal" uncertainty in our climate. As such, I would use a different term.
- Why did the authors choose to use GTAP, instead of EORA? I am not suggesting that EORA is better, just curious to see why GTAP is chosen, as EORA does contain more countries.

Referee #2 (Remarks to the Author):

A. Summary of the key results

The paper finds that economic losses from heat increase non-linearly and that indirect losses constitute a major driver in addition to direct losses on health and labor productivity. In addition to country hotspots, loss patterns are identified (clusters of countries) as well as propagation mechanisms (upstream vs downstream).

B. Originality and significance: if not novel, please include reference

The authors argue on page 2, 3rd paragraph, that “The direct mortality and productivity loss resulting from heat stress have been extensively studied. However, the indirect losses due to supply chain disruptions have not been fully quantified (Wang et al., 2021; Xia et al., 2019).”

This statement is not true: any CGE model captures these indirect losses (spillover effects across sectors and borders), and several papers have done so also in the context of heat stress (Knittel et al. 2020; Garcia-Leon et al. 2021; Takakura et al. 2018; Orlov et al. 2020). Similarly, MRIO/network based models have assessed indirect effects (Wenz and Leverman, 2016; Kulmer et al. 2020).

However, I agree with the authors that more research is needed to disentangle the indirect effects because (i) the previous literature has either devoted insufficient discussion to the indirect effects by only reporting the total/net effects; (ii) indirect effects can lead to an amplification or to a dampening of the direct effect and both directions deserve attention (Carter, 2021); and (iii) more research is needed on the influence of the underlying trade system that drive indirect effects (compare Willner et al., 2016, in the context of flooding).

Against this literature and research gap, I see two contributions of this manuscript: (i) a higher sectoral and regional resolution (65 sectors, 141 regions) which allows for looking into hotspots of this transmission; (ii) the identification of response patterns, or clusters, depending of sectoral composition of an economy and position in the supply chain.

C. Data & methodology: validity of approach, quality of data, quality of presentation

The climate, health and labor productivity modules are state-of-the art or go beyond that (by e.g. considering the rate of air-conditioning dependent on level of regional development in the different SSPs) and no changes are needed here. However, I do have concerns about the economic module: 1) I am concerned when key assumptions of a modeling class are relaxed to mimic behavior that is standard in other approaches. In the current model, this is the case when the authors introduce substitutability of products within a sector so that these products can be sourced from other regions in case of bottlenecks. This trade substitutability is indeed a key feature in CGE models (mostly based on the Armington, 1969, assumption of limited substitutability; occasionally the Melitz approach) but not in MRIO models to which the ARIO model belongs. Why is this substitution allowed in trade patterns but not in production where Leontief functions are used? Usually, fixed input structures and fixed trade patterns are argued to represent short term effects (within year effects) whereas substitutability in production processes and trade is argued to represent adjustment in the longer term (new equilibrium after several years). I find a mixture of trade substitutability and fixed production processes inconsistent.

Based on the available information, including the supplementary file, I also cannot assess which data is used to introduce this substitutability. Trade elasticities from GTAP or some other sources?

2) I also have concerns about the rationing approach, the second innovation of the paper. Wouldn't part of rationing be the result of increases in price or at least be accompanied by price increases? For instance, the Thailand flood 2011 led to substantial increases in prices of electronic components, in addition to disrupted supply.

3) Regarding the implementation of the labor productivity shock, more detail is needed on how the three types of labor are distributed to the sectors. Do you assume that e.g. all work in construction is heavy outdoor work? Or is there a combination depending on the shares of the three types of work?

4) If I understand the model setup correctly, it is assumed that capital is affected similarly as labor by the productivity shock due to the Leontief production function. But in contrast to flooding or a hurricane where capital is destroyed, in case of a heat-related labor productivity shock part of capital becomes idle (machinery is still there, but might not be used), so this is a temporary, not a permanent unavailability. How is this difference accounted for?

5) How are the different SSPs reflected in the economic module? It seems like the economic structure is entirely static to the base year dataset. E.g., in Figures 3-5, is the difference between 2060s and 2080s entirely owed to different changes in climate and changes in population? But SSP narratives also differentiate in terms of globalization and trade openness. At the very least, it should be clarified that this dimension is missing from the analysis.

6) Figure 1: While it seems to become established practice since AR6 to report fixed combinations of RCP and SSP scenarios, it would be interesting to see how much of this non-linear increase in GDP loss is driven by change in climate and by change in population.

7) I find the approach in Figure 3 very interesting as it illustrates patterns of transmission depending on a country's exposure and vulnerability and it does not focus on the usual large economies (US, EU, China). But I strongly recommend to use statistical analysis to identify these clusters and to be careful when stating which countries are similar in their response patterns (e.g. there is a considerable difference in exposure to transboundary risks across European countries, see Benzie et al., 2019). Moreover, it is vital to mention that the indirect loss depends strongly on the underlying data (GTAP vs EORA) and base year (2014 or other). Please consider conducting sensitivity checks similar to Willner et al. (2016).

8) While I consider backward and forward propagation a very important topic to look into, clarification is needed on the selection of the two cases in Figures 4 and 5. I am not questioning that they have an important contribution to domestic GDP and are reliant on international supply chains, but many other examples could be picked. Please motivate your choice against other potential alternatives.

Minor points:

1) Figure 1: It would be good to include such a decomposition in the Supplementary Material. And please add a row of maps for SSP2-4.5.

2) In Figure 2, please clarify what the error bars represent (variability across years?)

3) I suggest to increase consistency between Figure 2 and Figures 4 and 5: either look into differences across time slices for one scenario (SSP5-8.5) or into differences across scenarios for one time slice (SSP5-8.5 vs SSP1-2.6 for 2060). Again, discuss the caveat that the socio-economic structure is fixed at the base year.

4) In all figures, please clarify that the value added % numbers are % of total value added, ie GDP,

and not sectoral value added.

D. Appropriate use of statistics and treatment of uncertainties

While the description of the countries in the three clusters in Figure 3 is interesting and the cases are arguably selected as representative, please consider identifying the clusters by means of statistical analysis. Given the number of sectors, regions and variables, this should be feasible.

Similarly for Figures 4 and 5: Please use some traceable (statistical?) approach for the selection of the two cases.

E. Conclusions: robustness, validity, reliability

In the discussion section, please add that indirect effects can not only work as risk amplifier but also as a risk reducer by means of trade rerouting or diversification as well as building up stocks, the latter point being discussed in the sensitivity analysis in the Supplementary Material.

F. Suggested improvements: experiments, data for possible revision

One of the main socio-economic uncertainties is sectoral composition and trade structure. Given the focus of the paper, please consider different base years (e.g. GTAP 9 and earlier versions which are now included as part of the same license) and potentially alternative datasets (e.g. EORA instead of GTAP) to test for the robustness of results.

If you decide to keep substitutability in trade, expand the sensitivity checks (in Section 2 of the Supplementary Material) with regard to trade elasticities.

G. References: appropriate credit to previous work?

See my comments on point B above.

H. Clarity and context: lucidity of abstract/summary, appropriateness of abstract, introduction and conclusions

In the abstract, I am not sure about the sentence “Severely affected industries such as farming and construction in Tanzania are expected to lose 6.8% to 7.7% of their value added, while the food processing industry in Italy and India is expected to lose 4.9% to 6.5% of their value-added.” – Does “their value added” mean sectoral value added or economy-wide value added? In Figure 2 in the main text, my reading was that it was always % of economy-wide value added (GDP), not sectoral. Please clarify!

References cited:

Benzie, M., Carter, T. R., Carlsen, H., & Taylor, R. (2019). Cross-border climate change impacts: Implications for the European Union. *Regional Environmental Change*, 19(3), 763–776.

<https://doi.org/10.1007/s10113-018-1436-1>

Carter, T. R., Benzie, M., Campiglio, E., Carlsen, H., Fronzek, S., Hildén, M., Reyner, C. P. O., & West, C. (2021). A conceptual framework for cross-border impacts of climate change. *Global Environmental Change*, 69. Scopus. <https://doi.org/10.1016/j.gloenvcha.2021.102307>

García-León, D., Casanueva, A., Standardi, G., Burgstall, A., Flouris, A. D., & Nybo, L. (2021). Current and projected regional economic impacts of heatwaves in Europe. *Nature Communications*, 12(1),

5807. <https://doi.org/10.1038/s41467-021-26050-z>

Knittel, N., Jury, M. W., Bednar-Friedl, B., Bachner, G., & Steiner, A. K. (2020). A global analysis of heat-related labour productivity losses under climate change—Implications for Germany’s foreign trade. *Climatic Change*, 160(2), 251–269. Scopus. <https://doi.org/10.1007/s10584-020-02661-1>

Kulmer, V., Jury, M., Wong, S., & Kortschak, D. (2020). Global resource consumption effects of borderless climate change: EU’s indirect vulnerability. *Environmental and Sustainability Indicators*, 8, 100071. <https://doi.org/10.1016/j.indic.2020.100071>

Matsumoto, K. (2019). Climate change impacts on socioeconomic activities through labor productivity changes considering interactions between socioeconomic and climate systems. *Journal of Cleaner Production*, 216, 528–541. <https://doi.org/10.1016/j.jclepro.2018.12.127>

Orlov, A., Sillmann, J., Aunan, K., Kjellstrom, T., & Aaheim, A. (2020). Economic costs of heat-induced reductions in worker productivity due to global warming. *Global Environmental Change*, 63, 102087. <https://doi.org/10.1016/j.gloenvcha.2020.102087>

Takakura, J., Fujimori, S., Takahashi, K., Hijioka, Y., Hasegawa, T., Honda, Y., & Masui, T. (2017). Cost of preventing workplace heat-related illness through worker breaks and the benefit of climate-change mitigation. *Environmental Research Letters*, 12(6), 064010. <https://doi.org/10.1088/1748-9326/aa72cc>

Wenz, L., & Levermann, A. (2016). Enhanced economic connectivity to foster heat stress–related losses. *Science Advances*, 2(6), e1501026. <https://doi.org/10.1126/sciadv.1501026>

Willner, S. N., Otto, C., & Levermann, A. (2018). Global economic response to river floods. *Nature Climate Change*, 8(7), 594–598. <https://doi.org/10.1038/s41558-018-0173-2>

Author Rebuttals to Initial Comments:

Overall response to reviewers

Dear editor and reviewers,

Thank you very much for your consideration of our submission and your efforts in managing its review. We very much appreciate the reviews we received, which are very constructive and greatly improved our paper. We have revised the paper accordingly. Please find our point-by-point response to the comments as listed below.

Referee #1 (Remarks to the Author):

First of all, I appreciate all the work the authors have put in this article. The figures look neat, it is well-written and the results are nicely presented. There are, however, a few methodological issues/unclarities that are important to address before this piece is suitable for publication. In particular in a journal like Nature.

While I agree that the concept of a value of a statistical life is widely used through the world, it is also somewhat controversial (in my opinion). Why is a human life of less value in Africa, compared to a life in the US. I find it hard to grasp that a human life in the US is 75 times more worth than a human life in some of the African countries. It should be more extensively discussed why the authors think this is still appropriate to use. And if they decide to pursue this approach with good arguments, at least provide also the results if one would keep the value the same across the world, or provide bandwidths of VOSL estimates. I will come back to that point later, but I think one should not use simply one value. There is a large uncertainty around this estimate and many studies show different estimates. How would that influence the results? I think this meta-analysis is quite interesting to read: <https://doi.org/10.1002/pam.10026>. Also in the methods, the relative risk is not further explained in the main text, which seems to be a very important parameter in this analysis? I would move that from supplementary to the main text (or at least directly refer to the supplementary table).

RE: Thank you for your informative advice. Firstly, our core objective in this study is not to update and optimise the VSL, but to give an assessment of climate change losses based on the results of the established literature. However, we do agree that there is significant uncertainty (Mrozek & Taylor, 2002) in VSL. We have therefore carefully considered your recommended meta-analysis and done an uncertainty analysis using the average VSL. We have calculated a global average, equal VSL as you suggested and used it to recalculate the model. And we found that because of the large differences in economic size and income between countries in global-scale studies, a uniform VSL might lead to inequality in the calculations between developed and developing countries, ultimately underestimating global economic losses. We have added this point to the uncertainty

analysis for comparison. Please see the following paragraphs:

In supplementary materials 2.1 Uncertainty of health loss

The uncertainty of heatwave related mortality is propagated through the uncertainty of relative risk (RR) used in equation 4 (See Extended Data Table 2). The uncertainty of RR is determined by the meta-analysis conducted by Guo et al. (Guo et al., 2017) on heatwave related mortality data from 400 communities across 18 counties/regions with different climate. The uncertainty in VSL is also tested through an additional sensitivity test, which assumes that all life would be valued equally across the world. For such a test, an averaged VSL is calculated by summing up each country's income based VSL times its population then divided by the total population of the world. After compiling and calculating the data, we obtained a global average VSL of 2.89 million USD in year 2020. We use the globally equal VSL calculation above as a reference.

We found that under the assumption of a global average VSL, the economic inequality of global health losses would be substantially higher (overestimated) between developed and developing economies (Supplementary Figure 3). Under the assumption of global equality VSL, global economic losses fall by 5- 20% compared to the setting of heterogeneous VSL (Supplementary Figure 2). This is because the global average VSL is much lower than the benchmark VSL in heatwave-vulnerable countries such as Russia, the USA and France, which have large populations and a high number of excess deaths due to heatwaves each year. With a global average VSL the economic losses in these countries would be significantly reduced by 2 to 8 times. Correspondingly, health losses in heat wave vulnerable countries such as India, Bangladesh and Ethiopia would be 2 to 7 times higher, as the baseline VSL is 1.2 million USD for India and only 0.4 million USD for Ethiopia. The pattern of economic losses as a percentage of national GDP for each country under the two VSL scenarios is shown in Supplementary Figure 3 (patterns under SSP119 and SSP245 scenarios are similar). However, it is reasonable to believe that economic losses in developing countries are grossly overestimated, as many Latin American and African countries have economic losses of more than 10% of their GDP in that year, based on the global average VSL (which does not take into account socio-economic factors such as income). Tajikistan lost as much as 25.2% of its GDP in 2060, and Malawi as much as 65.9%. Such unreasonable results are also the reason why the World Bank and a host of previous studies recommend the benchmark VSL we use. We have therefore used the global average VSL results as a reference for the supporting material

and the results in the text are more scientific and reasonable.

To be honest, also after reading the supplement (which is almost the same explanation as the main text, which is the case for quite a few supplementary elements), I still don't really get how this labour productivity loss is estimated. I don't understand yet how the indoor labour loss is integrated into this formula and how this is integrated with labour loss due to outside labour activities.

The ERFs used should also be provided in this article as well, as it is now quite hard to understand this formula (while very important for the outcome of the estimations).

RE: Thank you for the kind correction. We have explained each formula in detail in the supplementary material. And a curve as a function of labour productivity loss ~ WBGT is specifically drawn so that the reader can visualise how productivity is lost due to climate change. The calculation of labour losses in the Indian food processing industry is used as an example of how the parameters are entered into the function. Please see the following paragraphs:

$$\text{Loss fraction} = \frac{1}{2} \left(1 + \text{ERF} \left(\frac{\text{WBGT}_{\text{outdoor or indoor}} - \text{Prod}_{\text{mean}}}{\text{Prod}_{\text{SD}} * \sqrt{2}} \right) \right) \quad (1.)$$

Of which:

$$\begin{aligned} & \text{ERF}(z) \\ &= \frac{2}{\sqrt{\pi}} \int_0^z e^{-t^2} dt \end{aligned} \quad (2.)$$

Fig. S1 Curve of the function of labour loss and WBGT (workload of 300W). As WBGT rises, labour losses increase rapidly in the interval of 30~40 degrees initially, eventually edging closer to 100%.

For example, to calculate workability loss fraction in India food production sector (300W, indoor), we bring the corresponding parameters including $WBGT_{indoor}$, $Prod_{mean}$ of 33.49, and $Prod_{SD}$ of 3.94 into formula (1). The calculation was conducted for all grids having a resolution of $0.5^{\circ} \times 0.5^{\circ}$ in India, and the results were then aggregated for each region and sector weighted by population distribution.

And again, it feels like just one approach is used, while one can understand that the uncertainty range is large here. For publication in Nature, you should really tackle all these uncertainty elements. Or provide more convincing arguments why this is the best approach used. "In this research, we adopt the cumulative normal distribution function for its accuracy." That is a statement that doesn't really say much, and is also not backed up by any literature.

We appreciate your rigorous and careful suggestion. **The reasons for the choice of function are explained in detail and alternative functions that have been used in the previous literature are newly applied as part of the robustness test of the assessment.** In fact, in our previous literature research, the different loss functions are very similar in terms of computational results. We chose the ERF loss function because it was applied and tested in the Lancet report for the three years 2019-2021. It is therefore currently considered by the academic community to be robust and reliable. Please see the following paragraphs.

1.2.2 Expose Function of labor productivity in SI

The increase in daily temperatures affects the efficiency of workers and reduces safe working time. A compromise in endurance capacity due to thermoregulatory stress was already evident at 21°C . Different studies used similar methods to evaluate the labor loss function. The form of logistic function with "S" shape has become the consensus of the academic community, but the specific functional equation and parameters are various in different studies. The loss functions used in mainstream research include exponential function (Bröde et al., 2018) as formula (5), cumulative normal distribution function (Cai et al., 2021; Watts et al., 2019a) as formula (6) and so on. In this research, we adopt the cumulative normal distribution function (6) as our benchmark functions because it was extensively applied and case proven in 3-years reports of the Lancet Countdown on health and climate change (Cai et al., 2021; Romanello et al., 2021; Watts et al., 2019a, 2021).

While the Hothaps function (5) is subject to great parameter uncertainty due to being based on a few empirical studies, so we use it to test for the sensitivity of our estimates.

2.2 Uncertainty of labor productivity loss

For the uncertainty analysis of labour productivity losses, we used the Hothaps labor loss function as a comparison. Where the set of parameters α_1 and α_2 were equal to (34.64, 22.72) for low, (32.93, 17.81) for moderate and (30.94, 16.64) for high workload, respectively (García-León et al., 2021).

In terms of the assessment of the absolute labour loss rate, the Hothaps logistic function has lower results than the ERF function due to the assumption that at least 10% of labour time is retained, i.e., working is possible for 6 min. within each hour even under extreme heat. The Hothaps logistic function, which assumes an upper limit of 90% labour productivity loss, results in a slightly lower total loss assessment of around 5%-12% compared to the ERF function. However, this study assesses the additional amount of damage caused by climate change. The difference in calculation of the function itself is offset when subtracted from the base period. So the global labour losses calculated using the two functions are therefore very close (within 5% of each other) for the uncertainty analysis. Even when looking at the subdivision of losses per country per sector, the two functions give similar results in measuring climate change labour losses and largely match in terms of relative loss size ranking. The comparison between two functions is shown in Supplementary Figure 4 and Supplementary Figure 5, using the construction sector as an example.

While the ARIO model is indeed widely used, I believe it is really less suitable for use within a multiregional setting. The model indeed incorporates some form of substitution, but is it really endogenously solved? It is still a mostly linear model. While this may work well for a single-country/single-region analyses, it is hard to believe that the entire world is negatively affected when a particular region is hit. While there might be indeed large negative ripple effects, there will also be some countries/ sectors that will gain from a disaster somewhere. This is nowhere mentioned in the article, and should be nuanced. Is the ARIO model really the right model to use here? Or are we actually inflating the losses because of the inherent linearity within the model? I would be happy to accept it if the authors can convince me with (quantitative) arguments on how

their adjusted-ARIO model behaves realistically in a multiregional setting.

RE: The ARIO model has been applied and tested in multi-regional scenarios in the case studies of Japan (Inoue & Todo, 2019), the United States (Markhvida et al., 2020; Wang et al., 2021) and the Europe (Mendoza-Tinoco et al., 2020). These are examples of latest quantitative research, published in a sub-series of *Nature* and in core journals in risk assessment research areas such as *Risk Analysis*. The substitutability issue you mention is important and in ARIO we have endogenously optimised the substitutability so that it is no longer a linear model. There are not only shocks in the dynamic ARIO model, but also recoveries, which create additional demand to rebuild the stock. Thus, as you say, not the entire world is negatively affected when a particular region is hit. Each country is affected in a heat stress situation in both positive and negative ways. On the one hand, economic output decreases when production is constrained in its own or closely related areas of the supply chain. On the other hand, countries that are minimally affected by climate change will benefit from this additional demand. To ensure greater robustness, we have added the use of dynamic CGE simulations after careful consideration of your comments, see Supplementary Figure 8. The same core conclusions and slightly lower results were obtained, but a segmented regional and sectoral perspective was not available (the flawed nature of CGE in multi-sectoral solving)

Another element I don't understand yet how the heatwaves are translated into supply chain disruption (e.g. the initial shock). It is the proportion of labour unavailable. Which is equation does that relate to in the labour productivity losses? And is this based on the situation on each day? So does the model run daily from now to 2060, while climate change gradually make things worse? This should really be explained more clearly, as I can't find it anywhere. The way the initial shock is constructed is really important for the loss estimation. You can only reduce the labour productivity in different regions if they indeed have a heatwave happening at the same time. Else you would overestimate the losses. And if the model runs daily from now to 2060, how does it work when a new heat wave happens when the system still recovers from the previous heatwave? Do we see additional impacts due to consecutive events? This is not discussed at all.

RE: Firstly, yes it is the proportion of labour unavailable that is translated as shocks into supply

chain modelling. Heatwaves are translated into supply chain disruption by labour productivity losses based on eq(1) above, from daily WBGT to labour productivity losses in each sector. The shocks of labour productivity losses are calculated daily and then aggregated to yearly loss and input into the ARIO model. The labour loss rate based on Equation 1 minus the 2015 baseline value is used as shocks, so that only additional losses due to climate change have an impact. And we have done robustness tests using more than one method from the established literature.

Secondly, the time step for the ARIO model is one year. In more detail, we multiply the daily productivity per worker by the productivity lost ratio to estimate economic costs of productivity losses due to heat exposure. The calculation was conducted for all grids having a resolution of $0.5^{\circ} \times 0.5^{\circ}$, and the results were then aggregated for each region and sector weighted by population distribution. Here we have calculated heat wave conditions and labour losses for each spatial grid, not a rough calculation of regional scale losses, so there is no overestimation of losses as you suggest. But we acknowledge that we have not taken into account situations such as workers working overtime at night. Finally we input the ARIO model using the annual total labour productivity loss per country per industry.

Thirdly, the recovery and shocks of the model are synchronised. Each year, if weather conditions improve compared to the base year (in 2015), productivity recovers with reduced severity of heat stress and inventories are replenished. If the heat stress becomes more severe compared to the base year, a new shock is imposed. We did see additional impacts due to consecutive events. So if you look at the evolution of global GDP losses as illustrated in ED figure 5, it shows an up-and-down sawtooth shape.

Finally, I think how uncertainty is considered here, is really marginal. Especially considering the large modelling chain that is created in this study. The robustness test in the supplementary is really marginal. I know that the ARIO model runs fast, and the labour productivity losses also do not seem to be very computationally intensive. As the authors make really strong assumptions (often choosing only one specific equation or one specific parameter value), I would really like to see a more elaborate uncertainty analysis that shows that the signs and the signals are indeed robust (e.g. which countries and regions most affected, which sectors are most affected) etc.

RE: We appreciate this point, and we agree that the robustness test for the modeling was perhaps

too marginal. We elaborate in more detail the uncertainty parameter intervals for the three types of losses and do a Monte Carlo analysis and the results of the classic literature on the assessment of heat-related economic losses have been collected for comparison. Specifically, we ran the parameter sets for each of the three types of losses 50 times and finally aggregated them into the model, using Monte Carlo to simulate the economic loss results for about 10000 periods. We have also summarised the results of previous assessments of different models based on CMIP5 data and similar RCP scenarios. The results are shown in ED figure 5. We have also added upper and lower bounds for many of the parameters and have shown them in Supplementary Figure 7.

As shown in Extended Data Fig. 5, Supplementary Figure 7 and Supplementary Figure 8, **the core finding of this study, that the global supply chain amplification effect leads to a non-linear increase in economic losses from heat stress, is robust to a large number of rigorous tests.** The difference is only that the point at which the exponential growth occurs changes with the adaptive setting. Supplementary Figure 7 e-h is the baseline scenario similar to the result in the main text, where parameters such as inventories and excess production capacity are used in line with the facts and are more common in previous studies. In the high adaptation scenario (Supplementary Figure 7 a-d), economic losses increase at a high rate from 2050 to 2060. This is because each country is assumed to have sufficient production stocks for vulnerable industries to maintain production for a short period of time even in the event of upstream supply disruptions. And when extreme heat is frequent, new adaptive technologies can be generated and widely used in time to offset the health and labour productivity losses of heat stress to some extent. In contrast, when the adaptation parameter is set at its lowest setting (Supplementary Figure 7 i-l), economic losses will rapidly increase in very near future. In summary, we find that the strong adaptation hypothesis can significantly reduce economic losses in the short term, under low emission scenarios. However, adaptation strategies for long-term climate change stress, especially under high emission scenarios, are not effective in reducing economic losses on long time scales (beyond 2060).

A few small things:

- Figure 1, I would make the x-axis the same for all top figures. This will make it easier to compare.

I also find it confusing that the SSPs are stacked, but that might be a matter of taste.

RE: Thank you for your advice. The x-axis of health loss, labour loss, and consequential loss is unified (0 to 3%). But the total loss added up to more than the upper limit. If we use the range of total loss (0~5%) as a criterion, the three types of loss can only effectively occupy one third of the figure. This can leave loss patterns and trends not clearly represented. As for why the SSP was stacked, it was because we wanted to visually demonstrate how much of the global economy could be saved under different emissions reduction efforts. If losses are stacked by time, there is a gradual reduction in losses in the SSP119 scenario, which cannot be stacked.

- In the description underneath Figure 1 (unfortunately page and line numbers are missing...) the authors say: "The global heatwave days would be 211% higher". This should be discussed more elaborately. That is a big statement. What is this based on, how is this indeed calculated? And what is the range within the climate models?

RE: We have added the following statement when the first heatwave days appeared in this section:
The number of heatwave days (definition and calculation detailed in the method section 1.1) would increase by 27% compared to 2022

We also added the following definition and calculation in the method section 1.1:

For the calculation of annual heatwave days in a country or the whole world, the number of days under heatwave periods of individual grid cells within one country or the entire world would be summed up.

- Chapter 6 in the supplementary is called 'Climate sensitivity'. This term is normally used to address the "normal" uncertainty in our climate. As such, I would use a different term.

RE: Thank you for your advice. Our intention was to express the vulnerability of these countries or sectors to climate change. We have changed the wording of the title to vulnerability.

- Why did the authors choose to use GTAP, instead of EORA? I am not suggesting that EORA is better, just curious to see why GTAP is chosen, as EORA does contain more countries.

RE: On the one hand, GTAP has a higher sectoral resolution and is better suited to the topic of discussion in this article. While EORA's global MRIO table has 26 consistent sectors. GTAP's database has been shown to be robust in several studies in climate change loss assessment especially related to trade (Aguilar et al., 2019; García-León et al., 2021; Knittel et al., 2020). On the other hand, price for data license and upgrade is also a consideration for us to be honest. the GTAP database is cheaper (\$730 for Academic license), EORA is more expensive for us (€3,990 for Academic license).

Referee #2 (Remarks to the Author):

A. Summary of the key results

The paper finds that economic losses from heat increase non-linearly and that indirect losses constitute a major driver in addition to direct losses on health and labor productivity. In addition to country hotspots, loss patterns are identified (clusters of countries) as well as propagation mechanisms (upstream vs downstream).

B. Originality and significance: if not novel, please include reference

The authors argue on page 2, 3rd paragraph, that “The direct mortality and productivity loss resulting from heat stress have been extensively studied. However, the indirect losses due to supply chain disruptions have not been fully quantified (Wang et al., 2021; Xia et al., 2019).”

This statement is not true: any CGE model captures these indirect losses (spillover effects across sectors and borders), and several papers have done so also in the context of heat stress (Knittel et al. 2020; Garcia-Leon et al. 2021; Takakura et al. 2018; Orlov et al. 2020). Similarly, MRIO/network based models have assessed indirect effects (Wenz and Leverman, 2016; Kulmer et al. 2020).

However, I agree with the authors that more research is needed to disentangle the indirect effects because (i) the previous literature has either devoted insufficient discussion to the indirect effects by only reporting the total/net effects; (ii) indirect effects can lead to an amplification or to a

dampening of the direct effect and both directions deserve attention (Carter, 2021); and (iii) more research is needed on the influence of the underlying trade system that drive indirect effects (compare Willner et al., 2016, in the context of flooding).

Against this literature and research gap, I see two contributions of this manuscript: (i) a higher sectoral and regional resolution (65 sectors, 141 regions) which allows for looking into hotspots of this transmission; (ii) the identification of response patterns, or clusters, depending of sectoral composition of an economy and position in the supply chain.

RE: Thank you for the correction and recognizing the value of our research. Our original intention was to indicate that previous studies have not specifically and adequately discussed indirect losses (i.e. its well-refined country and sectoral patterns and mechanism). As you can see, the articles you mention have been carefully read and studied by us and cited in the comparison of methods or results in previous manuscripts. The uncritical presentation of the literature review section has been revised and you are referred to the following paragraphs in Introduction:

However, the indirect losses due to supply chain disruptions have not been fully analyzed (Wang et al., 2021; Xia et al., 2019), as previous literature has either devoted insufficient discussion to the indirect effects by only reporting the total/net effects, or ignore the amplifying effect of the global trade system on direct losses (García-León et al., 2021; Knittel et al., 2020; Takakura et al., 2018). As climate change will make the impacts of heat stress worse over time, developing methodologies that allow comprehensive quantifications of both the direct and indirect impacts of heat stress on human systems can help policymakers to develop more effective climate change mitigation and adaptation policies.

We are also very grateful for your recognition of the innovative nature of our research. The analysis of indirect losses and their transmission mechanisms is unprecedented. And the increased country and sectoral resolution allows us to focus on more vulnerable countries and sectors, which is important for the accuracy of the calculations and the targeting of policy recommendations. We will discuss this later in your other comments.

C. Data & methodology: validity of approach, quality of data, quality of presentation

The climate, health and labor productivity modules are state-of-the art or go beyond that (by e.g. considering the rate of air-conditioning dependent on level of regional development in the different SSPs) and no changes are needed here. However, I do have concerns about the economic module:

1) I am concerned when key assumptions of a modeling class are relaxed to mimic behavior that is standard in other approaches. In the current model, this is the case when the authors introduce substitutability of products within a sector so that these products can be sourced from other regions in case of bottlenecks. This trade substitutability is indeed a key feature in CGE models (mostly based on the Armington, 1969, assumption of limited substitutability; occasionally the Melitz approach) but not in MRIO models to which the ARIO model belongs. Why is this substitution allowed in trade patterns but not in production where Leontief functions are used? Usually, fixed input structures and fixed trade patterns are argued to represent short term effects (within year effects) whereas substitutability in production processes and trade is argued to represent adjustment in the longer term (new equilibrium after several years). I find a mixture of trade substitutability and fixed production processes inconsistent.

Based on the available information, including the supplementary file, I also cannot assess which data is used to introduce this substitutability. Trade elasticities from GTAP or some other sources?

RE: Thank you for your advice and we fully understand your concerns. Firstly, this study assumes that the same products from different regions are substitutable. This is an improvement on current state-of-art ARIO model, as (Koks et al., 2016) found by comparing multiple hazard assessment models that traditional ARIO overestimates losses because it does not allow for substitution. For example, for a firm that extracts spices from bananas it does not make much of a difference if the bananas are sourced from the Philippines or Thailand. With symmetrical information and low transaction costs, goods can be fully circulated within a one-year time step in our model setting. Due to the simplicity of the ARIO model, we have not introduced trade elasticities parameters that would require extensive data calibration, which would be unrealistic for a model in sector 65, area 141. We are working on trade substitutability modelling in IO model and also hope to collaborate with knowledgeable people in the ARIO modelling field to accomplish methodological innovations

and publish papers to deepen the field's understanding of disaster modelling.

Secondly, we still have chosen a Leontief production function with a fixed ratio of capital to labour, considering the nature of disaster shocks. Heat stress usually lasts a few days or one to two weeks. It is almost impossible for the production function to change in the short term. When a heat wave hits, for example, companies are more likely to reduce working hours than to immediately introduce machines to replace worker labour. Furthermore, unlike substitution of the same product from different region (e.g. timber from the USA or Canada), substitution of capital and labour relies on production technology because they are not essentially same thing that can be substituted. This is a long-term strategy. Only in the case of significant technological progress (significant reduction in the cost of capital), or where it is certain that heat stress will become increasingly severe in the future (significant increase in the cost of labor), will producers systematically engage in capital-labour substitution in the light of cost considerations. Otherwise we will underestimate losses like CGE simulation as it allows extreme flexibility for no-cost substitution (Koks et al., 2016; Rose, 2004). This paper therefore assesses the economic impact of climate change under the assumption that there are no significant changes in production technology, which is also consistent with the constant technology assumption in intermediate products consumption. To make this clear, we add a sentence in line 535 to clarify our specific contribution, that “We improve the ARIO model in two ways. First, we allow the substitutability of products from the same sector sourced from different regions. Second, in our model, clients will choose their suppliers across regions based on their capacity. These two improvements contribute to a more realistic representation of bottlenecks along global supply chains, as traditional ARIO overestimates losses because it does not allow for substitution.”

Finally, methodological innovations are somewhat beyond the scope of this paper. The central contribution of this study is, on the one hand, a certain improvement in the substitution of the ARIO model. On the other hand, the integration of the ARIO model with other natural and social science models.

In order to ensure maximum robustness and scientific validity of the results, we have summed the sectors to 10 based on GTAP data and run a CGE simulation with production and trade elasticities, for comparison purpose. We have placed those specific results in the SI uncertainty analysis section. We found our results are in line with most of CGE simulations and relevant results in typical

literature (e.g. (Knittel et al., 2020; Orlov et al., 2020; Takakura et al., 2018)). The simulations we ran for CGE (2.2% in 2050) did result in about 10% lower global GDP losses than ARIO (2.5% in 2050). **But the core conclusion of this study, that the global supply chain amplification effect leads to a non-linear increase in economic losses from heat stress, is robust to a large number of rigorous tests.** As shown in Supplementary Figure 8, the indirect losses of the 10-region and 10-sector CGE model show a non-linear upward trend.

2) I also have concerns about the rationing approach, the second innovation of the paper. Wouldn't part of rationing be the result of increases in price or at least be accompanied by price increases? For instance, the Thailand flood 2011 led to substantial increases in prices of electronic components, in addition to disrupted supply.

RE: Firstly, we acknowledge that price factors are not quantified in our model. Price is not a parameter of the ARIO model. This is the next step in the progress of the ARIO model. But our core contribution in this article is to link climate-health-economy models, not to optimise a particular one. We are working on methodological innovations to investigate the use of the agent base model to optimise the IO model. However, this is beyond the scope of this paper.

Secondly, price is not a significant parameter of the model in this study. Unlike a flood, which destroys a large amount of capacity in a certain region in a short period of time, causing severe supply shortages and price increases. Heat stress temporarily affects the productivity of labour in a given area. A few days or weeks later cooler weather arrives and labour productivity can return immediately. No significant price fluctuations will occur in this case. To make this clear, we add a sentence in line 406 to clarify limitation of current ARIO model in terms of pricing, that "Given the unpredictable feature for global socioeconomic systems, this study applied the impact of future climate risks to current socioeconomic settings to analyse potential impact as most of literatures have adopted previously (e.g. (García-León et al., 2021; Parsons et al., 2021)). However, this ignores the differences of trade openness and globalization among SSP narratives, as well as the dynamic factors of technology and price."

3) Regarding the implementation of the labor productivity shock, more detail is needed on how the three types of labor are distributed to the sectors. Do you assume that e.g. all work in construction

is heavy outdoor work? Or is there a combination depending on the shares of the three types of work?

RE: Indeed, we appreciate the urgent need for micro-survey data on occupational segments under each sector. We eventually used the sectoral single labour type calculations, considering both data availability and Lyontief function properties. Firstly, in all previous studies of heat-related labour losses, there is no data or cases to support the splitting of labour types in a particular sector (Cai et al., 2021; Chavaillaz et al., 2019; García-León et al., 2021; Kjellstrom et al., 2018; Knittel et al., 2020; Orlov et al., 2020; Parsons et al., 2021; Watts et al., 2019b, 2021; Yin et al., 2021; Zhao et al., 2021). In the construction industry, for example, jobs such as steelworkers and welders are very labour intensive and are usually outdoors. However, renovation workers such as painters may work indoors and are less labour intensive. These proportional data available for scientific research in this field is almost blank in literatures. Secondly, because this study uses the Leontief production function, sectoral losses are determined by the job with the highest losses in that sector. We acknowledge that this is the limitation of this study in line 402, but this type of micro-level study would require somewhat billion-dollar funding to support. We will keep our eyes open for such research, and willing to collaborate on whatever possible with any relevant institutions.

4) If I understand the model setup correctly, it is assumed that capital is affected similarly as labor by the productivity shock due to the Leontief production function. But in contrast to flooding or a hurricane where capital is destroyed, in case of a heat-related labor productivity shock part of capital becomes idle (machinery is still there, but might not be used), so this is a temporary, not a permanent unavailability. How is this difference accounted for?

RE: You are right to understand. Idle capital is a component of economic loss, considering that the classical literature agrees that capital and labour are fixed in proportion (Li et al., 2013; Steenge & Bočkarjova, 2007) under shocks such as a heatwave. Since we measure the decline in production capacity due to climate change, this restricted capital is also considered part of the loss as they are not put into production as they should have been. Idle capital due to labour shortages is a temporary loss. Labour productivity will recover and the corresponding capital will be put into production and will no longer be idle when heatwave ends. In the case of a typhoon flood, for

example, the destroyed capital would take time, labour and materials to rebuild in ARIO model (Hallegatte, 2008; Mendoza-Tinoco et al., 2020). In this case the reduction in production capacity depends on the maximum of capital and labour constraints. Our model makes a clear distinction between the loss of capital in these two scenarios. To make this point clearer, we have rephrased the line 606, as “In contrast to flooding or a hurricane where capital is destroyed, part of capital becomes idle (machinery is still there, but might not be used) in case of a heat-related labor productivity shock. Idle capital is a component of economic loss, considering that the classical literature agrees that capital and labour are fixed in proportion (Li et al., 2013; Steenge & Bočkarjova, 2007) under shocks such as a heatwave. This is a temporary, not a permanent unavailability. If there is no heatwave the following year, this labour and capital can be restored to production immediately. While the capital destroyed by floods or hurricanes took time, labour and materials to rebuild.”

5) How are the different SSPs reflected in the economic module? It seems like the economic structure is entirely static to the base year dataset. E.g., in Figures 3-5, is the difference between 2060s and 2080s entirely owed to different changes in climate and changes in population? But SSP narratives also differentiate in terms of globalization and trade openness. At the very least, it should be clarified that this dimension is missing from the analysis.

RE: Thank you for your valuable corrections! Indeed the change from figure 3 to figure 5 is only due to climate change and population dynamics as all SSPs constructed for. Given the unpredictable feature for global socioeconomic systems, this study fixes the current economic and trade structure, and to some extent, provides a clearer picture of the impacts of climate change so that governments and industry can develop targeted adaptation strategies. In other words, we applied the impact of future climate risks to current socioeconomic settings to analyse potential impact as most of literatures have adopted previously (e.g. (García-León et al., 2021; Parsons et al., 2021)). We have clarified this point in line 407. As your next suggestion says, we can isolate the effects of trade dynamics, population dynamics and climate dynamics separately. This is a new perspective we are digging. We hope to work with all interested parties to get this approach completed, and possibly in another Nature paper.

6) Figure 1: While it seems to become established practice since AR6 to report fixed combinations

of RCP and SSP scenarios, it would be interesting to see how much of this non-linear increase in GDP loss is driven by change in climate and by change in population.

RE: That's a very interesting idea, thanks for the suggestion! We could fix the population grid using the base period and just let the climate data change over time, thus separating the two effects you are talking about. Please see the following paragraphs in SI:

6. Disaggregated impacts of climate change and population dynamics

By fixing the population grid (number and distribution) in current situation in year 2020, we decompose the respective contributions of climate change and population dynamics to global economic losses. As population size is only used as a weighted indicator in our model, only the spatial distribution of the population affects the assessment results, not the population size. We found that about 97% of global economic losses in this study are explained by climate change. Dynamic population scenario shows a slight increase of 0.11% in global economic losses compared to the fixed population scenario. The effects of spatial demographic change make a relatively weak contribution in individual countries. In Nepal, Pakistan and Niger, for example, population dynamics increase direct losses by an additional 0.7%, 0.4% and 0.4% of country's GDP respectively. This suggests that the spatial expansion of future populations in Nepal and Pakistan (i.e. the process of urbanisation) is sensitive to climate change risk and exposes greater proportion of the population to heat stress. Population dynamics in Qatar and UAE have reduced national GDP losses by 0.3% and 0.2%. This suggests that Qatar and the UAE are at low climate risk for future urbanisation or spatial distribution of population.

The population distribution in the existing SSP scenario does not consider preferences for future climate livability under the effects of climate change and climate-induced migration. Therefore future studies of urban expansion or population growth should pay more attention to climatic factors. Spatial population dynamics have the potential to reduce global climate risks if climate migration, risk aversion and other factors are fully considered.

7) I find the approach in Figure 3 very interesting as it illustrates patterns of transmission depending on a country's exposure and vulnerability and it does not focus on the usual large economies (US, EU, China). But I strongly recommend to use statistical analysis to identify these clusters and to be

careful when stating which countries are similar in their response patterns (e.g. there is a considerable difference in exposure to transboundary risks across European countries, see Benzie et al., 2019). Moreover, it is vital mention that the indirect loss depends strongly on the underlying data (GTAP vs EORA) and base year (2014 or other). Please consider conducting sensitivity checks similar to Willner et al. (2016).

RE: Thank you for your very insightful points. We did find that countries within the same region may face very different patterns of losses and were extra careful in selecting the countries. We selected a few representative countries from Figure 3 in the main text by means of a decision tree approach. For example, regions are divided into two categories according to whether indirect losses or direct losses dominate, etc. However, as you say, decision tree selection is subjective, relying heavily on the researchers themselves to draw up patterns. We supplemented the clustering method with an unsupervised classification (k-means) and obtained similar clustering results. We have clustered the indicators according to economic losses under different scenarios, as well as the sectoral categories with the highest losses. Also your point about using different versions of the base period database is very constructive. We have carried out robustness tests using the GTAP9 database as you you said. Please see the paragraph, Supplementary Figure 11 and ED fig.4:

4. Loss patterns in representative countries

Individual countries have different patterns of loss and vulnerability in the context of climate change. Based on the losses under the SSP119 scenario, the growth rate of losses over time and scenarios, and the sectors with the highest losses, we have conducted k-means (for continuous variables, e.g. percentage loss), EM algorithm (for ranking data, e.g. ranking of sectors with the highest percentage of losses) and k-Prototypes (for mixed variables, e.g. percentage loss, vulnerable sector category) clustering analysis of 141 regions. The k-means algorithm after the sector loss pattern quantification process performs best with minimized total within-cluster variation. GTAP 141 sectors were defined as 11 categories (1. GrainsCrops, 2. MeatLstk, 3. Extraction etc., based on official GTAP aggregation). National sectoral loss patterns are quantified as the number of occurrences in the top ten sectors of losses (Supplementary Table3). We use Elbow method to calculate the total within-cluster sum of square, and the appropriate number of

clustering categories was determined to be 4. The clustering results are shown in Supplementary Figure 11. According to the cluster centres, Cluster 1 is highly vulnerable to extreme heatwaves and health and labor productivity loss, with significant losses in sectors such as construction and agriculture under all the SSP119, SSP245 and SSP585 scenarios. Cluster 2 and suffers considerable both direct and indirect economic losses in SSP245 and SSP585 scenarios, with construction, agriculture, food processing and non-metallic manufacturing as the main sectors of loss. Cluster 3 is scenario change sensitive and these countries suffer almost no losses under the SSP119 scenario, but losses rise at a high rate under the SSP585 scenario. The main loss sector is metals and manufacturing. Cluster 4 is relatively resilient, with moderate levels of losses in all scenarios.

8) While I consider backward and forward propagation a very important topic to look into, clarification is needed on the selection of the two cases in Figures 4 and 5. I am not questioning that they have an important contribution to domestic GDP and are reliant on international supply chains, but many other examples could be picked. Please motivate your choice against other potential alternatives.

RE: We run the model of supply chain of high loss industries in all countries and can show any industry you want in the maintext. Due to space constraints we can only place more representative chains in the SI. We have selected these two industries following the principles of typicality and developing country first. In terms of typicality, food processing in India and tourism in the Dominican Republic represent two types of upstream and downstream constraints. India has the highest number of people living in extreme poverty in the world. The food processing industry in India is closely linked to the fight against poverty and hunger. Dominica, on the other hand, is a mono-industry country that relies heavily on tourism. Both the two industries occupy a very important position in national GDP and global markets. They are therefore typical industries.

In terms of developing country first, developing countries are particularly important in climate risk governance, but receive very little attention. In the second section of the main results, we discuss the significant inequality in the growth of climate loss between developed and developing countries. However, developing countries have almost never been present in previous studies. This study is based on the ARIO model, which does not require parameter correction for the long time scales of the CGE model and allows for greater disaggregation of regional and industry scales. In

contrast, traditional CGE models can only assess a small number of large economies and three to five sectors that are otherwise almost impossible to solve for. This paper therefore focuses more on developing countries, and typical sectors.

Minor points:

1) Figure 1: It would be good to include such a decomposition in the Supplementary Material. And please add a row of maps for SSP2-4.5.

RE: Thank you for your suggestion, it has been added to the additional information. But Figure 1 in the main text is a bit too informative, would the map for the SSP245 scenario be clearer in the SI? We have put Figure 1 of the 16-gallery in the SI, and if you think it would be better in the maintext, we would be happy to correct it further.

2) In Figure 2, please clarify what the error bars represent (variability across years?)

RE: Yes, for example the bar chart shows the percentage of losses in 2060 and the error bar shows the maximum and minimum values for the decade 2055 to 2065. To make it clear, we have added captions to the figure "The error bar shows the upper and lower limits of the decade"

3) I suggest to increase consistency between Figure 2 and Figures 4 and 5: either look into differences across time slices for one scenario (SSP5-8.5) or into differences across scenarios for one time slice (SSP5-8.5 vs SSP1-2.6 for 2060). Again, discuss the caveat that the socio-economic structure is fixed at the base year.

RE: Thank you for your advice. We have discussed the caveats on two fronts. On the one hand, we discussed the limitations of a constant industrial structure in the Discussion section. Uncertainty about the future dynamics of trade structures is high and this paper only provides macro insights into the impact of climate change on global value chains. On the other hand, we have added robustness tests for replacing the base period database (using GTAP9 2011's trade structure as you suggest), verifying that the core findings of this paper remain robust to changes in the trade structure.

4) In all figures, please clarify that the value added % numbers are % of total value added, ie GDP, and not sectoral value added.

RE: Thank you for your advice. We have clarified in the markup of all the diagrams.

D. Appropriate use of statistics and treatment of uncertainties

While the description of the countries in the three clusters in Figure 3 is interesting and the cases are arguably selected as representative, please consider identifying the clusters by means of statistical analysis. Given the number of sectors, regions and variables, this should be feasible.

Similarly for Figures 4 and 5: Please use some traceable (statistical?) approach for the selection of the two cases.

RE: Thank you for your suggestion. K-means clustering analysis has been implemented, please see the following paragraphs and Supplementary Figure 11 in SI section 4: Loss patterns in representative countries

E. Conclusions: robustness, validity, reliability

In the discussion section, please add that indirect effects can not only work as risk amplifier but also as a risk reducer by means of trade rerouting or diversification as well as building up stocks, the latter point being discussed in the sensitivity analysis in the Supplementary Material.

RE: Thank you for your advice. We fully explain that indirect effects can not only work as risk amplifier but also as a risk reducer by means of trade rerouting or diversification as well as building up stocks in our robustness tests. Please see the following paragraphs:

In SI Robustness tests:

In numerous experiments we have also found that indirect effects can not only work as risk amplifier but also as a risk reducer by means of trade rerouting or diversification as well as building up stocks. This reducer effect is particularly pronounced for countries with high risk in particular supply chains. For example, the indirect loss to Kyrgyzstan when trade substitution is not allowed is as high as 4.26% of GDP in 2060 under SSP85 scenario. When trade rerouting is allowed, the loss

is reduced to 1.24%. At the global scale, trade rerouting and diversification in supply chains can reduce GDP losses by 10% to 15%. Building up reasonable and adequate stocks can delay and slightly reduce the impact of disaster shocks, reducing indirect losses by about 5% to 10% on a global scale.

F. Suggested improvements: experiments, data for possible revision

One of the main socio-economic uncertainties is sectoral composition and trade structure. Given the focus of the paper, please consider different base years (e.g. GTAP 9 and earlier versions which are now included as part of the same license) and potentially alternative datasets (e.g. EORA instead of GTAP) to test for the robustness of results.

If you decide to keep substitutability in trade, expand the sensitivity checks (in Section 2 of the Supplementary Material) with regard to trade elasticities.

RE: Thank you for your very valuable advice. We have added database tests for GTAP9, including robustness tests for time series of global total indirect losses, vulnerability patterns at national and sectoral scales. With regard to trade elasticities, we acknowledge the absence of this parameter in the ARIO model, which is a limitation and a direction for the next improvement of the model. However, we have still tried to take a semi-quantitative approach to robustness testing of trade elasticities. That is, it is assumed that trade substitution is carried out with or without a certain upper bound. Please see the following paragraphs in SI:

1.3 Uncertainty of indirect loss

Uncertainty about indirect losses arising from (1) production side, which includes the base period production and trade structure, inventories, excess production capacity and the elasticity of trade substitution of products across countries; and (2) demand side, which includes the ability of the final demand account to adjust when there is a shortfall in demand or supply. While the ARIO model is widely used and work well for a single-country/single-region analyses, the substitutability of products in a multi-country scenario requires further discussion to verify robustness. To address the uncertainty in the structure of production and trade, we have used different years and versions of the input-output database for comparison tests (see Extended Data Fig. 4). For parameters such as the maximum stock ratio and excess production capacity, we repeated the experiment several times within the range of possible values from previous studies. For trade substitutability, upper

and lower bounds of perfect substitution and non-substitution (traditional static IO model) were used.

Estimates are displayed as 10-year averages for the year 2060, using the GTAP 2011 (a), GTAP 2014 (b) and EMERGING 2019 (c) databases separately. The colours of the bars represent GDP per capita from low to high. (d), indirect losses under different benchmark trade structures in each region. The horizontal axis measures the indirect losses as a percentage of GDP using the GTAP2014 trade structure and the vertical axis measures the indirect losses as a percentage of GDP using the GTAP2011 trade structure.

Supplementary Figure 6 Comparison of the impact of using different trade structures on the assessment of indirect losses using GTAP2011 and GTAP2014. The horizontal axis is the indirect losses as a percentage of GDP using the GTAP2014 trade structure and the vertical axis is the result of using the GTAP2011 trade structure.

Globally, the total amount of indirect losses remains robust to changes in the structure of the data over the base period. The results of the global scale loss assessment differ by less than 5% in 2060. The vast majority of countries are distributed around the $y=x$ line, which implies a consistent assessment across different trade structures.

Regionally, for a small number of countries, indirect loss assessments can show larger differences. By comparison, we find that when using GTAP 2014 data for the base period, indirect economic losses in East and Southeast Asian countries, such as Singapore, Korea, Japan, are amplified (see Extended Data Fig. 4 and Supplementary Figure6). This can be explained by the fact that, in GTAP2014, those countries have closer economic ties with climate-sensitive markets, including Malaysia, China, India and Vietnam. For instance, trade between Singapore and emerging economies like China and Vietnam had increased substantially from 2010 to 2014. According to the Singapore Department of Statistics (<https://www.singstat.gov.sg/>) and the United Nations Commodity Trade Statistics Database (<https://comtrade.un.org>), China became the largest trading partner of Singapore in 2014, up from 4th place in 2011, whereas Vietnam rose to the 13th largest partner in 2014, from the 20th place in 2011. Conversely, Singapore's total trade share with the EU and the USA decreased slightly over the same period. Similarly, Japan, Korea and Myanmar developed closer trade relationships with emerging markets such as China, India, and Vietnam.

The assessment of the different trade structures brings important insights, into the need for both developed and emerging developing countries, which will be increasingly involved in international trade in the future, to carefully consider the supply chain risks posed by climate.

We elaborate in more detail the uncertainty parameter intervals for the three main modules and

do a Monte Carlo analysis, including simulation of economic loss dynamic for 10000 periods. We have also summarized the results of previous assessments of different models based on CMIP5 data and similar RCP scenarios for comparison. The results are shown in Extended Data Fig. 5.

Supplementary Figure 7 a-d is the high adaptation scenario, with lowest economic losses. The scenario parameters for the high adaptation scenario losses are set as follows. The country provides producers with a six-month stock of raw materials through the mobilization of national strategic reserves etc (except for dairy and beverage products industry, whose stock was ranged for zero to one month at most). When there is a shortage of available labour facing heatwaves, up to 25% of excess productivity is provided based on the highest values from previous studies¹⁷. Products can be fully mobile within global trade networks and national products can be substituted for each other.

Supplementary Figure 7 e-h is the moderate adaptation scenario, and parameters such as inventory and excess capacity are used in line with the facts and are more common in previous studies¹⁷. The producer stocks one quarter's inventory of raw materials. When there is a shortage of available labour, a maximum of 10% excess productivity is provided. Products are partially mobile within global trade networks but need time to adjust. The allocation strategies of the countries in the model will be gradually shifted to unaffected countries (rather than in one step). Products can be substituted for each country except for construction.

Supplementary Figure 7 i-l is the low adaptation scenario, with highest economic losses. The parameters of the low adaptation scenario are set as follows. Producer stocks one month of raw material inventory. When the available labour force is insufficient, excess productivity cannot be provided through additional adaptation measures. The substitutability of products between countries is low due to information lags and trade barriers. When country A experiences a shortage in the supply of upstream intermediate products, it is unable to obtain substitute products from other countries. This scenario setting is close to a traditional static IO, which would overestimate economic losses and is the upper limit of the assessment. In the figure for the Monte Carlo analysis (Fig.4), we do not show the results under this one setting, as their curves are clearly biased towards the other type of clustering.

As shown in Extended Data Fig. 5, Supplementary Figure 7 and Supplementary Figure 8, firstly, the core finding of this study, that the global supply chain amplification effect leads to a non-linear increase in economic losses from heat stress, is robust to a large number of rigorous tests. The difference is only that the point at which the exponential growth occurs changes with the adaptive setting. Supplementary Figure 7 e-h is the baseline scenario similar to the result in the main text,

where parameters such as inventories and excess production capacity are used in line with the facts and are more common in previous studies. In the high adaptation scenario (Supplementary Figure 7 a-d), economic losses increase at a high rate from 2050 to 2060. This is because each country is assumed to have sufficient production stocks for vulnerable industries to maintain production for a short period of time even in the event of upstream supply disruptions. And when extreme heat is frequent, new adaptive technologies can be generated and widely used in time to offset the health and labour productivity losses of heat stress to some extent. In contrast, when the adaptation parameter is set at its lowest setting (Supplementary Figure 7 i-l), economic losses will rapidly increase in very near future. As for the signs and the signals (e.g. which countries and regions most affected, which sectors are most affected), country-scale vulnerability is very robust to changes in parameters other than trade structure. Changes in the structure of trade, such as the year of the MRIO table for the base period, can have a significant impact on specific regions whose position in GVC has change significantly between reference years (as shown Extended Data Fig. 4). Under the 2011 trade structure, for example, economic losses in Africa could hardly be transmitted to Asia. However, in 2014, with East Asian Southeast Asian countries such as China and India trading closely with Africa, the assessment of indirect losses in Southeast Asia improved significantly. This also illustrates how well our model captures the transmission of economic risks between trading partners.

We also employed a dynamic Computable General Equilibrium (CGE) model for a parallel assessment, as part of the robustness check of the ARIO results. Specifically, the CGE model we utilized is a G-RDEM with 10 regions and 10 sectors. G-RDEM is a well-designed CGE tool for long-term counterfactual analysis and economic baseline generation based on provided gross domestic product (GDP) and population projections. It has undergone various enhancements tailored for generating long-term scenarios and simulations¹⁸. It includes an implicitly directly additive demand system with non-linear Engel curves, incorporates debt accumulation from foreign savings, introduces sector-specific productivity changes, endogenously determines aggregate saving rates, and incorporates time-varying cost shares for value added and individual intermediates. The parameters for these relationships are estimated through econometric methods using the most recent available data or derived from published research.

We employed this model to assess the impacts of future heatwaves in a manner akin to the assessment performed by the ARIO model in this study. The evaluation outcomes from the CGE model are depicted in Supplementary Figure 8. As evident from the figure, the two models' evaluation results exhibit consistent trends and, to a certain extent, align in magnitude. While the ARIO model does not account for changes in future economic structure, the dynamic CGE model considers such variations. The comparison of these outcomes demonstrates, from one perspective, that "although neglecting changes in future economic structure could lead to some distortion in

ARIO model assessment outcomes, this weakness in ARIO does not significantly impact the evaluation results." Two reasonable explanations are as follows: (1) Although the economic structure may evolve in the future, it does so based on the existing economic framework. While disregarding this change could introduce bias in the results, this change is relatively minor compared to the current structure. Thus, capturing the current structure is more crucial. (2) The most significant changes in the future occur in regions with relatively smaller current economic scales, such as Africa, which constitutes a limited portion of the global economic volume. These changes have a marginal effect on the overall global evaluation results. Both these aspects contribute to the rationale behind the alignment in magnitude between considering structural changes (dynamic CGE) and disregarding structural changes (ARIO). But the core conclusion of this study, that the global supply chain amplification effect leads to a non-linear increase in economic losses from heat stress, is robust to a large number of rigorous tests. As shown in Supplementary Figure 8, the indirect losses of the 10-region and 10-sector CGE model show a non-linear upward trend.

Secondly, the results of our assessment are slightly higher than CGE modelling results of previous studies and within the confidence interval. The simulations we ran for dynamic CGE ($2.1\% \pm 0.4\%$ in 2060) result in about 10% to 20% lower global GDP losses than ARIO ($2.5\% \pm 0.7\%$ in 2060). We are more confident about the results of this research in two respects. On the one hand, we are using the latest, multi-model CMIP6 meteorological data and SSP socio-economic dynamics data. This is a significant improvement over the results of most previous assessments based on CMIP5 and RCP single-line scenario settings. On the other hand, Koks et al.¹⁹ found that CGE-based models tend to underestimate economic losses, while static IO-based models tend to overestimate economic losses. The simulations we ran under dynamic CGE model did result in about 10% to 20% lower global GDP losses than ARIO model. The ARIO model we use improves the IO model by introducing adaptations that are closer to the actual results. More importantly, a growing number of studies have found that sector aggregating in the CGE and IO models can introduce significant errors (often underestimations) into the indirect loss assessment. Even when based on the same data, the results of sector/area summation can vary several times from those calculated before the summation. For the first time, our model assesses global economic losses at a scale of 141 regions, 65 sectors, which fixes the bias introduced by the aggregation of the world into 10 or so regions and sectors.

In summary, we find that the strong adaptation hypothesis (quantified in the model in terms of the parameters associated with adaptation or substitution) can significantly reduce economic losses in the short term, under low emission scenarios. However, adaptation strategies for long-term climate change stress, especially under high emission scenarios, are not effective in reducing economic losses on long time scales (beyond 2060).

In numerous experiments we have also found that indirect effects can not only work as risk amplifier but also as a risk reducer by means of trade rerouting or diversification as well as building up stocks. This reducer effect is particularly pronounced for countries with high risk in particular supply chains. For example, the indirect loss to Kyrgyzstan when trade substitution is not allowed is as high as 4.26% of GDP in 2060 under SSP85 scenario. When trade rerouting is allowed, the loss is reduced to 1.24%. At the global scale, trade rerouting and diversification in supply chains can reduce GDP losses by 10% to 15%. Building up reasonable and adequate stocks can delay and slightly reduce the impact of disaster shocks, reducing indirect losses by about 5% to 10% on a global scale.

G. References: appropriate credit to previous work?

See my comments on point B above.

H. Clarity and context: lucidity of abstract/summary, appropriateness of abstract, introduction and conclusions

In the abstract, I am not sure about the sentence “Severely affected industries such as farming and construction in Tanzania are expected to lose 6.8% to 7.7% of their value added, while the food processing industry in Italy and India is expected to lose 4.9% to 6.5% of their value-added.” – Does “their value added” mean sectoral value added or economy-wide value added? In Figure 2 in the main text, my reading was that it was always % of economy-wide value added (GDP), not sectoral. Please clarify!

RE: Thank you for the correction. The sectoral losses we have mentioned are as a percentage of the value added of the sector. So “their value added” mean sectoral value added, and we have clarified this in the main text.

References cited:

Benzie, M., Carter, T. R., Carlsen, H., & Taylor, R. (2019). Cross-border climate change impacts: Implications for the European Union. *Regional Environmental Change*, 19(3), 763– 776. <https://doi.org/10.1007/s10113-018-1436-1>

Carter, T. R., Benzie, M., Campiglio, E., Carlsen, H., Fronzek, S., Hildén, M., Reyer, C. P. O., & West, C. (2021). A conceptual framework for cross-border impacts of climate change. *Global Environmental Change*, 69. Scopus. <https://doi.org/10.1016/j.gloenvcha.2021.102307>

García-León, D., Casanueva, A., Standardi, G., Burgstall, A., Flouris, A. D., & Nybo, L. (2021). Current and projected regional economic impacts of heatwaves in Europe. *Nature Communications*, 12(1), 5807. <https://doi.org/10.1038/s41467-021-26050-z>

Knittel, N., Jury, M. W., Bednar-Friedl, B., Bachner, G., & Steiner, A. K. (2020). A global analysis of heat-related labour productivity losses under climate change—Implications for Germany’s foreign trade. *Climatic Change*, 160(2), 251–269. Scopus. <https://doi.org/10.1007/s10584-020-02661-1>

Kulmer, V., Jury, M., Wong, S., & Kortschak, D. (2020). Global resource consumption effects of borderless climate change: EU’s indirect vulnerability. *Environmental and Sustainability Indicators*, 8, 100071. <https://doi.org/10.1016/j.indic.2020.100071>

Matsumoto, K. (2019). Climate change impacts on socioeconomic activities through labor productivity changes considering interactions between socioeconomic and climate systems. *Journal of Cleaner Production*, 216, 528–541. <https://doi.org/10.1016/j.jclepro.2018.12.127>

Orlov, A., Sillmann, J., Aunan, K., Kjellstrom, T., & Aaheim, A. (2020). Economic costs of heat-induced reductions in worker productivity due to global warming. *Global Environmental Change*, 63, 102087. <https://doi.org/10.1016/j.gloenvcha.2020.102087>

Takakura, J., Fujimori, S., Takahashi, K., Hijioka, Y., Hasegawa, T., Honda, Y., & Masui, T. (2017). Cost of preventing workplace heat-related illness through worker breaks and the benefit of climate-change mitigation. *Environmental Research Letters*, 12(6), 064010. <https://doi.org/10.1088/1748-9326/aa72cc>

Wenz, L., & Levermann, A. (2016). Enhanced economic connectivity to foster heat stress-related losses. *Science Advances*, 2(6), e1501026. <https://doi.org/10.1126/sciadv.1501026>

Willner, S. N., Otto, C., & Levermann, A. (2018). Global economic response to river floods. *Nature Climate Change*, 8(7), 594–598. <https://doi.org/10.1038/s41558-018-0173-2>

References cited in replies:

Aguiar, A., Chepeliev, M., Corong, E., & Mcdougall, R. (2019). The GTAP Data Base: Version 10. *Journal of Global Economic Analysis*, 4(1), 27.

Ara, K. (1959). The Aggregation Problem in Input-Output Analysis. *Econometrica*, 27(2), 257–262. <https://doi.org/10.2307/1909446>

- Bröde, P., Fiala, D., Lemke, B., & Kjellstrom, T. (2018). Estimated work ability in warm outdoor environments depends on the chosen heat stress assessment metric. *International Journal of Biometeorology*, 62(3), 331–345. <https://doi.org/10/gc89nh>
- Cai, W., Zhang, C., Suen, H. P., Ai, S., Bai, Y., Bao, J., Chen, B., Cheng, L., Cui, X., Dai, H., Di, Q., Dong, W., Dou, D., Fan, W., Fan, X., Gao, T., Geng, Y., Guan, D., Guo, Y., ... Gong, P. (2021). The 2020 China report of the Lancet Countdown on health and climate change. *The Lancet Public Health*, 6(1), e64–e81. <https://doi.org/10/ghn8t5>
- Chavaillaz, Y., Roy, P., Partanen, A.-I., Da Silva, L., Bresson, É., Mengis, N., Chaumont, D., & Matthews, H. D. Fei, J. C.-H. (1956). A Fundamental Theorem for the Aggregation Problem of Input-Output Analysis. *Econometrica*, 24(4), 400–412. <https://doi.org/10.2307/1905491>
- García-León, D., Casanueva, A., Standardi, G., Burgstall, A., Flouris, A. D., & Nybo, L. (2021). Current and projected regional economic impacts of heatwaves in Europe. *Nature Communications*, 12(1), Article 1. <https://doi.org/10.1038/s41467-021-26050-z>
- Guan, D., Wang, D., Hallegatte, S., Davis, S. J., Huo, J., Li, S., Bai, Y., Lei, T., Xue, Q., Coffman, D., Cheng, D., Chen, P., Liang, X., Xu, B., Lu, X., Wang, S., Hubacek, K., & Gong, P. (2020). Global supply-chain effects of COVID-19 control measures. *Nature Human Behaviour*, 4(6), 577–587. <https://doi.org/10.1038/s41562-020-0896-8>
- Guo, Y., Gasparrini, A., Armstrong, B. G., Tawatsupa, B., Tobias, A., Lavigne, E., Coelho, M. de S. Z. S., Pan, X., Kim, H., Hashizume, M., Honda, Y., Guo, Y.-L. L., Wu, C.-F., Zanobetti, A., Schwartz, J. D., Bell, M. L., Scortichini, M., Michelozzi, P., Punnasiri, K., ... Tong, S. (2017). Heat Wave and Mortality: A Multicountry, Multicommunity Study. *Environmental Health Perspectives*, 125(8), 087006. <https://doi.org/10/gbwzpx>

Hallegatte, S. (2008). An Adaptive Regional Input-Output Model and its Application to the Assessment of the Economic Cost of Katrina. *Risk Analysis*, 28(3), 779–799. <https://doi.org/10.1111/j.1539-6924.2008.01046.x>

Inoue, H., & Todo, Y. (2019). Firm-level propagation of shocks through supply-chain networks. *Nature Sustainability*, 2(9), Article 9. <https://doi.org/10.1038/s41893-019-0351-x> Kjellstrom, T., Freyberg, C., Lemke, B., Otto, M., & Briggs, D. (2018). Estimating population heat exposure and impacts on working people in conjunction with climate change. *International Journal of Biometeorology*, 62(3), 291–306. <https://doi.org/10/gkzwtx>

Knittel, N., Jury, M. W., Bednar-Friedl, B., Bachner, G., & Steiner, A. K. (2020). A global analysis of heat-related labour productivity losses under climate change—Implications for Germany’s foreign trade. *Climatic Change*, 160(2), 251–269. <https://doi.org/10/gjkjrq>

Koks, E. E., Carrera, L., Jonkeren, O., Aerts, J. C. J. H., Husby, T. G., Thissen, M., Standardi, G., & Mysiak, J. (2016). Regional disaster impact analysis: Comparing input–output and computable general equilibrium models. *Natural Hazards and Earth System Sciences*, 16(8), 1911–1924. <https://doi.org/10.5194/nhess-16-1911-2016>

Lenzen, M. (2019). Aggregating input–output systems with minimum error. *Economic Systems Research*, 31(4), 594–616. <https://doi.org/10.1080/09535314.2019.1609911>

Li, J., Crawford-Brown, D., Syddall, M., & Guan, D. (2013). Modeling imbalanced economic recovery following a natural disaster using input-output analysis. *Risk Analysis*, 33(10), 1908–1923. <https://doi.org/10.1111/risa.12040>

Markhvida, M., Walsh, B., Hallegatte, S., & Baker, J. (2020). Quantification of disaster impacts through household well-being losses. *Nature Sustainability*, 3(7), Article 7. <https://doi.org/10.1038/s41893-020-0508-7>

Mendoza-Tinoco, D., Hu, Y., Zeng, Z., Chalvatzis, K. J., Zhang, N., Steenge, A. E., & Guan, D. (2020).

Flood Footprint Assessment: A Multiregional Case of 2009 Central European Floods. *Risk Analysis*, *40*(8), 1612–1631. <https://doi.org/10.1111/risa.13497>

Mrozek, J. R., & Taylor, L. O. (2002). What determines the value of life? A meta-analysis. *Journal of Policy Analysis and Management*, *21*(2), 253–270. <https://doi.org/10.1002/pam.10026>

Orlov, A., Sillmann, J., Aunan, K., Kjellstrom, T., & Aaheim, A. (2020). Economic costs of heat-induced reductions in worker productivity due to global warming. *Global Environmental Change*, *63*, 102087. <https://doi.org/10/gmhq8z>

Parsons, L. A., Shindell, D., Tigchelaar, M., Zhang, Y., & Spector, J. T. (2021). Increased labor losses and decreased adaptation potential in a warmer world. *Nature Communications*, *12*(1), Article 1. <https://doi.org/10/gphstd>

Romanello, M., McGushin, A., Di Napoli, C., Drummond, P., Hughes, N., Jamart, L., Kennard, H., Lampard, P., Solano Rodriguez, B., Arnell, N., Ayeb-Karlsson, S., Belesova, K., Cai, W., Campbell-Lendrum, D., Capstick, S., Chambers, J., Chu, L., Ciampi, L., Dalin, C., ... Hamilton, I. (2021). The 2021 report of the Lancet Countdown on health and climate change: Code red for a healthy future. *The Lancet*, *398*(10311), 1619–1662. [https://doi.org/10.1016/S0140-6736\(21\)01787-6](https://doi.org/10.1016/S0140-6736(21)01787-6)

Rose, A. (2004). Economic Principles, Issues, and Research Priorities in Hazard Loss Estimation. In in *Modeling Spatial and Economic Impacts of Disasters* (eds. Okuyama, Y. & Chang, S. E.) 13–36 (Springer, 2004). doi:10.1007/978-3-540-24787-6_2.

Steenge, A. E., & Bočkarjova, M. (2007). Thinking about Imbalances in Post-catastrophe Economies: An Input–Output based Proposition. *Economic Systems Research*, *19*(2), 205–223. <https://doi.org/10.1080/09535310701330308>

- Takakura, J., Fujimori, S., Takahashi, K., Hasegawa, T., Honda, Y., Hanasaki, N., Hijioka, Y., & Masui, T. (2018). Limited Role of Working Time Shift in Offsetting the Increasing Occupational-Health Cost of Heat Exposure. *Earth's Future*, 6(11), 1588–1602. <https://doi.org/10.1029/2018EF000883>
- Wang, D., Guan, D., Zhu, S., Kinnon, M. M., Geng, G., Zhang, Q., Zheng, H., Lei, T., Shao, S., Gong, P., & Davis, S. J. (2021). Economic footprint of California wildfires in 2018. *Nature Sustainability*, 4(3), Article 3. <https://doi.org/10.1038/s41893-020-00646-7>
- Watts, N., Amann, M., Arnell, N., Ayeb-Karlsson, S., Beagley, J., Belesova, K., Boykoff, M., Byass, P., Cai, W., Campbell-Lendrum, D., Capstick, S., Chambers, J., Coleman, S., Dalin, C., Daly, M., Dasandi, N., Dasgupta, S., Davies, M., Di Napoli, C., ... Costello, A. (2021). The 2020 report of The Lancet Countdown on health and climate change: Responding to converging crises. *The Lancet*, 397(10269), 129–170. <https://doi.org/10/fmb7>
- Watts, N., Amann, M., Arnell, N., Ayeb-Karlsson, S., Belesova, K., Boykoff, M., Byass, P., Cai, W., Campbell-Lendrum, D., Capstick, S., Chambers, J., Dalin, C., Daly, M., Dasandi, N., Davies, M., Drummond, P., Dubrow, R., Ebi, K. L., Eckelman, M., ... Montgomery, H. (2019a). The 2019 report of The Lancet Countdown on health and climate change: Ensuring that the health of a child born today is not defined by a changing climate. *The Lancet*, 394(10211), 1836–1878. [https://doi.org/10.1016/s0140-6736\(19\)32596-6](https://doi.org/10.1016/s0140-6736(19)32596-6)
- Watts, N., Amann, M., Arnell, N., Ayeb-Karlsson, S., Belesova, K., Boykoff, M., Byass, P., Cai, W., Campbell-Lendrum, D., Capstick, S., Chambers, J., Dalin, C., Daly, M., Dasandi, N., Davies, M., Drummond, P., Dubrow, R., Ebi, K. L., Eckelman, M., ... Montgomery, H. (2019b). The 2019 report of The Lancet Countdown on health and climate change: Ensuring that the

health of a child born today is not defined by a changing climate. *The Lancet*, 394(10211), 1836–1878. [https://doi.org/10.1016/s0140-6736\(19\)32596-6](https://doi.org/10.1016/s0140-6736(19)32596-6)

Xia, Y., Guan, D., Steenge, A. E., Dietzenbacher, E., Meng, J., & Mendoza Tinoco, D. (2019). Assessing the economic impacts of IT service shutdown during the York flood of 2015 in the UK. *Proceedings of the Royal Society A: Mathematical, Physical and Engineering Sciences*, 475(2224), 20180871. <https://doi.org/10.1098/rspa.2018.0871>

Yin, H., Brauer, M., Zhang, J. (Jim), Cai, W., Navrud, S., Burnett, R., Howard, C., Deng, Z., Kammen, D. M., Schellnhuber, H. J., Chen, K., Kan, H., Chen, Z.-M., Chen, B., Zhang, N., Mi, Z., Coffman, D., Cohen, A. J., Guan, D., ... Liu, Z. (2021). Population ageing and deaths attributable to ambient PM_{2.5} pollution: A global analysis of economic cost. *The Lancet Planetary Health*, 5(6), e356–e367. [https://doi.org/10.1016/S2542-5196\(21\)00131-5](https://doi.org/10.1016/S2542-5196(21)00131-5)

Zhao, M., Lee, J. K. W., Kjellstrom, T., & Cai, W. (2021). Assessment of the economic impact of heat-related labor productivity loss: A systematic review. *Climatic Change*, 167(1–2), 22. <https://doi.org/10/gm5qrt>

Reviewer Reports on the First Revision:

Referees' comments:

Referee #1 (Remarks to the Author):

Dear authors,

Thank you for the work you have done on the manuscript. I am actually very impressed by the level of detail and rigorous work that you have put in answering my concerns. Instead of trying to dodge them, you did a full analysis to show proof why your choices are justified. Well done.

I was triggered though by your answer on the VOSL: "might lead to inequality in the calculations between developed and developing countries, ultimately underestimating global economic losses.". On what is this statement with respect to underestimating global economic losses based? Validation numbers, in particular on economic losses, are scarce. How do you know you are underestimating losses? I fully agree that natural hazards and climate extremes are causing large impacts to our (global) economy, but we should be cautious with just searching for the largest values.

In line with my previous comment, there is one big thing missing: validation of your results. The word validate or validation is actually not even in the manuscript, nor in the supplementary. I believe it is very important to give the reader some context on whether these numbers are indeed in line with what we can expect in the real world. We know that modelled results are never the same as real-world losses, but are these numbers in the same ball-park as what we have seen in the past, or what we perhaps can extract from events in the past? Even comparing the numbers with two or three specific events in some specific countries would already help to understand whether these numbers are indeed in line what we can expect (i.e. some anecdotal evidence to validate some results would already be better than nothing). Else we are just looking at model outputs again without knowing how much they make sense in the real world.

Some small comments:

- Some of the sentences in the abstract are a bit complex to read sometimes. But I assume that a journal like Nature will do some typesetting and will suggest perhaps some edits. A native speaker can probably provide better suggestions than I can do.
- Line 53: Personal preference perhaps, but I would refrain from using words like "Far worse". I think the sentence is just as clear without the "far worse".
- Figure 3: there is a typo I think. Should "Tpye" be "Type"?

Referee #2 (Remarks to the Author):

I thank the authors for the additional analyses conducted to address the concerns of both reviewers. They thereby demonstrate the validity of their findings for different health impact functions, trade network structures and substitutability assumptions. They also clarify in their response to the reviews that the intention of the paper is not to improve the ARIO model, but to apply it.

However, I am puzzled by the fact that only marginal changes were made to the main text: all new analyzes and figures were only added to the supplementary material and even there these figures were not properly integrated (for instance the cluster analysis is added but afterwards the text and figures proceed with the old country types 1-3; according to my inspection, India, Brazil and China are not even in the same cluster; similarly for Italy, Germany and Estonia); none of the figures in the main text were revised e.g. to capture uncertainties; no additional caveats of the analyses (e.g. arbitrary trade elasticities; no substitution in production) were added to the caveat paragraph in the discussion section but only a reference is made to the supplementary material (lines 414-415).

Given the comments on the earlier version of the paper, I strongly suggest to add at least two uncertainty analyses to the main text, i.e. Figure S10 (GTAP 2011 vs GTAP 14) and Figure S11 (evolutionary trends in global economic losses). Similarly, countries picked in Figures 3-5 and Figures S19-S22 should represent different clusters according to Figure S18 and clusters/types/regions should be named consistently throughout the paper. In turn, I still find the two cases (Indian food production and Dominican tourism industry) too extensive and text could be reduced and the figures moved to the supplementary material.

Finally, I am still missing the diligence needed (imprecise language, typos, inconsistent time frames and country examples across analyzes, imprecise citations) for making the manuscript publishable, particularly in a journal like Nature.

For example, in the abstract, lines 25-27: How can global economic loss increase non-linearly by 0.16% per year? What is actually meant is that global economic loss increases from xx% per year in the 2030s to xx% in the 2060s. Similarly in lines 33-36: "[...] farming and construction in Tanzania are expected to lose 6.8% to 7.7% [...]" - this is not a confidence interval, but estimates for two different sectors. So it should be: "[...] farming and construction in Tanzania are expected to lose 6.8% and 7.7% respectively [...]"

As examples of inconsistent timelines: for example, Figures 1 and 3 display years 2040, 2050, 2060 whereas Figures 2, 4 and 5 refer to decadal means (2040s and 2060s respectively).

As examples of imprecise citations: In line 47-49: Knittel et al. 2020, Orlov et al. 2020, Takakura et al. 2018 are not biometereological studies but CGE analysis investigating heat induced labor productivity losses; in line 65-67: Garcia-Leon et al. 2021, Knittel et al 2020 and Takakura et al 2018 are examples of papers capturing the indirect effect, not ignoring it.

Author Rebuttals to First Revision:

Overall response to reviewers

Dear editor and reviewers,

Thank you very much for your consideration of our submission and your efforts in managing its review. We very much appreciate the reviews we received, which are very constructive and greatly improved our paper. We have added a large number of experiments, provided validation of the model using multiple sources of historical data. The article has also been restructured for clarity and completeness (robustness tests have been placed in the main text as suggested). All comments have been well addressed in the updated version of manuscript.

Referee #1 (Remarks to the Author):

Dear authors,

Thank you for the work you have done on the manuscript. I am actually very impressed by the level of detail and rigorous work that you have put in answering my concerns. Instead of trying to dodge them, you did a full analysis to show proof why your choices are justified. Well done.

I was triggered though by your answer on the VOSL: "might lead to inequality in the calculations between developed and developing countries, ultimately underestimating global economic losses.". On what is this statement with respect to underestimating global economic losses based? Validation numbers, in particular on economic losses, are scarce. How do you know you are underestimating losses? I fully agree that natural hazards and climate extremes are causing large impacts to our (global) economy, but we should be cautious with just searching for the largest values.

Thank you for the suggestion. We follow the VSL approach used in existing relevant studies, as improving the method to estimate the VSL is beyond the scope of this study. We fully understand your concerns and therefore we have removed the expression "ultimately underestimating global economic losses" from the text and highlighted in the SI that we should consider VSL estimates

based on different assumptions, and be cautious with just searching for the largest values.

The VSL can be estimated assuming either heterogeneous or equal individual preferences, from now on 'heterogeneous VSL' and 'equal VSL' for simplicity. Estimating the health losses using both assumptions offer a comprehensive picture and the possibility to draw reliable insights rather than trying to conclude whether one approach underestimates or overestimates the true economic losses. For example, the 'equal VSL' estimation reveals a very important insight: the inequality of health losses from climate change is likely to be higher in a future scenario where developing countries value human capital more than at present (getting closer to the global average). As a result, without a concerted global effort to reduce emissions, deaths due to warming and heatwaves in developing countries are likely to have an increasingly higher economic cost. We have therefore modified the relevant paragraphs in the SI.

2.1 Uncertainty and validation of health loss

Under a future global average VSL, the economic inequality of global health losses would become more pronounced due to the higher value of health losses in most LDCs compared to the value of health losses in most developed countries. Our findings indicate that global economic losses are 5-20% lower under a global 'equal VSL' compared to a global 'heterogeneous VSL'. (fig.S7).

This is because the global average VSL is significantly lower than the

'heterogeneous VSL' in heatwave vulnerable countries such as Russia, the USA and France, which have large populations and a high number of excess deaths due to heatwaves annually. Economic losses in these three countries would be reduced by 2 to 8 times under 'equal VSL'. Conversely, economic losses in heatwave-vulnerable countries like India, Bangladesh and Ethiopia would be 2 to 7 times higher under 'equal VSL', as the 'heterogeneous VSL' is 1.2 million USD for India and 0.4 million USD for Ethiopia.

The distribution of economic losses as a percentage of national GDP for each country under the two VSL approaches is illustrated in fig.S8 (patterns under

SSP119 and SSP245 scenarios are comparable). The 'equal VSL' assumption reveals that the inequality of health losses from climate change is likely to be greater in a future scenario where developing countries value human capital more than they currently do (approaching the global average). In many Latin American and African countries, for instance, economic losses amount to more than 10% of their GDP by midcentury. By 2060, economic losses in Tajikistan would represent 25.2% of GDP, while in Malawi, they would account for 65.9% of GDP. Consequently, without a coordinated global effort to reduce emissions, deaths due to warming and heatwaves in developing countries are likely to incur an increasingly high economic cost. We have adopted the heterogeneous VSL as benchmark VSL in the main text and used the global average VSL results as a reference for the supporting material.

In line with my previous comment, there is one big thing missing: validation of your results. The word validate or validation is actually not even in the manuscript, nor in the supplementary. I believe it is very important to give the reader some context on whether these numbers are indeed in line with what we can expect in the real world. We know that modelled results are never the same as real-world losses, but are these numbers in the same ball-park as what we have seen in the past, or what we perhaps can extract from events in the past? Even comparing the numbers with two or three specific events in some specific countries would already help to understand whether these numbers are indeed in line with what we can expect (i.e. some anecdotal evidence to validate some results would already be better than nothing). Else we are just looking at model outputs again without knowing how much they make sense in the real world.

Thank you for your advice. Please let us clarify that our study does not aim at predicting the true dynamics of GDP at regional or national levels; many of the costs we estimate are not included in GDP accounts. Existing studies do not

have comprehensive country-sector data on the various losses due to extreme heat, but there are global, or region/sector-specific, partial statistics from anecdotes, statistical reports, and counterfactual econometric studies.

In order to achieve the Validation as you requested, we have spent last two months in a full gear mode to compile records from thousands of documents, reports, and news items spanning the past three decades, matching all heatwave related events during the same time period. We have carried out a full simulation for the historical period from 1990 to 2020. We compiled 40,000 periods of maximum temperature, humidity, and other meteorological data at 0.25° resolution from ERA5, and have generated about two billion results about gridded productivity and health losses.

Similarly, historical period simulations were performed 30 times with different RR, exposure response functions, trade substitutability and other parameters. We used a 1000-core supercomputer that we ran for 2 weeks to calculate and compile the results.

Hundreds of pieces of useful information were compiled. Please see the following examples:

1. The report by the economic research company Prognos (cited by Germany's economy and environment ministries) shows that nine billion euros in damages were caused by workers' lower productivity in industry and commerce sectors due to heat waves in 2018 and 2019¹. In comparison, we estimate that the productivity losses were 10.7 billion euros.
2. The report by the Rockefeller Foundation Resilience Center found that worker productivity losses totalled approximately \$100 billion² in the United States in 2020. In comparison, we estimate that the productivity losses were \$72.1 billion.
3. Based on subnational economic data and empirical regression methods, Callahan & Mankin³ found that cumulative 1992–2013 losses from

anthropogenic extreme heat fall between \$5 trillion and \$29.3 trillion (2010\$) globally (average more than \$16 trillion). The paper was published in *Science Advance*. In comparison, the evaluation results of our model range from \$15.0 trillion to \$28.4 trillion (2010\$) globally.

4. Solomon (2010)⁴ used distributed-lag, autoregressive regression and found that in the Caribbean and Central America, output losses occurring in non-agricultural production ($-2.4\%/+1\text{ }^{\circ}\text{C}$) substantially exceed losses occurring in agricultural production ($-0.1\%/+1\text{ }^{\circ}\text{C}$), with 2.25% as indirect loss in the next year. The paper was published in *PNAS*. In comparison, our results show that economic losses in the Caribbean and Central America from 1990-2010 were 1.97% (0.91%~2.39%) per $^{\circ}\text{C}$.
5. Wholesale, retail, restaurants, and hotels show statistically significant responses to heat stress ($-6.1\%/+1\text{ }^{\circ}\text{C}$) in the Caribbean and Central America. Our assessment for the accommodation, food, and service activities sector in the Caribbean and Central America shows similar vulnerability (-3.5% to -5.7%/+1 $^{\circ}\text{C}$).
6. An empirical study based on US state panel data from 1997 to 2011 similarly found a cascading effect due to warming summer⁵. The article points out that the rise in average summer temperatures has a significant direct impact on labour productivity in the agricultural sector and, indirectly, in the food services and drinking sectors (-0.39% per $^{\circ}\text{F}$) due to fluctuations in supply and demand. In comparison, the evaluation results of our model are -0.28% per $^{\circ}\text{F}$.
7. Several assessments for the most impactful and well-known heatwave events between 2000 and 2022 as presented in figure S6. The best model performance is found for the 2020 Western Europe heatwave event, and the reported deaths from the 2022 Europe and 2006 Western Europe

heatwave events are within the uncertainty range of model results based on the 95% heatwave definition.

8. Anecdotes and news also reveal complex supply chain spillover effects. For example, the 2022 heatwave in Sichuan, China, has caused Toyota and Contemporary Amperex Technology, the world's largest battery maker, to suspend operations for one month. Numerous other manufacturers had to shut down during the blistering heat. SAIC Motor (China's largest automaker) and Tesla have also seen their operations in Shanghai, far from Sichuan, impacted because suppliers in the province were unable to ship needed parts⁶. These cases show the cascading effect of heat stress through the supply chain.

For full details of the verification please see the following paragraphs in SI:

1.1 Uncertainty and validation of health loss

1.2 Uncertainty and validation of labor productivity loss

1.3 Uncertainty and validation of indirect loss

Some small comments:

- Some of the sentences in the abstract are a bit complex to read sometimes. But I assume that a journal like Nature will do some typesetting and will suggest perhaps some edits. A native speaker can probably provide better suggestions than I can do.
- Line 53: Personal preference perhaps, but I would refrain from using words like "Far worse". I think the sentence is just as clear without the "far worse".
- Figure 3: there is a typo I think. Should "Tpye" be "Type"?

Thank you for your attentiveness. We have removed the redundant phrases. We have invited three native speakers to help us polish all the unclear expressions.

Referee #2 (Remarks to the Author):

I thank the authors for the additional analyses conducted to address the concerns of both reviewers. They thereby demonstrate the validity of their findings for different health impact functions, trade network structures and substitutability assumptions. They also clarify in their response to the reviews that the intention of the paper is not to improve the ARIO model, but to apply it.

However, I am puzzled by the fact that only marginal changes were made to the main text: all new analyzes and figures were only added to the supplementary material and even there these figures were not properly integrated (for instance the cluster analysis is added but afterwards the text and figures proceed with the old country types 1-3; according to my inspection, India, Brazil and China are not even in the same cluster; similarly for Italy, Germany and Estonia); none of the figures in the main text were revised e.g. to capture uncertainties; no additional caveats of the analyses (e.g. arbitrary

trade elasticities; no substitution in production) were added to the caveat paragraph in the discussion section but only a reference is made to the supplementary material (lines 414-415).

Thank you for your valuable advice. We were concerned about space constraints and put most of the content in the SI. This time we have added uncertainty analysis to the text as you suggested, refined the description according to the results of the cluster analysis and reduced the text.

1. For the clustering, we have modified the relevant graphics in the text and SI so that the examples for each category are based on the clustering results. Please see the following paragraph:

In **Results** section, **Asymmetric effects of heat stress on global supply chains**, line 239-354

2. For caveats about uncertainty, we have moved the Monte Carlo analysis to the main text and, using trade structure as an example, we have explained how the results are affected by different benchmark trade structures. We have also elaborated in the Discussion section against trade elasticity, production substitutability, see the following paragraphs:

In the **Discussion** section, line 395-404:

Considering the challenges of predicting changes to socioeconomic systems globally, we have followed the approach from the literature (García-León et al., 2021; Parsons et al., 2021) to simulate supply chain indirect losses by considering the impact of future climate risks on current socioeconomic settings. We have not considered the potential substitution of labour with capital resulting from technological advances, such as mechanization.

Our analysis ignores the different levels of trade openness and globalization among SSP narratives, as well as the role of dynamic factors such as technology and price. Again, although we have done robustness tests for different degrees of trade substitutability, the relevant parameter is set randomly in the Monte

Carlo simulation rather than derived through a general equilibrium model.

Given the comments on the earlier version of the paper, I strongly suggest to add at least two uncertainty analyses to the main text, i.e. Figure S10 (GTAP 2011 vs GTAP 14) and Figure S11 (evolutionary trends in global economic losses). Similarly, countries picked in Figures 3-5 and Figures S19-S22 should represent different clusters according to Figure S18 and clusters/types/regions should be named consistently throughout the paper. In turn, I still find the two cases (Indian food production and Dominican tourism industry) too extensive and text could be reduced and the figures moved to the supplementary material.

Thank you very much for your valuable comments to improve the structure of the article! We have followed your suggestion and added Figures S10 and S11 to the main text, along with additional information. For example, the impact of different trade structures on each of the global economies in addition to those with the highest losses. Overly detailed descriptions have also been removed. Please see the following paragraphs:

✧ In **Discussion** line 405-451:

To quantify some of the uncertainties, we conducted a comprehensive sensitivity analysis. Details are available in the Methods and Supplementary Information. While the ARIO model is widely used and work well for single- country/single-region analyses, the substitutability of products in a multi- country scenario requires further discussion to verify robustness. To analyse the uncertainty in the production and trade structure, we have used different years and versions of the input-output database for comparison tests (see Fig.).

Globally, the estimate of the total amount of indirect losses is robust to changes in the data used (GTAP2011 and GTAP2014) for the base period. The results of the loss assessment at global scale differ by less than 5% in 2060. Most

countries are distributed around the $y=x$ line, which suggests a consistent assessment across different trade structures.

Regionally, for a small number of countries, indirect loss assessments can show larger differences. By comparison, we find that when using GTAP 2014 data for the base period, indirect economic losses in East and Southeast Asian countries such as Singapore, Korea, Japan and Myanmar are amplified (see Fig. and fig.S12). This is explained by the fact that, in GTAP2014, those countries have closer economic ties with climate change sensitive markets including Malaysia, China, India and Vietnam. For example, trade between Singapore and emerging economies such as China and Vietnam had increased substantially during this period of 2010 to 2014. According to the Singapore Department of Statistics (DOS, <https://www.singstat.gov.sg/>) and the United Nations Commodity Trade Statistics Database (UN Comtrade, <https://comtrade.un.org/>), China became the largest trading partner of Singapore in 2014, up from 4th place in 2011, whereas Vietnam became the 13th largest partner in 2014, up from the 20th place in 2011. On the contrary, the share of Singapore's total trade with the EU and the United States had decreased slightly over the same period. Similarly, Japan, Korea and Myanmar developed closer trade relationships with emerging markets such as China, India and Vietnam. The assessment of indirect losses under different trade relationships offers important insights about the likely supply chain risks posed by climate change. As Africa, South America and South East Asia become increasingly involved in global value chains (GVCs), the impact of climate change on the resilience of GVCs needs to be properly assessed, rather than just considered in terms of scale effects and comparative advantage about economic efficiency.

For parameters such as the maximum stock ratio and excess production capacity, we repeated the experiment several times within the range of possible values from previous studies. For trade substitutability, upper and lower bounds of perfect substitution and non-substitution (traditional static IO model) were

used. We elaborate in more detail about the intervals of the uncertainty parameter for the three main modules and perform a Monte Carlo analysis, including simulation of economic loss dynamics for 10000 periods. We have also conducted a historical validation utilizing multiple authentic data sources (see robustness tests and validation in SI), encompassing government statistics, empirical studies and institution reports (see table S10), in addition to a comparative analysis of previous studies concerning future periods based on CMIP5 data and similar RCP scenarios (see fig.5).

- ✧ **Asymmetric effects of heat stress on global supply chains**, line 239-354
- ✧ In SI, **7. Sectoral loss patterns in different types of countries**, Fig.S21 to Fig.S24. Due to its length, it is not shown here.

Finally, I am still missing the diligence needed (imprecise language, typos, inconsistent time frames and country examples across analyzes, imprecise citations) for making the manuscript publishable, particularly in a journal like Nature.

For example, in the abstract, lines 25-27: How can global economic loss increase non-linearly by 0.16% per year? What is actually meant is that global economic loss increases from xx% per year in the 2030s to xx% in the 2060s. Similarly in lines 33-36: "[...] farming and construction in Tanzania are expected to lose 6.8% to 7.7% [...]"- this is not a confidence interval, but estimates for two different sectors. So it should be: "[...] farming and construction in Tanzania are expected to lose 6.8% and 7.7% respectively [...]". As examples of inconsistent timelines: for example, Figures 1 and 3 display years 2040, 2050, 2060 whereas Figures 2, 4 and 5 refer to decadal means (2040s and 2060s respectively).

As examples of imprecise citations: In line 47-49: Knittel et al. 2020, Orlov et al. 2020, Takakura et al. 2018 are not biometereological studies but CGE analysis investigating heat induced labor productivity losses; in line 65-67: Garcia-Leon et al. 2021, Knittel et al 2020 and Takakura et al 2018 are examples of papers capturing the indirect effect, not ignoring it.

Thank you for your comments. In terms of language, we have invited three native speakers to go through the entire documents several times to polish our expressions and descriptions. We believe we have done our best to improve the quality of the manuscript., please see the following paragraphs:

In Abstract

Our results show that global economic losses due to increasingly severe heat stress will rise non-linearly. Specifically, we estimate a 0.15 percentage points increase in global GDP losses annually from 2050 to 2060, a threefold increase compared to 2030-2040 projection (0.05 percentage points) under SSP585 scenario. By 2060, the expected economic losses reach a total of 4.0% (ranging from 3.0% to 4.4%), with losses attributed to indirect loss (43%), health loss (40%) and labor productivity loss (17%).

Severely affected industries such as Tanzanian farming and Indian ferrous metals industry are expected to lose 6.3% and 4.3% of sectoral value added per year respectively.

In terms of timelines, we illustrate in the legend of each diagram that "The values displayed are 10-year averages (for example loss reported in 2060 represents the average loss calculated over the period from 2055 to 2065)."

In terms of citations, please see the following paragraphs:

Line 49-54: On the other hand, biometeorological studies suggest that heat stress can seriously decrease labour productivity⁹⁻¹², measured in terms of lost worktime from recommended work/rest ratios during heat stress, reduced work efficiency as estimated from exposure-response functions, and self-reported

reduced work efficiency ^{9,13,14}.

Line 64-69: The direct mortality and productivity loss resulting from heat stress have been extensively studied. However, the indirect losses due to supply chain disruptions have not been fully analyzed ^{15,16}, as previous literature has either devoted insufficient discussion to the indirect effects by only reporting the total/net effects ^{7,17,18}, or ignored the amplifying effect of the global trade system on direct losses.

References

1. Reuters. Climate change extreme weather costs Germany billions annually. *Reuters* (2022).
2. Hot cities, chilled economies: Impacts of extreme heat on global cities. *Arsht-Rock* <https://onebillionresilient.org/hot-cities-chilled-economies/>.
3. Callahan, C. W. & Mankin, J. S. Globally unequal effect of extreme heat on economic growth. *Science Advances* **8**, eadd3726 (2022).
4. Hsiang, S. M. Temperatures and cyclones strongly associated with economic production in the Caribbean and Central America. *Proceedings of the National Academy of Sciences* **107**, 15367–15372 (2010).
5. Colacito, R., Hoffmann, B. & Phan, T. Temperature and Growth: A Panel Analysis of the United States. *Journal of Money, Credit and Banking* **51**, 313–368 (2019).
6. Kahn, B. China's supply chain is melting in extreme heat. Whose will be next? <https://www.protocol.com/climate/china-heat-wave-supply-chain> (2022).
7. García-León, D. *et al.* Current and projected regional economic impacts of heatwaves in Europe. *Nat Commun* **12**, 5807 (2021).
8. Parsons, L. A., Shindell, D., Tigchelaar, M., Zhang, Y. & Spector, J. T. Increased labor losses and decreased adaptation potential in a warmer world. *Nat Commun* **12**, 7286 (2021).
9. Dunne, J. P., Stouffer, R. J. & John, J. G. Reductions in labour capacity from

- heat stress under climate warming. *Nature Climate Change* **3**, 563–566 (2013).
10. Kjellstrom, T., Freyberg, C., Lemke, B., Otto, M. & Briggs, D. Estimating population heat exposure and impacts on working people in conjunction with climate change. *Int J Biometeorol* **62**, 291–306 (2018).
 11. Lee, S.-W., Lee, K. & Lim, B. Effects of climate change-related heat stress on labor productivity in South Korea. *Int J Biometeorol* **62**, 2119–2129 (2018).
 12. Nunfam, V. F., Adusei-Asante, K., Frimpong, K., Van Etten, E. J. & Oosthuizen, J. Barriers to occupational heat stress risk adaptation of mining workers in Ghana. *Int J Biometeorol* **64**, 1085–1101 (2020).
 13. Borg, M. A. *et al.* Occupational heat stress and economic burden: A review of global evidence. *Environmental Research* **195**, 110781 (2021).
 14. Kjellstrom, T., MSc, R. S. K., MSc, S. J. L., PhD, T. H. & PhD, R. S. J. T. The Direct Impact of Climate Change on Regional Labor Productivity. *Archives of Environmental & Occupational Health* **64**, 217–227 (2009).
 15. Wang, D. *et al.* Economic footprint of California wildfires in 2018. *Nature Sustainability* **4**, 252–260 (2021).
 16. Xia, Y. *et al.* Assessing the economic impacts of IT service shutdown during the York flood of 2015 in the UK. *Proc. R. Soc. A.* **475**, 20180871 (2019).
 17. Knittel, N., Jury, M. W., Bednar-Friedl, B., Bachner, G. & Steiner, A. K. A global analysis of heat-related labour productivity losses under climate change—implications for Germany’s foreign trade. *Climatic Change* **160**, 251–269 (2020).
 18. Takakura, J. *et al.* Limited Role of Working Time Shift in Offsetting the Increasing Occupational-Health Cost of Heat Exposure. *Earth’s Future* **6**, 1588–1602 (2018).

Reviewer Reports on the Second Revision:

Referees' comments:

Referee #3 (Remarks to the Author):

I would like to thank the authors for a very interesting and well-done paper. I was asked to step in for a reviewer who previously commented and reviewed the IO modeling. Therefore, I will keep my comments limited to that portion of the paper. I find it unfair for the authors to consider my comments on other portions as this stage of the review process, so my review is very limited.

My experience as a researcher has been primarily limited to IO and MRIO modeling in the United States, so I am less familiar with the databases used by the authors in this paper. After reading the background papers on the GTAP 10 database, I am confident that this is an appropriate source for this type of paper. Perhaps an additional citation or two of how GRAP 10 (or earlier versions) have been used would be beneficial, especially in applications beyond five years--as noted below.

I do have some concerns about the extension of IO modeling, even with the caveats noted here, until 2060. This assumption, unless I missed something, appears to be based on the underlying economic structure from the 2014 GTAP model. Given the static nature of IO modeling, most papers caution on extending these models more than 5 years.

For example,

Kay, D., & Jolley, G. J. (2023). Using input–output models to estimate sectoral effects of carbon tax policy: Applications of the NGFS scenarios. *American Journal of Economics and Sociology*, 82(3), 187-222.

In this paper, Kay and Jolley note that the increase in pricing associated with a carbon tax in the United States would be short run and likely upper bound estimates, as industries would adjust.

A good rule of thumb for IO modeling in the US is to not extend analyses past 5 years, which is how often the underlying IO matrix is updated by BEA. I do not have a specific citation off hand for this rule given its common nature.

Given the static nature of IO modeling, this leaves three options in my view: 1) reject the paper, 2) suggest a different method or 3) provide additional caveats and acknowledgments of IO/MIRO/ARIO limitations. In my view, this paper is too important to reject and I am unaware of any dynamic modeling data (e.g., a global REMI model) with this degree of specificity (e.g., the IMPLAN vs. REMI debate in the US). Therefore, I suggest additional caveats and acknowledgments of the limitations of ARIO are needed.

I suggest that the authors include a few citations about IO modeling and its time limitations due to the static nature of the models and existing economic structure. Given the limited availability of global dynamic economic models, the authors can then note why they used the GTAP 10 model and

the limitations of extending it. I do not see these clarifications as a substantial rewrite, but rather a modest acknowledgment of the limited nature of the best available data.

Referee #4 (Remarks to the Author):

GENERAL COMMENTS

This paper presents an extensive analysis combining climate, epidemiological, and economic models to estimate the mid-century economic impacts of heat stress and the associated socio-economic costs. However, there are some flaws which, in this Reviewer's view, prohibit its publication at its present state. A detailed list of points is provided below. Despite these limitations, the results presented are of immediate interest to many people across different scientific disciplines as well as policy makers. The most outstanding and innovative feature of this work is that it quantifies indirect effects that were not analyzed before, providing insight into the far-reaching impacts of heat stress across global supply chains and how such impacts evolve spatially and over long-time scales.

SPECIFIC COMMENTS

L45: Consider citing also a previous systematic review and meta-analysis on the impacts of heat stress on health and productivity: Flouris AD, Dinas PC, Ioannou LG, Nybo L, Havenith G, Kenny GP, Kjellstrom T. Workers' health and productivity under occupational heat strain: a systematic review and meta-analysis. *Lancet Planet Health*. 2018 Dec;2(12):e521-e531.

L79: The authors should confirm that the cited study is appropriate to support this model. To this Reviewer's knowledge, the cited paper is not relevant to the CMIP6.

L108: It is not clear why health losses in Fig1 are higher in Russia and Southern Africa, as dangerous heat levels are expected to be more prevalent in countries closer to the equator.

L159-160: This statement is unfounded since the EU is planning to phase out fossil fuels well ahead of 2050.

L178-181: It is unclear why Croatia ranks 1st or 2nd in the two scenarios of Fig2. What makes Croatia so vulnerable to health impacts from heat? Why is the frequency of extreme weather so much higher in this small country compared to other countries in the region?

L183: "...heatwaves occur over sudden and do not allow..." There must be an error here.

L241: Please replacing "Dlt_" which is typed before SSP585 and SSP245 in the figure legend with something more intuitive.

L492-493: This is a major limitation of the analysis and should be either rectified by reanalyzing the data or by highlighting this issue as a limitation in the Discussion. Previous studies (e.g., García-León et al., 2021) used the approach proposed by Liljegren et al. (2008) which is much more appropriate and considers the impacts of solar radiation. Calculating WBGT based on temperature and relative

humidity assumes that indoor and outdoor workers / occupations will be impacted to the same extent. As many recent studies have shown, this is not the case.

Referee #5 (Remarks to the Author):

As a climate scientist specialising in weather and climate extremes, I am not qualified to review this paper in its entirety and instead focus only on the elements I am familiar with – namely, heat extremes.

Overall I found the paper well-written, with the sections outside of my expertise presented clearly and insightfully. The figures are well thought through, and clearly presented.

The first sentence of both the abstract and the main text is worded to make it sound like there has been a recent step-change in the climate and heatwaves. This is not true. Trends in temperature, and associated extremes, have been known about since the 1930s (<https://doi.org/10.1002/qj.49706427503>). As the two sentences are so prominent I think they should be reworded to convey that the increase in heat extremes has been known about for a long time. I am not keen on the use of 'recent' or the use of perfect tense 'have been'. The increases in extremes are still on-going.

Lines 181-189 - some clumsy working in this section, particularly 'This is because heatwaves occur over sudden...'. More evidence to support these statements would be helpful, particularly in stating Portugal and South Africa are more vulnerable, and 'climate is not very hot' (it is all relative). I think the use of 'scorching' needs some kind of definition or description.

Method section 1.1, on the definition of heatwaves and the use of wet bulb, is very good. Choices do have to be made when assessing heat extremes, and the choices made in this paper are well backed-up and allow for the global coverage of the study well.

One small comment on the grammar - you use UK / U.K. / United Kingdom interchangeable, it would be better to stick with one. Same for US / U.S..

Dr. Vikki Thompson

Author Rebuttals to Second Revision:

Response to reviewers' comments

We greatly appreciate the constructive comments and suggestions provided by the three reviewers. We have addressed all issues raised and revised the manuscript thoroughly. Below we include our point-to-point responses to the reviewers' comments.

Referee #3 (Remarks to the Author):

I would like to thank the authors for a very interesting and well-done paper. I was asked to step in for a reviewer who previously commented and reviewed the IO modeling. Therefore, I will keep my comments limited to that portion of the paper. I find it unfair for the authors to consider my comments on other portions as this stage of the review process, so my review is very limited.

My experience as a researcher has been primarily limited to IO and MRIO modeling in the United States, so I am less familiar with the databases used by the authors in this paper. After reading the background papers on the GTAP 10 database, I am confident that this is an appropriate source for this type of paper. Perhaps an additional citation or two of how GTAP 10 (or earlier versions) have been used would be beneficial, especially in applications beyond five years--as noted below.

Response

We appreciate the comments. We have added 4 new citations (there could be more, but we are limited by the limited number of references allowed by Nature). Xie et.al¹ used GTAP7 (2004 economic structure) to evaluate the effects of drought and heat extremes on beer supply and trade projected under future climate scenarios till 2100. García-León et.al² used GTAP8 (2007 economic structure) to assess economic losses in Europe till year 2060. Hallegatte³ used the ARIIO model to assess the economic impact ten years after Hurricane Katrina in 2008. Hsiang et al.⁴ use current economic data to derive empirically estimates of climate change-related economic damage in the United

States until 2100.

I do have some concerns about the extension of IO modeling, even with the caveats noted here, until 2060. This assumption, unless I missed something, appears to be based on the underlying economic structure from the 2014 GTAP model. Given the static nature of IO modeling, most papers caution on extending these models more than 5 years.

For example,

Kay, D., & Jolley, G. J. (2023). Using input–output models to estimate sectoral effects of carbon tax policy: Applications of the NGFS scenarios. *American Journal of Economics and Sociology*, 82(3), 187-222.

In this paper, Kay and Jolley note that the increase in pricing associated with a carbon tax in the United States would be short run and likely upper bound estimates, as industries would adjust.

A good rule of thumb for IO modeling in the US is to not extend analyses past 5 years, which is how often the underlying IO matrix is updated by BEA. I do not have a specific citation off hand for this rule given it's common nature.

Given the static nature of IO modeling, this leaves three options in my view: 1) reject the paper, 2) suggest a different method or 3) provide additional caveats and acknowledgments of IO/MIRO/AIRO limitations. In my view, this paper is too important to reject and I am unaware of any dynamic modeling data (e.g., a global REMI model) with this degree of specificity (e.g., the IMPLAN vs. REMI debate in the US). Therefore, I suggest additional caveats and acknowledgments of the limitations of ARIIO are needed.

I suggest that the authors include a few citations about IO modeling and its time limitations due to the static nature of the models and existing economic structure. Given the limited availability of global dynamic economic models, the authors can then note why they used the GTAP 10 model and the limitations of extending it. I do not see these clarifications as a substantial rewrite, but rather a modest acknowledgment of the limited nature of the best available data.

Response

Thanks for the suggestions. During the first round of review the referee put forward similar questions about the static nature of IO modeling. We fully acknowledge the limitation and caveats by adding additional comparison about ARIO and CGE based modelling results in SI section 10. In line with your advice here, for comparison and validation purposes, we run (additionally) a dynamic GTAP-based CGE model (ten regions by ten sectors), which considers economic structure changes, substitution and price dynamics, (see 2.3 Uncertainty and validation of indirect loss in SI). We compare and found similar results and explained the reasons in detail in the SI. We understand this is far beyond from accurate economic modelling, but we have tried our best. We acknowledge the limitations and add additional caveats in the main text, as suggested (see line 847-859 in Method).

Line 847-859

Considering the challenges of predicting changes to socioeconomic systems globally, we have followed the approach from the literature ^{1,2,4,5} to simulate supply chain indirect losses by considering the impact of future climate risks on current socioeconomic settings. We have not considered the potential substitution of labour with capital resulting from technological advances, such as mechanization. Our analysis ignores the different levels of trade openness and globalization among SSP narratives, as well as the role of dynamic factors such as technology and price. Again, although we have conducted robustness tests for different degrees of trade substitutability, the relevant parameter is set randomly in the Monte Carlo simulation rather than derived through a

general equilibrium model. The results should therefore be interpreted with caution as indicating potential future climate change risks to the existing economy rather than as quantitative predictions, given that the static representation of the economic structure in our model inevitably skews the assessment in the long run.

Referee #4 (Remarks to the Author):

GENERAL COMMENTS

This paper presents an extensive analysis combining climate, epidemiological, and economic models to estimate the mid-century economic impacts of heat stress and the associated socio-economic costs. However, there are some flaws which, in this Reviewer's view, prohibit its publication at its present state. A detailed list of points is provided below. Despite these limitations, the results presented are of immediate interest to many people across different scientific disciplines as well as policy makers. The most outstanding and innovative feature of this work is that it quantifies indirect effects that were not analyzed before, providing insight into the far-reaching impacts of heat stress across global supply chains and how such impacts evolve spatially and over long-time scales.

Response

Thank you for the positive and constructive comments.

SPECIFIC COMMENTS

L45: Consider citing also a previous systematic review and meta-analysis on the impacts of heat stress on health and productivity: Flouris AD, Dinas PC, Ioannou LG, Nybo L, Havenith G, Kenny GP, Kjellstrom T. Workers' health and productivity under occupational heat strain: a systematic review and meta-analysis. *Lancet Planet Health*. 2018 Dec;2(12):e521-e531.

Response

Thanks for the advice. This article provides a valuable systematic review and meta-analysis of occupational heat strain on workers' health and productivity outcomes. We have added the reference in line 47.

L79: The authors should confirm that the cited study is appropriate to support this model. To this Reviewer's knowledge, the cited paper is not relevant to the CMIP6.

Response

Thanks for the advice. We have included the following two references for the CMIP6 model development and evaluation in line 75:

30. Eyring, V. *et al.* Overview of the Coupled Model Intercomparison Project

Phase 6 (CMIP6) experimental design and organization. *Geoscientific Model*

Development **9**, 1937–1958 (2016).

31. Fasullo, J. T. Evaluating simulated climate patterns from the CMIP archives

using satellite and reanalysis datasets using the Climate Model Assessment

Tool (CMATv1). *Geoscientific Model Development* **13**, 3627–3642 (2020).

L108: It is not clear why health losses in Fig1 are higher in Russia and Southern Africa, as dangerous heat levels are expected to be more prevalent in countries closer to the equator.

Response

Thank you. In the previous analysis we used aggregated mortality rates (country level). In this revision, we have contacted directly regional / local health authorities and have been able to update information for over 30 countries (e.g. China, Brazil, Russia, Southern Africa, India and others) to a

much finer regional / local scale (see comparison maps below). The detailed procedures and corresponding results are presented in the following Figure.

Country-level mortality statistics (a). Mortality statistics at the subregional level based on official national statistics (b).

We have also run (additionally) all 14 available climate models (with bias-corrected) under the three SSP scenarios (refer to Extended Data Table 1) in CMIP6, ranging from 0.5 degree to 2.5 degree, and additionally secured the finest gridded population data with 0.125-degree (instead of 0.5 degree in previous runs) resolution, see updated result below. We have updated the results in the main text accordingly. We can now see that health losses in south-central Africa are the highest globally, while Eastern European countries including Hungary, Romania and Croatia have the highest health losses at similar latitudes. Some specific results are below (also available at SI 2.1): Health loss is 8.1% (7.1% to 10.6%) in Angola, possibly the highest in Central and Southern Africa. Health loss is 4.2% (3.7% to 4.8%) in Hungary, 3.5% (2.4% to 4.1%) in Croatia and 2.8% (2.4% to 3.4%) in Russia, compared with 1.2% (0.9% to 1.4%) in China and 1.1% (0.8% to 1.4%) in USA.

L159-160: This statement is unfounded since the EU is planning to phase out fossil fuels well ahead of 2050.

Response

Thank you for your advice. We have deleted this sentence from the text.

L178-181: It is unclear why Croatia ranks 1st or 2nd in the two scenarios of Fig2. What makes Croatia so vulnerable to health impacts from heat? Why is the frequency of extreme weather so much higher in this small country compared to other countries in the region?

Thank you. As discussed in previous comments, we have now increased the resolutions of mortality rates and population grid, and our updated results shows that health loss of Croatia is 1.1% (0.6% to 1.5%) in 2060 under SSP119, ranging to 3.5% (2.4% to 4.1%) under SSP585.

L183: "...heatwaves occur over sudden and do not allow..." There must be an error here.

Thank you. We have rephrased as:

For example, Hungary and Croatia suffer considerable health losses, even though in these countries the climate is cooler than in the Middle East and North Africa.

L241: Please replacing "Dlt_" which is typed before SSP585 and SSP245 in the figure legend with something more intuitive.

Thank you for your advice. We have updated consistently the text by using "delta (Δ)" throughout the paper

L492-493: This is a major limitation of the analysis and should be either rectified by reanalyzing the data or by highlighting this issue as a limitation in the Discussion. Previous studies (e.g., García-León et al., 2021) used the approach proposed by Liljegren et al. (2008) which is much more appropriate and considers the impacts of solar radiation. Calculating WBGT based on temperature and relative humidity assumes that indoor and outdoor workers / occupations will be impacted to the same extent. As many recent studies have shown, this this not the case.

Response

Thank you for your advice. We fully acknowledge that the simplified version of the WBGT formula we used does not consider the effect caused by background radiation and wind speed, but it distinguishes between indoor and outdoor labor losses⁶. The simplified WBGT formula has been widely used by the Australian Bureau of Meteorology^{7,8}, Kjellstrom⁶ (2009), Chavaillaz et,al.⁹ (2019), Dasgupta (2021)¹⁰, Parsons et,al.⁵ (2021), and others^{11,12}. Previous studies also suggested that the effects due to changes of solar radiation and wind speed under climate change are smaller compared with temperature and humidity¹³⁻¹⁶. The variability of solar radiation is mainly due to changes in cloudiness and aerosols, which might introduce much more uncertainty^{17,18} in global scale estimations.

Since the advancement of WBGT is not the aim of our paper, we shall acknowledge the caveats in Supplementary Information section 1.2, line 173-178.

Supplementary Information section 1.1, line 173-178

...In addition, the simplified WBGT model only quantify the effects of changes in temperature and humidity due to climate change. Other meteorological variables, such as strong radiation or low wind speed, can exacerbate heat stress, although it primarily depends on temperature and humidity^{13,19}. The simplified formula assumes moderately high levels of heat radiation in light wind conditions, disregarding long-term trends of changes in solar radiation and wind speed.

Referee #5 (Remarks to the Author):

As a climate scientist specialising in weather and climate extremes, I am not qualified to review this paper in its entirety and instead focus only on the elements I am familiar with – namely, heat extremes.

Overall I found the paper well-written, with the sections outside of my expertise presented clearly and insightfully. The figures are well thought through, and clearly presented.

The first sentence of both the abstract and the main text is worded to make it sound like there has been a recent step-change in the climate and heatwaves. This is not true. Trends in temperature, and associated extremes, have been known about since the 1930s (<https://doi.org/10.1002/qj.49706427503>). As the two sentences are so prominent I think they should be reworded to convey that the increase in heat extremes has been known about for a long time. I am not keen on the use of 'recent' or the use of perfect tense 'have been'. The increases in extremes are still on-going.

Response

Thank you. We have carefully revised the Abstract and Introduction sections in accordance with your comments, please see the following paragraphs:

In Abstract:

Evidence shows an ongoing increase in the frequency and severity of global heatwaves, raising concerns about the future impacts of climate change and the associated socio-economic costs.

In Introduction:

Research has been showing a trend in rising temperature and increasing occurrence of extreme heatwaves since the 1930s. This continuous pattern raises concerns about the potential impacts of climate change and its associated socio-economic costs.

Lines 181-189 - some clumsy working in this section, particularly 'This is because heatwaves occur over sudden...'. More evidence to support these statements would be helpful, particularly in stating Portugal and South Africa are more vulnerable, and 'climate is not very hot' (it is all relative). I think the use of 'scorching' needs some kind of definition or description.

Response

Thank you for pointing out the unclear statement. We have edited the unclear expression, please see the following paragraph in line 158-163:

Vulnerability to health impacts depends on the frequency of extreme weather events and the level of adaptive capacity. For example, Hungary and Croatia suffer considerable health losses, even though in these countries the climate is cooler than in the Middle East and North Africa. Unlike labour losses, which occur in regions with very high average temperature and humidity, health losses depend largely on the variance and abrupt changes in summer temperatures.

Method section 1.1, on the definition of heatwaves and the use of wet bulb, is very good. Choices do have to be made when assessing heat extremes, and the choices made in this paper are well backed-up and allow for the global coverage of the study well.

Response

Thank you for the positive comments.

One small comment on the grammar - you use UK / U.K. / United Kingdom interchangeably, it would be better to stick with one. Same for US / U.S..

Response

Thank you for your advice. We have harmonized all country abbreviations in this article.

Dr. Vikki Thompson

Reference

1. Xie, W. *et al.* Decreases in global beer supply due to extreme drought and heat. *Nature Plants* **4**, 964–973 (2018).
2. García-León, D. *et al.* Current and projected regional economic impacts of heatwaves in Europe. *Nat Commun* **12**, 5807 (2021).
3. Hallegatte, S. Modeling the Role of Inventories and Heterogeneity in the Assessment of the Economic Costs of Natural Disasters: Modeling the Role of Inventories and Heterogeneity. *Risk Analysis* **34**, 152–167 (2014).
4. Hsiang, S. *et al.* Estimating economic damage from climate change in the United States. *Science* **356**, 1362–1369 (2017).
5. Parsons, L. A., Shindell, D., Tigchelaar, M., Zhang, Y. & Spector, J. T. Increased labor losses and decreased adaptation potential in a warmer world. *Nat Commun* **12**, 7286 (2021).

6. Kjellstrom, T., MSc, R. S. K., MSc, S. J. L., PhD, T. H. & PhD, R. S. J. T. The Direct Impact of Climate Change on Regional Labor Productivity. *Archives of Environmental & Occupational Health* **64**, 217–227 (2009).
7. Australian Bureau of Meteorology. About the WBGT and Apparent Temperature Indices. http://www.bom.gov.au/info/thermal_stress/.
8. Prevention of thermal injuries during distance running. Position stand. American College of Sports Medicine. *Med J Aust* **141**, 876–879 (1984).
9. Chavaillaz, Y. *et al.* Exposure to excessive heat and impacts on labour productivity linked to cumulative CO2 emissions. *Sci Rep* **9**, 13711 (2019).
10. Dasgupta, S. *et al.* Effects of climate change on combined labour productivity and supply: an empirical, multi-model study. *The Lancet Planetary Health* **5**, e455–e465 (2021).
11. Zhao, Y., Ducharne, A., Sultan, B., Braconnot, P. & Vautard, R. Estimating heat stress from climate-based indicators: present-day biases and future spreads in the CMIP5 global climate model ensemble. *Environ. Res. Lett.* **10**, 084013 (2015).
12. Buzan, J. R., Oleson, K. & Huber, M. Implementation and comparison of a suite of heat stress metrics within the Community Land Model version 4.5. *Geoscientific Model Development* **8**, 151–170 (2015).
13. Casanueva, A. *et al.* Climate projections of a multivariate heat stress index: the role of downscaling and bias correction. *Geoscientific Model Development* **12**, 3419–3438 (2019).

14. Somanathan, E., Somanathan, R., Sudarshan, A. & Tewari, M. The Impact of Temperature on Productivity and Labor Supply: Evidence from Indian Manufacturing. *Journal of Political Economy* **129**, 1797–1827 (2021).
15. Matsumoto, K., Tachiiri, K. & Su, X. Heat stress, labor productivity, and economic impacts: analysis of climate change impacts using two-way coupled modeling. *Environ. Res. Commun.* **3**, 125001 (2021).
16. Zhang, P., Deschenes, O., Meng, K. & Zhang, J. Temperature effects on productivity and factor reallocation: Evidence from a half million chinese manufacturing plants. *Journal of Environmental Economics and Management* **88**, 1–17 (2018).
17. Gutiérrez, C. *et al.* Future evolution of surface solar radiation and photovoltaic potential in Europe: investigating the role of aerosols. *Environ. Res. Lett.* **15**, 034035 (2020).
18. He, Y. *et al.* Constrained future brightening of solar radiation and its implication for China's solar power. *National Science Review* **10**, nwac242 (2023).
19. Lemke, B. & Kjellstrom, T. Calculating Workplace WBGT from Meteorological Data: A Tool for Climate Change Assessment. *Industrial Health* **50**, 267–278 (2012).

Reviewer Reports on the Third Revision:

Referees' comments:

Referee #3 (Remarks to the Author):

I appreciate the authors responses to my questions about IO modeling and its limitations, especially the static nature of these models. I am confident that these shortcomings were adequately addressed. I still have some reservations about the extensions of IO modeling such a long distance into the future, yet these are not strong enough to warrant rejecting the paper. I find the assumptions made by the authors to be reasonable and transparent.

Referee #4 (Remarks to the Author):

The amendments made to the paper have addressed all the necessary key points.

Referee #5 (Remarks to the Author):

Thank-you for taking my comments into account and adjusting the paper accordingly. My only remaining comment is about the first sentence of the main text:

Research has been showing a trend in rising temperature and increasing occurrence of extreme heatwaves since the 1930s.

I think the dates may be misleading - it could be better to remove 'since the 1930s' or change it to '1950s' to align with the IPCC AR6 statement (chapter 11 executive summary):

The frequency and intensity of hot extremes (including heatwaves) have increased, and those of cold extremes have decreased on the global scale since 1950 (virtually certain). This also applies at regional scale, with more than 80% of AR6 regions showing similar changes assessed to be at least likely

Author Rebuttals to Third Revision:

Referees' comments:

Referee #3 (Remarks to the Author):

I appreciate the authors responses to my questions about IO modeling and its limitations, especially the static nature of these models. I am confident that these shortcoming were adequately addressed. I still have some reservations about the extensions of IO modeling such a long distance into the future, yet these are not strong enough to warrant rejecting the paper. I find the assumptions made by the authors to be reasonable and transparent.

Thank you for your advice. We have emphasized the caveats of the static nature of the hybrid model in the main text so that readers can grasp the scope of the model and its applicability.

Referee #4 (Remarks to the Author):

The amendments made to the paper have addressed all the necessary key points. Thank you for your advice and efforts.

Referee #5 (Remarks to the Author):

Thank-you for taking my comments into account and adjusting the paper accordingly. My only remaining comment is about the first sentence of the main text:

Research has been showing a trend in rising temperature and increasing occurrence of extreme heatwaves since the 1930s.

I think the dates may be misleading - it could be better to remove 'since the 1930s' or change it to '1950s' to align with the IPCC AR6 statement (chapter 11 executive summary):

The frequency and intensity of hot extremes (including heatwaves) have increased, and those of cold extremes have decreased on the global scale since 1950 (virtually certain). This also applies at regional scale, with more than 80% of AR6 regions showing similar changes assessed to be at least likely

Thank you for your advice. We have changed the relevant expression to '1950s' and added relative references to align with the IPCC AR6 statement